Resource

# Cell-specific DNA methylation in human alpha and beta cells regulates gene expression in type 2 diabetes

Jones K. Ofori [1,9], Sabrina Ruhrmann[1,9], Axel Lindström [1,2], Alexander Perfilyev[3], Melina Martin[1], Alexandros Karagiannopoulos [4], Lucia Scisciola [5], Katja Kost[1], Josefine Jönsson [1], Åsa Nilsson[4], Boris Kantor[6], Monika Dudenhöffer-Pfeifer[4], Marianne G. Rots[7], Anna Wendt[8], Tina Rönn[1], Lena Eliasson [2,8], Karl Bacos [1,10] & Charlotte Ling [1,10] ✉

Epigenome-wide studies of pancreatic islets provide valuable insights into type 2 diabetes (T2D) but lack methylomes from individual cell types. Here we show changes to alpha and beta cell-specific methylomes and transcriptomes from people with or without T2D, using whole-genome bisulfite sequencing and RNA sequencing. We discover 22,544 differentially methylated regions annotated to 7,975 genes in alpha versus beta cells, such as *INS*, *GCG*, *PDX1* and *PCSK1*, with ~50% showing differential expression. CRISPR–dCas9–DNMT3A-based epigenetic editing increases *INS* and *TH* DNA methylation, while CRISPR–dCas9–TET1-based editing decreases *GCG* methylation, each altering *INS, TH* or GCG expression and content in beta cells. Pre-T2D/T2D-associated differentially methylated regions in alpha and beta cells overlap 12–18% of T2D-associated genome-wide association study candidates. Additionally, *ONECUT2* is epigenetically upregulated in beta cells from people with pre-T2D/T2D and elevated in male Goto-Kakizaki rat islets. ONECUT2 overexpression in beta cells/islets downregulates gene sets impacting insulin secretion and glucose homeostasis, and reduces mitochondrial activity, ATP/ADP ratio and insulin secretion. We also provide 'alpha-beta-methylome' (https://alpha-beta-methylome.serve.scilifelab.se/app/alpha-beta-methylome/), a resource exploring T2D, age and sex associations on methylation, highlighting cell-specific epigenetic regulation and dysfunctions contributing to T2D.

T2D, characterized by hyperglycaemia driven by impaired insulin secretion from pancreatic islets and peripheral insulin resistance[1,2], is rapidly increasing worldwide[3]. Pancreatic islets contain several cell types controlling glucose homeostasis. In the fasted state, alpha cells secrete glucagon to raise plasma glucose levels, while in the fed state, beta cells secrete insulin to decrease glucose levels. Impaired insulin secretion is required for T2D development, but excess glucagon also contributes to hyperglycaemia[1,4].

Genome-wide association studies (GWAS) identified single-nucleotide polymorphisms (SNPs) linked to T2D and islet dysfunction[2]. Additionally, epigenetic mechanisms, linking the environment and genome, are of importance for T2D[5,6]. Indeed, earlier studies discovered alterations in DNA methylation in pancreatic islets, liver, adipose tissue and muscle from people with T2D versus controls[7–19]. However, prior studies analysed methylomes in whole tissues, containing different cell types, although it is established that epigenetics

controls cell-type-specific gene expression. Hence, studies focusing on DNA methylation in individual cell types in tissues from people with T2D are needed. To our knowledge, a few studies have analysed cell-specific methylomes in humans[20], but none were done in tissues from people with T2D. A better understanding of epigenetic and molecular defects in individual cell types in islets from people with or at risk of T2D could identify previously unrecognized aetiopathogenetic mechanisms and drug targets. Additionally, studies exploring whether altering DNA methylation in regulatory regions of alpha and beta cell-specific genes causes expression changes and alters hormone secretion are desirable.

Single-cell RNA sequencing (scRNA-seq) and single-cell assay for transposase-accessible chromatin using sequencing (scATAC-seq) can identify transcriptome and open chromatin differences in individual cell types of a tissue[21]. However, most single-cell studies are based on smaller numbers of cells from each tissue, and individual, thus representing only minute parts of a tissue. scRNA-seq studies identified differentially expressed genes (DEGs) in islets from donors with T2D versus controls[21,22]. The DEGs identified in these studies are largely non-overlapping, potentially because many scRNA-seq protocols capture only 5–40% of transcripts in a cell, have low power due to analysis of smaller cohorts, and/or are affected by selection biases for certain cell types[22]. Methods capturing more transcripts from each cell, and more individual cells from each donor, may resolve these issues.

To fill these knowledge gaps, we used Method for Analyzing RNA following Intracellular Sorting (MARIS)[23] to sort larger numbers (~300,000 per donor) of alpha and beta cells from human pancreatic islets of donors with or without T2D, followed by whole-genome bisulfite sequencing (WGBS), the most comprehensive method to study methylation genome wide, and RNA-seq. Firstly, we aimed to characterize the global methylome and transcriptome of human alpha and beta cells and use epigenetic editing to investigate causative impacts of DNA methylation on transcription of alpha and beta cell-specific genes. Secondly, we aimed to identify previously unrecognized candidate genes contributing to impaired insulin or glucagon secretion by studying the impact of pre-T2D/T2D on methylomes and transcriptomes in human alpha and beta cells, followed by functional validation by manipulation of identified T2D-associated candidates in human islets and cell lines together with analyses of diabetic Goto-Kakizaki (GK) rats. The methylomes in alpha and beta cells were also integrated with transcription

factor (TF) binding[24] and T2D genetics[2]. Finally, we developed the web tool alpha-beta-methylome (https://alpha-beta-methylome.serve.scilifelab.se/app/alpha-beta-methylome/), a comprehensive resource for investigating the impact of age, sex and T2D on DNA methylation and gene expression in human alpha and beta cells.

## Results

### WGBS and RNA-seq of human alpha and beta cells
Using MARIS for cell sorting of human islets[23], we obtained on average 322,627 ± 73,055 alpha cells and 311,550 ± 61,250 beta cells from 24 donors with or without T2D (Fig. 1a and Extended Data Fig. 1a). Characteristics of all donors in this study (17 non-diabetic controls and 7 pre-T2D/T2D) and donors for each analysis are presented in Supplementary Table 1. WGBS analysed genome-wide DNA methylation at base-pair resolution on alpha and beta cell samples where we had enough DNA (Fig. 1b). Supplementary Table 2 reports sequencing information for WGBS data passing quality control ($n = 10$, alpha cells; $n = 14$, beta cells). After deduplication, ~84% of reads mapped uniquely to GRCh38. After quality control, the transcriptome was analysed by RNA-seq in alpha and beta cells from 22 donors (Fig. 1b and Supplementary Table 1b).

### Global methylome of human alpha and beta cells
We next characterized the global methylome in alpha and beta cells, from non-diabetic donors and with WGBS data available from both cell types (Supplementary Tables 1 and 2). DNA methylation of 27.54 million cytosine-phosphate-guanine (CpG) sites was profiled and the distribution of methylation levels through different genomic regions visualized with dmrseq[25]. Globally, methylation is bimodal, with most CpGs hypermethylated (right peaks) or unmethylated (left peaks), resulting in average methylation levels of 76.1% and 77.2% in alpha and beta cells, respectively (Fig. 1c). We also analysed the genome-wide methylation levels in relation to different genomic regions and CG density (Extended Data Fig. 1b and Fig. 1d,e). Introns, 3′ untranslated regions (UTRs) and intergenic regions exhibited the highest methylation levels, whereas promoters and 5′ UTRs had the lowest degree of methylation (Fig. 1d). Based on CG density, CpG islands had the lowest methylation level, while Open Sea and Shelves had the highest (Fig. 1e).

We then used RNA-seq data from alpha and beta cells ($n = 16$; Supplementary Table 1b) to explore whether methylation levels in

---

**Fig. 1 | Analysis of global DNA methylome in human alpha and beta cells.**
**a**, Workflow for the current study. Human pancreatic islets from 24 donors with or without T2D were used for sorting of alpha and beta cells after staining for glucagon and insulin. WGBS and RNA-seq were used to analyse the methylome and transcriptome in sorted alpha and beta cells. Bioinformatic analyses were used to identify differential DNA methylation and gene expression. Follow-up experiments were done in human islets, EndoC-βH1 cells, INS-1 beta cells and GK rats, by epigenetic editing, analysing gene expression, insulin secretion and metabolism. **b**, Filtering islet samples for WGBS and RNA-seq. The Lund University Diabetes Centre (LUDC) islet sorting cohort consists of alpha and beta cells sorted from 24 human islet donors. Based on samples with enough DNA, WGBS was performed on sorted alpha and beta cells from 11 and 14 islet preparations, respectively. Among WGBS samples passing quality control (one alpha cell sample did not), 3 donors had pre-T2D/T2D (both alpha and beta cell), and 7 (alpha cell) and 11 (beta cell) were controls. RNA-seq was performed on sorted alpha and beta cells from 22 and 24 islet preparations, respectively. Among RNA-seq samples passing quality control (two beta cell samples did not), 7 donors had pre-T2D/T2D and 15 were controls (both alpha cells and beta cells). **c**, Density plot showing the global methylome of human alpha (mean 76.1%) and beta (mean 77.2%; $n = 7$ donors) cells. Peaks near 0% and 100% methylation. **d,e**, Density plots showing the degree of DNA methylation in alpha and beta cells in different genomic (**d**) and CpG island (**e**) regions. **d**, The region 1 to 5 kilobase pairs (kb) upstream of the transcription start site (TSS) had mean methylation levels of 66.7% (alpha cells) and 67.0% (beta cells). Promoters (1,000 bp upstream of TSS) had mean methylation levels of 25.5% (alpha cells) and 25.4% (beta cells).

5′ UTRs had mean methylation levels of 13.5% (alpha cells) and 13.1% (beta cells). Exons had mean methylation levels of 62.4% (alpha cells) and 62.2% (beta cells). Introns had mean methylation levels of 79.8% (alpha cells) and 79.9% (beta cells). 3′ UTRs had mean methylation levels of 80.6% (alpha cells) and 80.0% (beta cells). Intergenic regions had mean methylation levels of 78.1% (alpha cells) and 80.2% (beta cells). **e**, The mean degree of DNA methylation of Shelves (81.9% in alpha and 82.6% in beta cells), Shores (62.2% in alpha and 62.4% in beta cells), CpG islands (15.8% in alpha and in beta cells) and Open sea (82.4% in alpha and 83.7% in beta cells). A CpG island is defined as a ≥200-bp-long stretch of DNA with a CG content of ≥50% and an observed CpG/expected CpG in excess of 0.6. The Shores are the 2-kb flanking regions of CpG islands, the Shelves are the 2-kb regions outside island shores, and the Open Sea is everything else. **f,g**, Average DNA methylation levels, using the WGBS data, in different gene regions of genes divided according to expression level (no, low, medium and high expression) in alpha (**f**; $n = 7$ donors) and beta cells (**g**; $n = 11$ donors), respectively. The WGBS data were analysed by dmrseq[25], and the Friedman rank-sum test was used to test differences in the average DNA methylation levels between the groups. Data are presented as the mean ± s.e.m.; for **f**, ***$P_{1\text{-}5kb} = 0.000105$, $P_{promoters} = 0.000105$, $P_{5'UTR} = 0.000105$, $P_{exons} = 0.000105$, $P_{introns} = 0.000172$, $P_{3'UTR} = 0.000105$; for **g**, ***$P_{1\text{-}5kb} = 0.000000835$, $P_{promoters} = 0.000000322$, $P_{5'UTR} = 0.000000322$, $P_{exons} = 0.000000835$, $P_{introns} = 0.000000322$, $P_{3'UTR} = 0.000000322$, as analysed by two-sided Friedman rank-sum tests, not corrected for multiple testing. The vertical lines in **c**–**e** represent the mean DNA methylation. Schematic in **a** created in BioRender; Ofori, J. https://biorender.com/h6zumch (2026).

different genomic regions associate with the expression level of genes in alpha and beta cells. Genes with <2 mean normalized counts were categorized as not expressed, and the remaining 24,730 and 24,726 genes in alpha and beta cells, respectively, were divided into three groups of similar size, categorized into low (0–33% quantile), medium (33–66% quantile) and high (66–100% quantile) expression. We found associations between the mean DNA methylation levels in different genomic regions and expression levels (Fig. 1f,g; Friedman rank-sum test). In promoters and 5′ UTR regions, lower methylation was associated with higher gene expression; while in intragenic regions, for example, exons, introns and 3′ UTR, lower methylation was associated with lower expression (Fig. 1f,g). Additionally, we found small significant differences between alpha and beta cells for the mean degree of methylation for different genomic regions linked to the four categories of expressed genes (Extended Data Fig. 1c–h).

## DMRs in human alpha versus beta cells

To further dissect the epigenetic differences between alpha and beta cells, we used dmrseq[25] and WGBS data from seven donors not diagnosed with T2D and with the methylome available in both cell types (Supplementary Tables 1 and 2 and Extended Data Fig. 1i). On autosomal chromosomes, WGBS covered 25,907,683 CpGs. We found 22,544 differentially methylated regions (DMRs) annotated to 7,975 autosomal genes in alpha versus beta cells, when requiring three consecutive differentially methylated CpGs and false discovery rate (FDR) ≤ 5% ($q < 0.05$; Fig. 2a,b, Extended Data Fig. 1j and Supplementary Table 3a–c). These genes are enriched in Kyoto Encyclopedia of Genes and Genomes (KEGG) pathways including insulin secretion, cAMP signalling, glucagon signalling, calcium signalling, metabolic pathways and T2D mellitus (Fig. 2c and Supplementary Table 3d). The DMRs covered 3–379 CpGs (average 10 CpGs) and 9–10,752 base pairs (bp; mean length 637 bp).

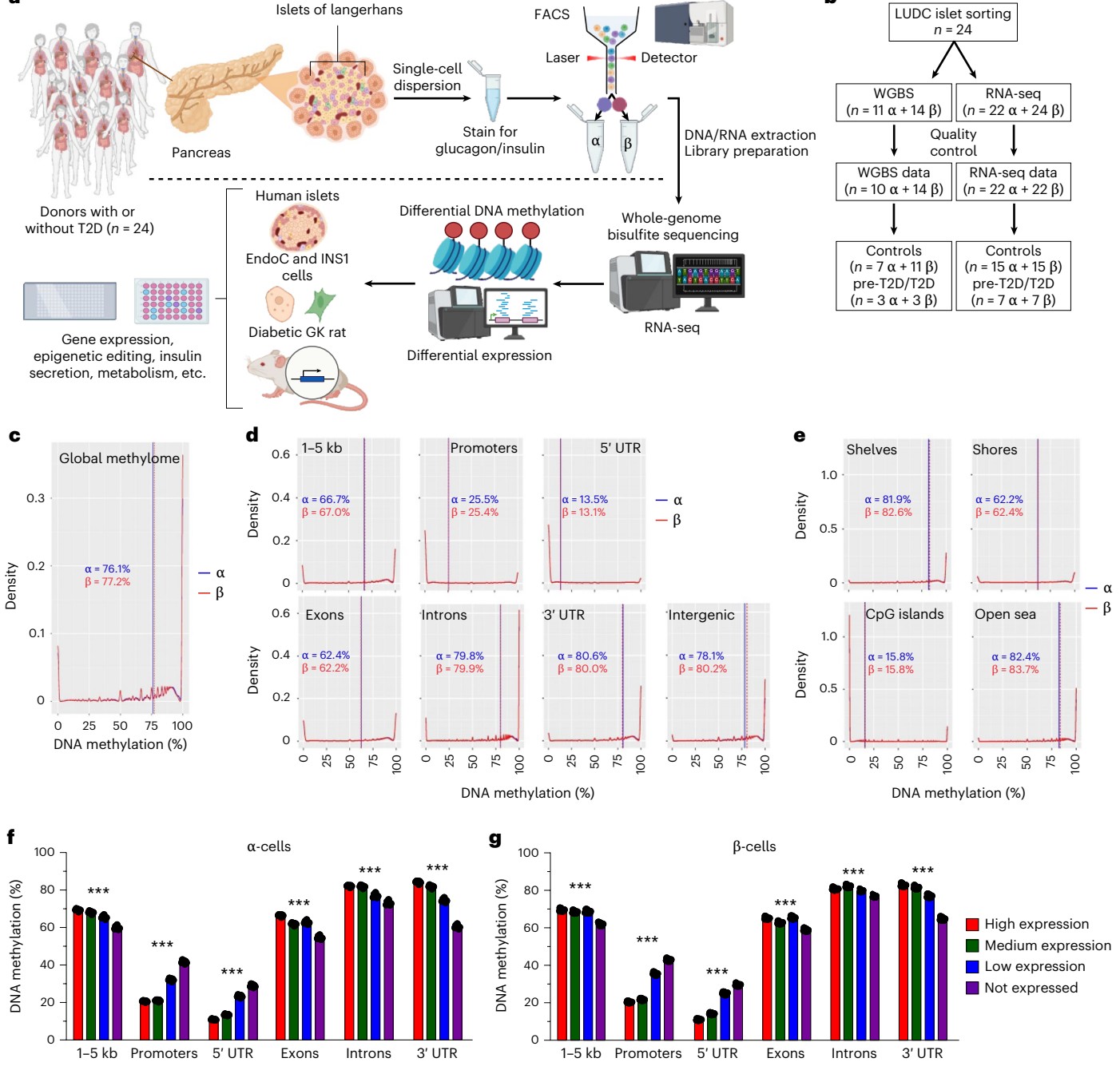

Extended Data Fig. 1k shows the annotation of the DMRs in relation to different genomic regions and CG density (Supplementary Table 3b,c). Interestingly, DMRs annotated to key alpha and beta cell genes[21,24,26,27], for example, *PDX1*, *PLUT*, *INS*, *INS-IGF2*, *PCSK1*, *GCG*, *SLC2A2*, *MAFA*, *GATA6*, *SIX2* and *GPR119*, were among the most significant (Fig. 2d–i, Extended Data Fig. 1l–n and Supplementary Table 3b). Several genes have multiple DMRs, for example *POU6F2* ($n = 16$), *PLUT* ($n = 11$), *PCSK1* ($n = 10$), *INS/INS-IGF2* ($n = 7$) and *PDX1* ($n = 3$; Fig. 2d,e, Extended Data Figs. 2 and 3 and Supplementary Table 3b). The longest DMRs ($\geq6,104$ bp) were annotated to, for example, *GCG*, *PLUT*, *PDX*, *HADH*, *RXRG*, *PAX3*, *MAFA*, *ZC3H3* and *KCNQ5*, while DMRs annotated to *PLUT*, *PDX1* and *MAFA/ZC3H3* included most CpGs ($\geq212$; Supplementary Table 3b, Fig. 2d,g and Extended Data Fig. 4a). We then compared the list of 7,975 genes with alpha versus beta cell DMRs identified in the current study with a list of genes presented in Supplementary Tables 9 and 10 in a similar study by Manduchi et al.[28]. This showed that 468 of 706 (66%) genes, including *GCG*, *INS*, *MAFA* and *PDX1*, presented in their study exhibit DMRs also identified in our study (hypergeometric enrichment score (HES) = 2.07, $P = 2.63 \times 10^{-58}$).

Epigenetics may regulate expression by controlling TF binding to the genome[29]. Therefore, we investigated TF motifs and binding in identified DMRs. Using HOMER, we found enrichment of TF motifs, including NEUROD1, FOXA2, PDX1, FOXA3, FOXP1 and MAFB, in these DMRs (Fig. 2j and Supplementary Table 3e). Additionally, we present DMRs that overlap with sequences bound by the islet-specific TFs, FOXA2, MAFB, NKX2.2, NKX6.1 and PDX1 in Fig. 2k, and Supplementary Table 3f (enriched overlap, $P \leq 0.001$)[24]. Moreover, 270 DMRs, annotated to for example, *CREB5*, *GLP1R*, *HDAC9*, *OPRD1* and *SYT13*, were bound by all five TFs.

Next, since genetics affects T2D, especially via islet dysfunction[2], and because we previously demonstrated interactions between genetics and epigenetics[30,31], we studied the overlap between genes linked to T2D-asociated SNPs[2] and the 7,975 genes with DMRs presented in Supplementary Table 3. Interestingly, 381 of the 849 (45%) T2D candidate genes also have DMRs in alpha versus beta cells (Fig. 2l and Supplementary Table 3g, HES = 1.74, $P = 2.63 \times 10^{-34}$). Additionally, 25 T2D-associated SNPs are located within DMRs annotated to for example, *INS-IGF2*, *CDKN2B*, *PCSK1*, *GLP1R*, *MEG3* and *KCNQ1* (Fig. 2m and Supplementary Table 3g). Notably, we found that 446 of 1,289 T2D SNPs[2] are in CpGs (Supplementary Table 3g). Together, this supports that T2D-risk alleles can affect DNA methylation, and potentially alter T2D risk, via effects on methylation[30]. Additionally, 831 of the DMRs, for example, annotated to *GCG*, *SLC2A2* and *AKT1*, overlap with 1,086 CpGs with differential islet methylation in T2D (Supplementary Table 3a)[12]. Finally, we intersected our alpha versus

beta cell DMRs with three-dimensional chromatin map data[32]. Here, 30 of 89 chromatin interaction points that differ between alpha and beta cells overlap with an alpha versus beta cell DMR, with 15 overlapping with more than one DMR (Supplementary Table 3h, overlap $z$-score = 8.3, $P \leq 0.001$).

Overall, these data clearly show cell-specific differences in the islet epigenome, overlapping key TF binding sites[24], T2D-associated SNPs[2] and T2D-associated islet DNA methylation[12].

Epigenetic differences on sex chromosomes in alpha versus beta cells were analysed separately in males and females. In males, we analysed 1,298,421 CpGs on X and Y chromosomes. We identified 453 DMRs, annotated to 178 genes, on the X chromosome, but none on the Y chromosome (Supplementary Table 4). Some of the most significant DMRs on the male X chromosome are annotated to *ARX*, *USP27X*, *PRKX*, *GPR119*, *POLA1*, *MPP1*, *DLG3*, *HDAC8* and *KDM6A* (Fig. 2n, Extended Data Fig. 4b,c and Supplementary Table 4b). The longest X-chromosome DMRs ($\geq2,685$ bp) are annotated to *NEXMIF*, *GPR119*, *HS6ST2* and *DACH2*, while DMRs annotated to *BCOR*, *HCFC1* and *GPR119* have most CpGs (Supplementary Table 4b). Additionally, we present X-chromosome DMRs that overlap with sequences bound by FOXA2, MAFB, NKX2.2, NKX6.1 and/or PDX1 in Extended Data Fig. 4d and Supplementary Table 4d (enriched overlap, $P \leq 0.001$). As there were only two non-diabetic females with WGBS data available in alpha and beta cells, we could not identify X-chromosome DMRs in females.

## DEGs in human alpha versus beta cells

Next, we studied gene expression in alpha and beta cells from 16 non-diabetic donors (Supplementary Table 1b). Using RNA-seq, we found 13,001 and 12,876 genes expressed in alpha and beta cells, respectively ($\geq20$ mean normalized counts in $\geq80\%$ of samples, unique Ensembl gene ID; Supplementary Table 5a,b). Notably, 93% and 94% of the genes found to be expressed in alpha and beta cells in our dataset overlap with previously published work[27], after reanalysing that data using the same conditions as ours (Supplementary Table 5a,b). While 12,384 genes were expressed in both cell types, 617 and 492 transcripts were uniquely expressed in either alpha or beta cells (Fig. 3a and Supplementary Table 5a,b). Based on gene ontology (http://www.webgestalt.org/), transcripts expressed in alpha cells, but not beta cells, showed enrichment in, for example, cAMP-mediated signalling, regulation of ion transport, regulation of secretion, exocytosis, regulation of apoptotic process and cell proliferation (Fig. 3b and Supplementary Table 5c). The 492 transcripts expressed only in beta cells, showed enrichment for 12 biological processes, for example, gland development (including *NKX3.1*), regulation of hormone levels (including *MAFA*, *PDX1*, *GLP1R* and *SLC2A2*), synaptic signalling (including

**Fig. 2 | Genes important for islet cell function and T2D exhibit widespread DNA methylation differences in sorted human alpha versus beta cells. a**, Heat map displaying significant DMRs between human alpha and beta cells ($n = 7$ donors) sorted from the same islet preparations (FDR < 5%, $q < 0.05$). **b**, Volcano plot showing hypomethylated and hypermethylated DMRs identified by dmrseq[25] in human beta compared to alpha cells ($n = 7$ donors). $q < 0.05$ indicates the cut-off (dashed line) and the 'methylation difference' on the $x$ axis represents the beta coefficient generated by dmrseq and included in Supplementary Table 3a. **c**, KEGG pathways with FDR below 5% ($q < 0.05$) based on a WebGestalt analysis of 7,975 genes with DMRs between sorted alpha and beta cells. **d–i**, Left, Top DMRs analysed and visualized by dmrseq[25] in alpha versus beta cells are annotated to *PDX1* (**d**), *INS/INS-IGF2* (**e**), *PCSK1* (**f**), *GCG* (**g**), *SLC2A2* (**h**) and *MAFA* (**i**; $n = 7$ donors). Right, Expression of respective gene in sorted alpha versus beta cells ($n = 16$ donors, ***$q < 0.001$ based on DESeq2 (ref. 82). Data are presented as the mean ± s.e.m. Women and men are represented by filled and open circles, respectively. **j**, The significant DMRs between alpha and beta cells ($q < 0.05$, based on dmrseq[25]) are enriched for putative binding motifs of TFs such as NEUROD1, FOXA2, FOXA3, FOXP1 and MAFB ($q < 0.05$, as analysed by HOMER, http://homer.ucsd.edu/homer/motif/). **k**, Venn diagram and a table showing the significant DMRs between alpha and beta cells ($q < 0.05$) overlapping with binding sites

for five islet-specific TFs: FOXA2, MAFB, NKX2.2, NKX6.1 and PDX1 (ref. 24). Overlaps are enriched with $P \leq 0.001$ based on region-based permutation tests. For individual $P$ values, see Supplementary Table 3f. **l**, 381 of 849 genes (45%) with T2D-associated SNPs[2] overlapped with genes having DMRs in alpha versus beta cells (HES = 1.74, $P = 2.63 \times 10^{-34}$, based on one-tailed hypergeometric test). **m**, A significant DMR in alpha versus beta cells annotated to *INS-1GF2* directly overlaps the T2D-associated SNP (rs1124699, dashed line). **n**, The top X-chromosome DMR (only in males, five donors) annotated to *ARX* ($q < 0.0082$, based on dmrseq[25]) and with expression in sorted alpha versus beta cells ($n = 10$ male donors, ***$q < 0.001$, based on DESeq2 (ref. 82). Data are presented as the mean ± s.e.m. Women and men are represented by filled and open circles, respectively. Supplementary Table 6 presents $P$ values for DEGs in **d–i**. In all DMR plots, locations of the methylation loci are shown with black lines at the bottom, and average, smoothed lines based on the dmrseq analysis are shown in pink (beta cells) or blue (alpha cells), including the standard deviation. The shaded area shows the DMR. Annotations to CpG island regions and genes are shown below each graph. Women and men are represented by filled and open circles, respectively. $P$ and $q$ values for **a–i**, **m** and **n** were corrected for multiple testing. CPM, counts per million.

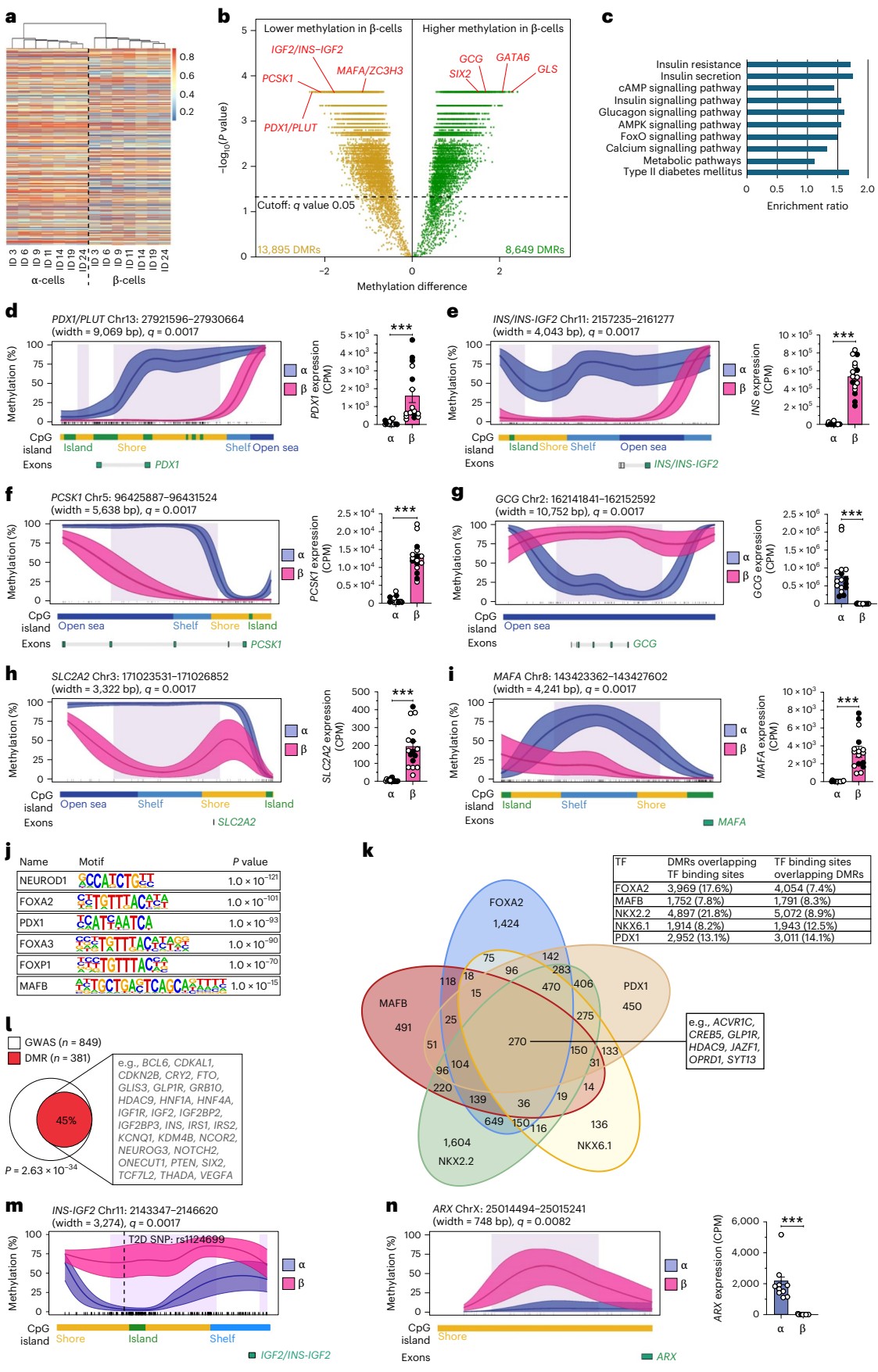

*NPY* and *SYT1*), cell–cell signalling (including *GAD1* and *GLP1R*) and ion transport (including *GLRA1*; Fig. 3c and Supplementary Table 5d). In both alpha and beta cells, ~40% of the 20 most abundantly expressed genes are mitochondrially encoded (Fig. 3d,e).

Additionally, 8,434 protein-coding genes were DEGs in alpha versus beta cells ($q$ < 0.05; Fig. 3f and Supplementary Table 5e). Notably, ~40% (3,502) of these DEGs also have DMRs (Supplementary Table 5f), supporting epigenetic regulation of cell-specific expression (see for example, Fig. 2d–i, Extended Data Fig. 1l–n and Fig. 3g–o). Additionally, the DMRs correlated negatively with the DEGs (Supplementary Table 5g). We proceeded to focus on the biological relevance of the most significant DEGs in alpha versus beta cells ($q$ < 10 × 10$^{-8}$; Supplementary Table 6a). Of these 3,618 DEGs, ~50% had higher expression in each cell type, and they were enriched in KEGG pathways including insulin secretion, T2D mellitus and circadian entrainment, as well as glucagon, cAMP, AMPK, FoxO and calcium signalling (Fig. 3p and Supplementary Table 6b–d). DEGs showing higher expression in alpha cells ($q$ < 10 × 10$^{-8}$) were enriched for ten pathways, for example, cAMP signalling, while DEGs showing higher expression in beta cells ($q$ < 10 × 10$^{-8}$) were enriched for pathways related to, for example, insulin secretion and maturity-onset diabetes of the young (Supplementary Table 6e,f). Among DEGs with $q$ < 10 × 10$^{-8}$ in alpha versus beta cells, 1,835 (51%) also have DMRs annotated to them (Supplementary Tables 3 and 6).

Next, we intersected these DEGs with the 849 GWAS T2D candidate genes[2] and 186 (22%) genes, including *ADCY5*, *GLIS3*, *GLP1R*, *HNF4A* and *INS*, were overlapping (Fig. 3q and Supplementary Table 6g, HES = 1.67, $P$ = 1.07 × 10$^{31}$).

### Epigenetic editing reveals a causal role for DNA methylation in regulation of *INS*, *TH* and *GCG* expression

The cell-specific DNA methylation differences, which we found linked to large differences in expression of key alpha and beta cell genes, suggest that the methylome directly regulates expression and potentially hormone levels in individual islet cell types (Fig. 2d–i and Supplementary Table 3b). To test this, in human EndoC-βH1 beta cells, we used epigenetic editing with CRISPR–dCas9–DNMT3A or CRISPR–dCas9–TET1 (ref. 33) together with guide RNAs (gRNAs) targeting three alpha versus beta cell DMRs annotated to *INS*, *TH* and *GCG* (Figs. 2e,g and 4a–f).

CRISPR–dCas9–DNMT3A-mediated editing, with the *INS*-DMR-gRNA targeting the *INS* DMR (Figs. 2e and 4a,b) and the *TH*-DMR-gRNA targeting a region annotated to *TH* (Fig. 4c,e), resulted in increased methylation of 22 and 13 CpG sites, respectively, as analysed by pyrosequencing and EPIC v2.0 methylation arrays (Fig. 4g–j). Notably, after editing, 5 CpGs near the *INS*-DMR-gRNA target location showed absolute increased methylation of 20–43%, representing 2.46- to 30-fold increases, and these sites may have a stronger impact on gene expression than the other sites (Fig. 4h). DNMT3A-mediated editing using *INS*-DMR-gRNA and *TH*-DMR-gRNA caused reduced *INS* and *TH* expression, respectively (Fig. 4k,l). The targeted region in *TH* is located very near a potential *INS* enhancer (Fig. 4a,c) but editing using the *TH*-DMR-gRNA did not alter *INS* expression (Fig. 4k). Importantly,

DNMT3A-mediated editing using the *INS*-DMR-gRNA led to reduced insulin content in human EndoC-βH1 beta cells (Fig. 4m).

Next, we used CRISPR–dCas9–TET1 and a gRNA (*GCG*-DMR-gRNA) targeting a DMR located in the promoter of *GCG*, encoding glucagon (Figs. 2g and 4d). This led to decreased DNA methylation at two CpG sites analysed by pyrosequencing and EPIC v2.0 (Fig. 4n,o). Additionally, the ten-eleven translocation methylcytosine dioxygenase 1 (TET1)-based editing using the *GCG*-DMR-gRNA resulted in increased *GCG* expression and glucagon content in the EndoC-βH1 beta cells (Fig. 4p,q).

Importantly, to our knowledge, this is the first study where epigenetic editing reveals a causal role for DNA methylation on *INS*, *TH* and *GCG* expression in beta cells, supporting future development of epigenetic-based therapies for regulation of, for example, insulin.

### Pre-T2D/T2D-associated DMRs in human alpha and beta cells

To explore the epigenetic component of T2D, we compared the methylome in alpha and beta cells from donors with pre-T2D/T2D, based on having a haemoglobin A1c (HbA1c) ≥ 6% (≥42 mmol mol$^{-1}$) and/or a T2D diagnosis, with cells from controls, including non-diabetic donors with HbA1c < 6% (Supplementary Table 1a,b). In alpha cells, we found 3,207 autosomal DMRs associated with pre-T2D/T2D (hereafter called T2D-associated; permutation $P$ < 0.05, Fig. 5a and Supplementary Table 7a–c). These DMRs were annotated to 1,871 genes, including *GCG*, *INS-IGF2* and *PDX1*, enriched in KEGG pathways, including circadian entrainment (Fig. 5b and Supplementary Table 7d). The DMRs covered 3–257 CpGs and 10–18,509 bp. For protein-coding genes, T2D-associated alpha cell DMRs with lowest $P$ values were annotated to *ZNF566*, *UNC93A*, *CROCCP2*, *TBCA*, *USP18* or *SNRPN*, and the longest DMRs (>5,025 bp) were annotated to *TUBBP5* and *FLG*, with the one annotated to *TUBBP5* also covering most CpGs (Fig. 5a and Supplementary Table 7b). Figure 5c–h presents some T2D-associated alpha cell DMRs annotated to genes previously linked to T2D and/or islet function, for example, *SNRPN*[34], *SLC52A2/TMEM249* (ref. 35), *CHURC1* (ref. 36) and *TULP4* (ref. 37) or encoding epigenetic enzymes, for example *KDM1B* and *KDM3B*. Using HOMER, we found enrichment of TF motifs, including HNF1B and OCT4, in the T2D-associated alpha cell DMRs (Fig. 5i and Supplementary Table 7e), and we identified DMRs that overlap with sequences bound by the islet-specific TFs FOXA2, MAFB, NKX2.2, NKX6.1 and PDX1 (Fig. 5j and Supplementary Table 7f; enriched overlap, $P$ ≤ 0.004). Additionally, 103 of 849 (12%) T2D candidate genes discovered by GWAS[2] also have T2D-associated alpha cell DMRs (Fig. 5k and Supplementary Table 7g; HES = 7.34, $P$ = 1.37 × 10$^{-238}$). This represents 5.5% of the 1,871 genes annotated to T2D-associated alpha cell DMRs. The overlapping genes include *GIP*, *GLIS3*, *GRB10*, *HDAC9*, *IGF2*, *IGF2BP3*, *KCNQ1*, *KDM4B*, *MEG3*, *TH* and *TCF7L2*. The T2D-associated SNP rs883541 (ref. 2) lies within a DMR annotated to *WIPI1* (Fig. 5l). Additionally, 28 T2D-associated alpha cell DMRs overlap with 32 CpGs previously found to be differentially methylated in islets from donors with T2D versus controls[12] (Supplementary Table 7a). T2D-associated differences in methylation of individual CpGs of *GCG*, *INS*, *PDX1/PLUT* and *MAFB* in alpha cells are shown in Fig. 5m and Extended Data Fig. 4e–g.

---

**Fig. 3 | The transcriptome of sorted human alpha and beta cells. a**, Venn diagram showing the number of overlapping (12,384) and uniquely expressed protein-coding genes in human alpha and beta cells, with 617 and 492 genes, respectively, requiring ≥20 mean normalized counts in ≥80% of the samples. **b,c**, Gene ontology of the protein-coding genes uniquely expressed in human alpha (**b**) or beta (**c**) cells, showing significantly enriched pathways ($q$ < 0.05). **d,e**, The 20 most abundantly expressed genes in alpha (**d**) and beta (**e**) cells of 16 donors. Data are presented as the mean ± s.e.m. Women and men are represented by filled and open circles, respectively. **f**, Volcano plot showing DEGs at $q$ < 10 × 10$^{-8}$ and $q$ < 0.05, based on DESeq2 (ref. 82), in human alpha versus beta cells sorted from the same islet preparations ($n$ = 16 donors). **g–o**, Alpha versus beta cell DEGs based on DESeq2 (ref. 82; ***$q$ < 0.001, $n$ = 16 donors) that also have significant

DMRs ($q$ < 0.05, based on dmrseq[25]): *TM4SF4* (**g**), *F10* (**h**), *IRX2* (**i**), *FEV* (**j**), *LDHA* (**k**), *ADCY2* (**l**), *NR3C1* (**m**), *GLIS3* (**n**) and *SYT13* (**o**). Data are presented as the mean ± s.e.m. Women and men are represented by filled and open circles, respectively. **p**, Selected KEGG pathways with significant enrichment ($q$ < 0.05 based on WebGestalt analysis with correction for multiple testing) of DEGs in alpha versus beta cells with $q$ < 10 × 10$^{-8}$. **q**, 186 of 849 genes (22%) with T2D-associated SNPs identified by GWAS[2] overlapped with the top DEGs ($q$ < 10 × 10$^{-8}$, based on DESeq2)[82] in human alpha versus beta cells (HES = 1.67, $P$ = 1.07 × 10$^{-31}$, based on one-tailed hypergeometric test). Exact $P$ values for the expression data and DEGs are found in Supplementary Table 6. $P/q$ values for **b**, **c** and **f–p** were corrected for multiple testing.

In the beta cells, we discovered 5,106 T2D-associated autosomal DMRs annotated to 3,020 genes, for example, *INS* and *INS-IGF2* (permutation *P* < 0.05; Fig. 6a and Supplementary Table 8a–c). The 3,020 genes were enriched in KEGG pathways including cAMP, Wnt, calcium, PI3K-Akt, FoxO and AMPK signalling (Fig. 6b and Supplementary Table 8d). T2D was associated with lower methylation of 75.5% (3,857) of these DMRs. The DMRs covered 3–315 CpGs (average 10 CpGs) and 11–7,594 bp (mean length 662 bp). The longest DMRs (>3,587 bp) were annotated to, for example *TUBBP5*, *LINC01597*, *MYO16*, *SHISAL2B*, *FAM242A*, *LOC101927078*, *OR5D16* and *MIR4321*, while a DMR annotated to *MIR4321*, *AMH* and *JSRP1* has the most CpGs (315 loci; Fig. 6a and Supplementary Table 8b). Figure 6c–j presents some T2D-associated beta cell DMRs annotated to genes previously linked to T2D and/or islet function, including *EPS8L1* (ref. 38), *VEGFA*[39], *NKX6.3* (ref. 40), *CTNND2* (ref. 41), *RGS6* (ref. 42), *FEM1B*[43], *SEC16A*[44] and *SORBS1* (ref. 45). Using HOMER, we

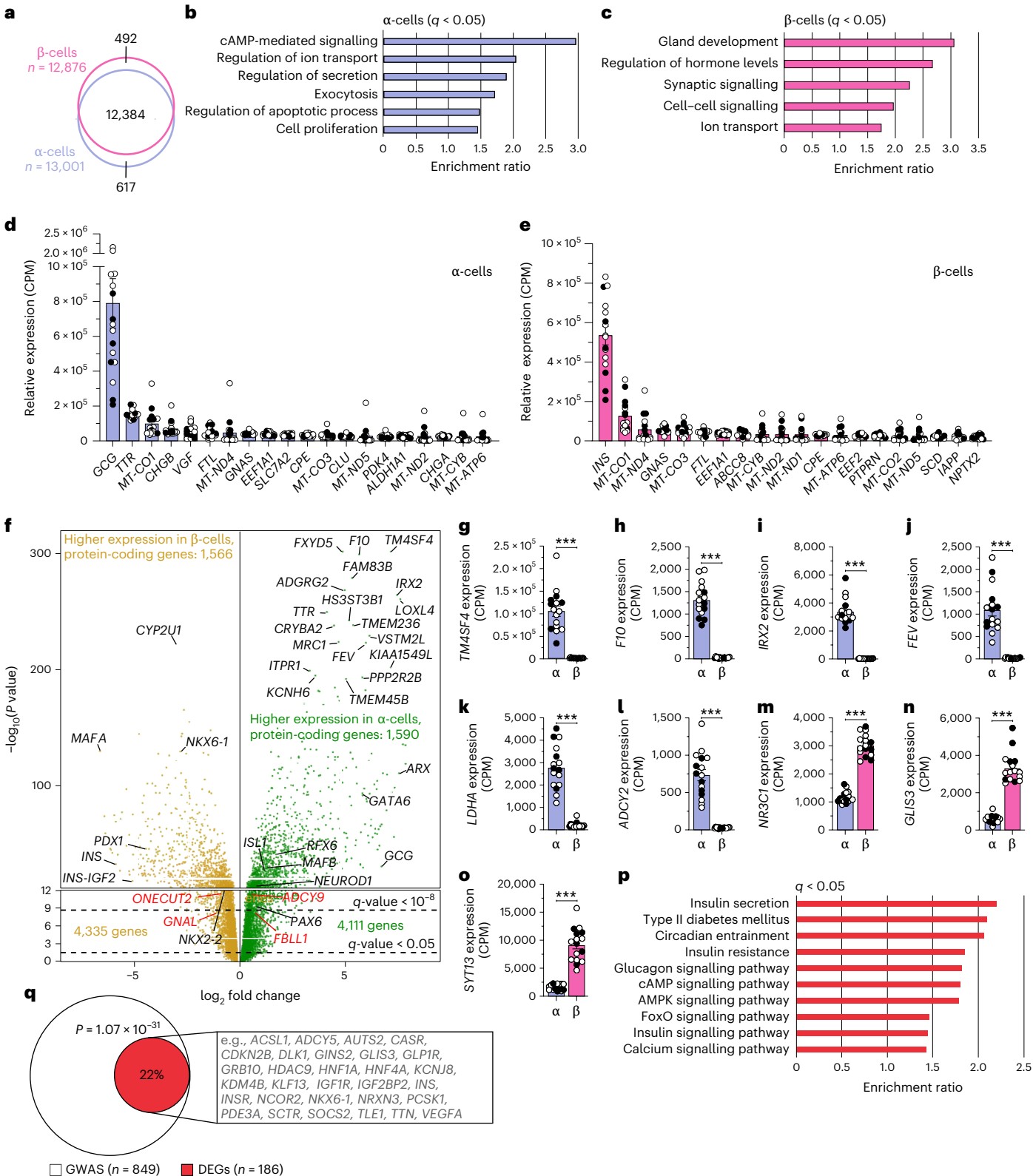

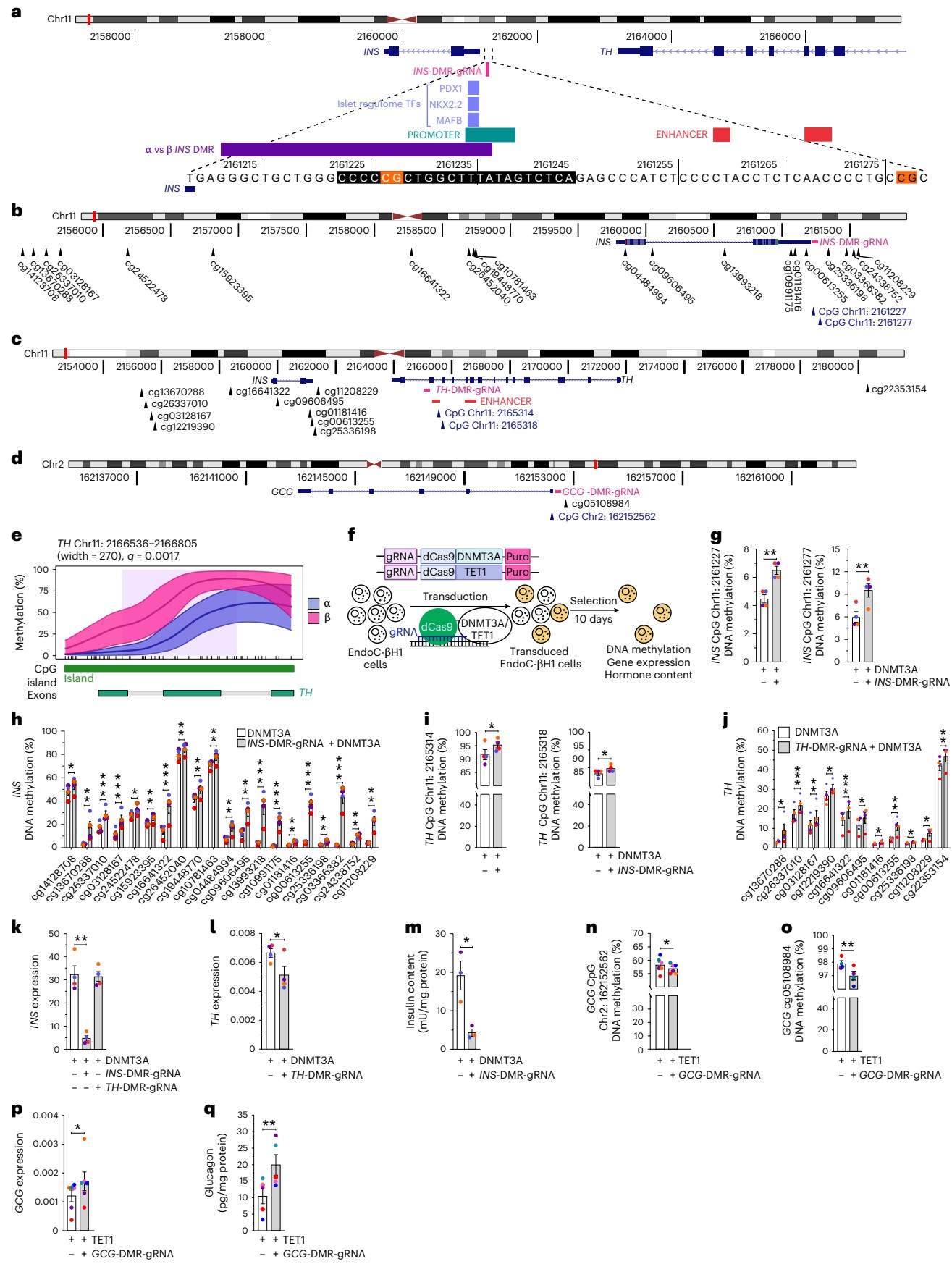

**Fig. 4 | Epigenetic editing shows a causative role for DNA methylation of *INS*, *TH* and *GCG* on gene expression in beta cells. a**, The location of a human alpha versus beta cell *INS* DMR (purple; see also Fig. 2e), gRNA binding site (pink), TF binding in islets (https://pasqualilab.upf.edu/app/isletregulome/; blue), promoter (green) and enhancers (red). The zoomed-in part (bottom) shows the sequence targeted by the gRNA (black) and the two CpG sites covered by the pyrosequencing assay (orange). The figure was made with the help of the UCSC Genome Browser. *INS* runs from right to left in this image. **b–d**, The gRNA binding sites (pink), enhancer (**c**; red) and CpG sites where DNA methylation changed after epigenetic editing ($P < 0.05$) within the targeted loci for *INS* (**b**), *TH* (**c**) and *GCG* (**d**) analysed by pyrosequencing (displayed in blue), or EPIC v2.0 array (displayed in black). **e**, The *TH* DMR with differential DNA methylation in human alpha versus beta cells, in the region targeted by epigenetic editing. *q* value was corrected for multiple testing. Locations of the methylation loci are shown with black lines at the bottom, and average, smoothed lines based on the dmrseq[25] analysis are shown in pink (beta cells) or blue (alpha cells), including the standard deviation. The shaded area shows the DMR. Annotations to CpG island regions and genes are shown below the graph. **f**, Design of the epigenetic editing experiments performed to investigate the causal relationship between altered DNA methylation and changes in gene expression/cell function. **g**, Pyrosequencing shows that epigenetic editing with CRISPR–dCas9–DNMT3A and the gRNA targeting the *INS* DMR (*INS*-DMR-gRNA) increased DNA methylation of two analysed sites (CpG Chr11: 2161227 and CpG Chr11: 2161277, blue; **b**), compared with the negative control including DNMT3A but not the *INS*-DMR-gRNA in EndoC-βH1 beta cells. $n = 4$ biological replicates, \**$P < 0.01$, based on one-tailed paired *t*-tests (left, $P = 0.008$; right, $P = 0.006$). **h**, The EPIC v2.0 array identified edited CpG sites between 10 kb upstream and downstream of *INS*, discovering absolute methylation changes of up to 40% between beta cells that had received both CRISPR–dCas9–DNMT3A and the *INS*-DMR-gRNA and the negative control (only DNMT3A). $n = 4$ biological replicates, \**$P < 0.05$, \*\***$P < 0.01$ and \*\*\***$P < 0.001$ based on one-tailed paired *t*-tests. **i**, Pyrosequencing data of two CpG sites with altered methylation after epigenetic editing with CRISPR–dCas9–DNMT3A together with *TH*-DMR-gRNA versus control (only DNMT3A). $n = 4$ biological replicates, \**$P ≤ 0.05$ based on one-tailed paired *t*-tests (left, $P = 0.05$; right, $P = 0.04$). **j**, presents CpG sites identified by EPIC v2.0 array, where DNA methylation was altered after epigenetic editing with CRISPR–dCas9–DNMT3A and the *TH*-DMR-gRNA versus control (only DNMT3A). $n = 4$ biological replicates, \**$P < 0.05$, \*\***$P < 0.01$ and \*\*\***$P < 0.001$ based on one-tailed paired *t*-tests. **k–m**, CRISPR–dCas9–DNMT3A-based editing of *INS* or *TH* reduced *INS* ($P = 0.001$) and *TH* ($P = 0.023$) expression, respectively (quantitative PCR (qPCR), $n = 4$ biological replicates, **k** and **l**), and insulin content ($P = 0.021$, only after editing with *INS*-DMR-gRNA, $n = 3$ biological replicates, **m**) in EndoC-βH1 beta cells. \**$P < 0.05$, \*\***$P < 0.01$, based on one-tailed paired *t*-tests. **n,o**, Pyrosequencing and EPIC v2.0 array data show that epigenetic editing with CRISPR–dCas9–TET1 and the gRNA, *GCG*-DMR-gRNA, targeting the *GCG* DMR in Fig. 2g decreased DNA methylation of two CpG sites (CpG Chr2: 162152662 for pyrosequencing and cg05108984 for EPIC v2.0) versus the control (only TET1) in EndoC-βH1 beta cells. $n = 6$ biological replicates for pyrosequencing, \**$P = 0.05$ based on one-tailed paired *t*-test. $n = 4$ biological replicates for EPIC v2.0 array, \*\***$P = 0.00741$, based on one-tailed paired *t*-test. **p,q**, CRISPR–dCas9–TET1-based editing with *GCG*-DMR-gRNA caused increased *GCG* expression (\**$P = 0.047$, qPCR, $n = 6$ biological replicates, **p**), and glucagon content (\*\***$P = 0.008$, ELISA, $n = 5$ biological replicates, **q**) versus the control (only TET1) in EndoC-βH1 beta cells, based on one-tailed paired *t*-tests. The colours of the dots in **g–m** and **n–q** represent samples from the same experiments. Data are presented as the mean ± s.e.m. Bars shown in light grey include CRISPR–dCas9–DNMT3A or CRISPR–dCas9–TET1 plus gRNA, while white bars are controls only including CRISPR–dCas9–DNMT3A or CRISPR–dCas9–TET1, and no gRNA.

found enrichment of TF motifs including FOXA3, FOXA2, HNF1B, OCT4, FOXA1, PAX7, FOXP1, BACH2 and PDX1 in T2D-associated beta cell DMRs (Fig. 6k and Supplementary Table 8e). Additionally, we present T2D-associated beta cell DMRs that overlap with sequences bound by the islet-specific TFs FOXA2, MAFB, NKX2.2, NKX6.1 and PDX1 (Fig. 6l and Supplementary Table 8f; enriched overlap, $P ≤ 0.001$). Further, 153 of 849 (18%) T2D candidate genes identified by GWAS[2] also have T2D-associated beta cell DMRs, including *CDKN2A*, *GLIS3*, *INS*, *KCNQ1*, *NKX6-3* and *LPL* (Fig. 6m and Supplementary Table 8f; HES = 2.69, $P = 3.13 × 10^{-85}$). This represents 5.1% of the 3,020 genes annotated to T2D-associated beta cell DMRs. Three T2D SNPs[2] (rs362307/*HTT*, rs150015181/*NKX6-3* and rs34245505/*BMF*) are located within T2D-associated beta cell DMRs (Fig. 6n and Supplementary Table 8g). Notably, 65 T2D-associated beta cell DMRs overlap 91 CpGs exhibiting differential islet methylation in T2D[12] (Supplementary Table 8a). These include DMRs annotated to *FER*, *ASCL5*, *HDAC4*, *CIT*, *PLX2* and *FGF8*. Additionally, Fig. 6o–q presents T2D-associated methylation differences of individual CpGs of *GCG*, *INS* and *PDX1/PLUT* in beta cells.

## Pre-T2D/T2D-associated DEGs in human alpha and beta cells

We next investigated the impact of pre-T2D/T2D on gene expression in alpha and beta cells. By comparing RNA-seq data of alpha cells from 7 pre-T2D/T2D donors with 15 controls (Supplementary Table 1a,b), we identified 345 protein-coding DEGs with $P < 0.05$ (Fig. 7a and Supplementary Table 9a). When integrating these data with our published RNA-seq data from whole islets of a large T2D case–control cohort[26], and a sorted alpha cell T2D case–control cohort[46], we found 11 and 3 overlapping DEGs, respectively, including *GNAL* (Extended Data Fig. 5a and Supplementary Table 9b,c). Additionally, 26 T2D-associated DEGs also have DMRs in alpha cells, for example, *BACH1* (Supplementary Table 9a).

To identify regulators of insulin secretion that may contribute to T2D, we compared RNA-seq data of beta cells from 7 pre-T2D/T2D donors with 15 controls (Supplementary Table 1a,b). This identified 573 protein-coding DEGs with $P < 0.05$, including *ADCY9*, *FBLL1*, *ZNF431*, *ZSWIM9*, *GNAL*, *ONECUT2*, *SNX1* and *TRAPPC13* with FDR < 10% (Fig. 7b,c and Supplementary Table 10a). Notably, 43 of the DEGs found in beta cells overlap with T2D-associated DEGs found in whole

**Fig. 5 | Key islet and T2D candidate genes exhibit pre-T2D/T2D-associated DMRs in human alpha cells. a**, Volcano plot showing hypomethylated and hypermethylated DMRs in pre-T2D/T2D cases ($n = 3$) versus non-diabetic controls ($n = 7$) in alpha cells ($P < 0.05$ based on dmrseq[25] and 10 permutations). The 'methylation difference' on the *x* axis represents the beta coefficient values generated by dmrseq and included in Supplementary Table 7a. **b**, Selected enriched KEGG pathways among the 1,871 genes with pre-T2D/T2D-associated DMRs in alpha cells. For exact *q* values, corrected for multiple testing, see Supplementary Table 7d. **c–h**, T2D-associated DMRs in alpha cells of genes previously linked to T2D and/or islet function, or in genes encoding epigenetic enzymes: *SNRPN* (**c**), *SLC52A2/(TMEM249)* **d**), *CHURC1* (**e**), *TULP4* (**f**), *KDM1B* (**g**) and *KDM3B* (**h**). The DMRs were analysed and visualized by dmrseq[25] including ten permutations. **i**, The pre-T2D/T2D-associated DMRs in alpha cells are enriched for putative binding motifs of TFs such as HNF1B and OCT4, as analysed by HOMER (http://homer.ucsd.edu/homer/motif/; $q < 0.05$). **j**, Venn diagram and table showing pre-T2D/T2D-associated DMRs in alpha cells overlapping with binding sites for five islet-specific TFs: FOXA2, MAFB, NKX2.2, NKX6.1 and PDX1 (ref. 24). Overlaps are enriched with $P ≤ 0.004$ based on region-based permutation tests. For individual *P* values, see Supplementary Table 7f. **k**, 103 of 849 (12%) T2D candidate genes identified by GWAS[2] overlap with genes that had T2D-associated DMRs in alpha cells annotated to them (HES = 7.34, $P = 1.37 × 10^{-238}$ based on a one-tailed hypergeometric test). **l**, A T2D-associated SNP[2] (rs883541, dashed line) is located in a pre-T2D/T2D-associated alpha cell DMR, annotated to *WIPI1* (based on dmrseq[25]). **m**, pre-T2D/T2D-associated differences in the absolute DNA methylation level (%) of individual CpG sites annotated to *GCG* in alpha cells from seven non-diabetic donors and three donors with pre-T2D/T2D, \**$P < 0.05$, \*\***$P < 0.01$ based on two-tailed Wilcoxon rank-sum test. Data are presented as the mean ± s.e.m. Women and men are represented by filled and open circles, respectively. In **c–h** and **l**, locations of the methylation loci are shown with black lines at the bottom, and average smoothed lines based on the dmrseq analysis are shown in pink (pre-T2D/T2D) or blue (control), including the standard deviation. The shaded area shows the DMR. Annotations to CpG island regions and genes are shown below each graph.

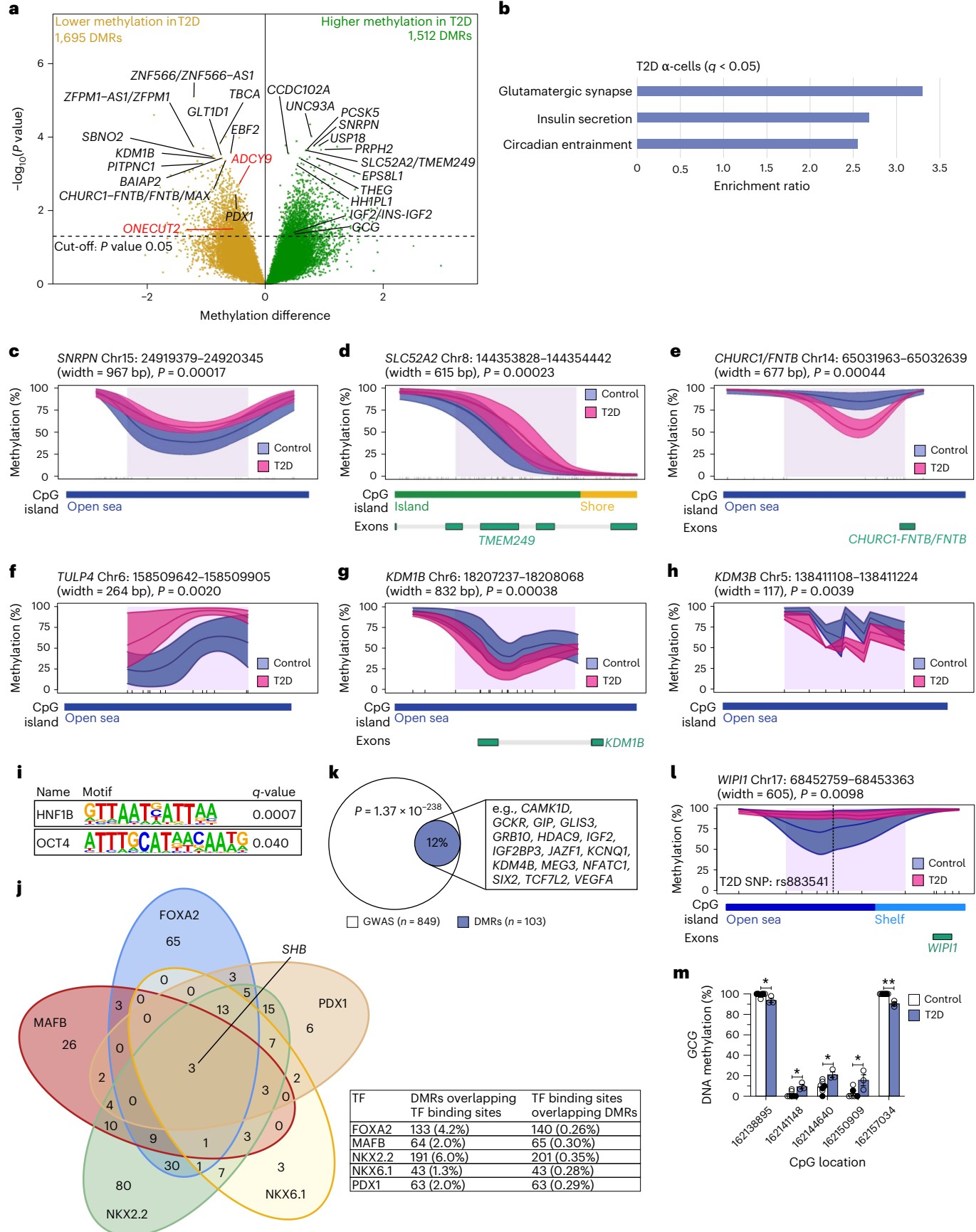

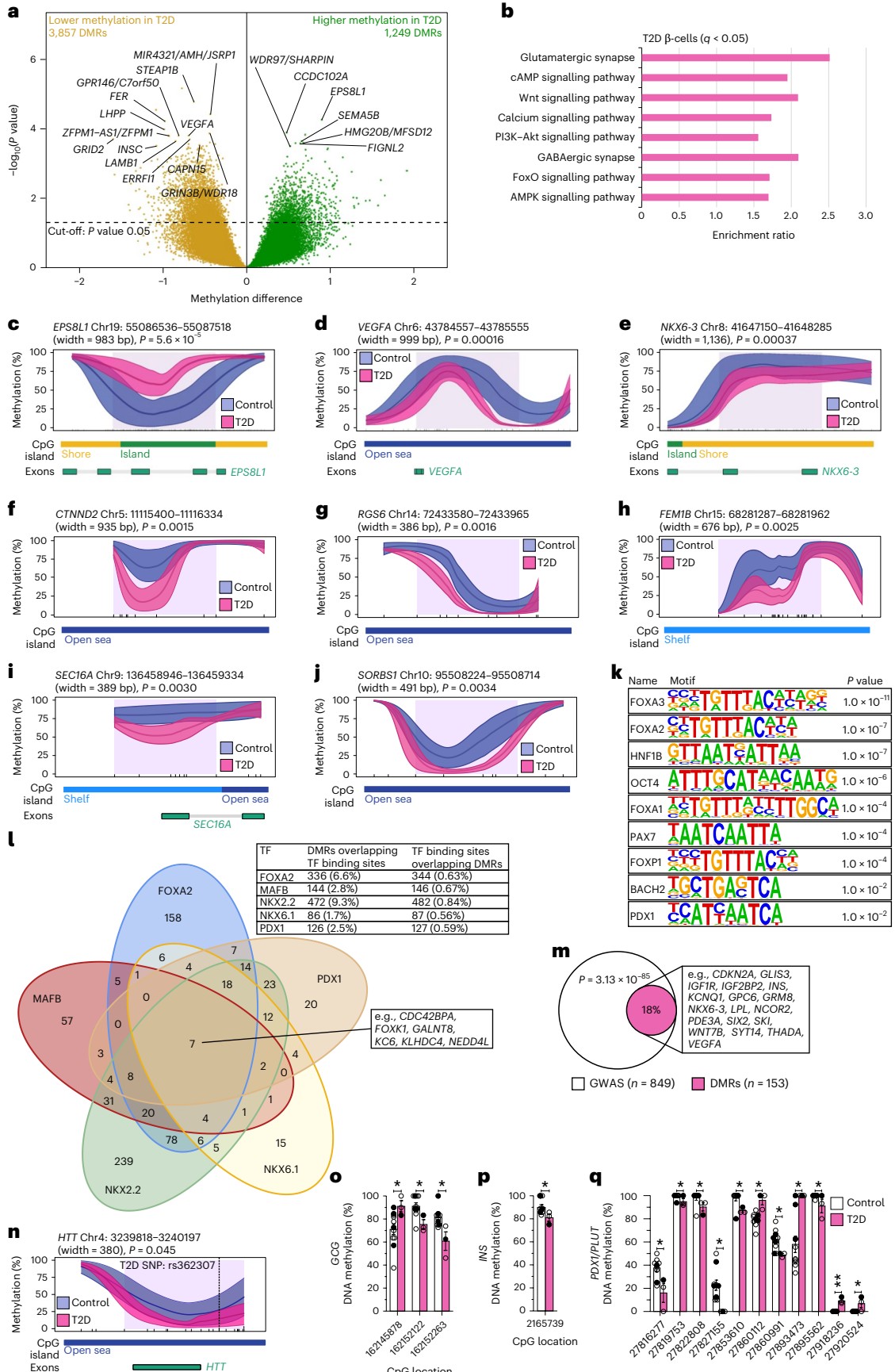

islets,[26] including *GABRA2*, *SOCS1*, *SYT1*, *PAX5*, *CABLES1* and *TBC1D4*; and 90% of these went in the same direction (Extended Data Fig. 5a–c and Supplementary Table 10b). Moreover, 20 and 19 DEGs found in our beta cells overlap with T2D-associated beta cell DEGs found in studies by Walker et al. and Martínez-López et al., respectively (Supplementary Table 10c,d)[46,47]. Additionally, 88 genes exhibited both T2D-associated DEGs and DMRs in beta cells (Supplementary Table 10e). For the top DEGs (Fig. 7c), we identified individual CpGs annotated

**Fig. 6 | Key islet and T2D genes exhibit pre-T2D/T2D-associated DMRs in human beta cells. a**, Volcano plot showing hypomethylated and hypermethylated DMRs in pre-T2D/T2D cases (*n* = 3) versus non-diabetic controls (*n* = 11) in beta cells (*P* < 0.05 based on dmrseq[25] and ten permutations). The 'methylation difference' on the *x* axis represents the beta coefficient values generated by dmrseq and included in Supplementary Table 8a. **b**, Selected enriched KEGG pathways among the 3,020 genes with T2D-associated DMRs in beta cells. For exact *q* values, corrected for multiple testing, see Supplementary Table 8d. **c–j**, T2D-associated DMRs in beta cells of genes previously linked to T2D and/or islet function, including *EPS8L1* (**c**), *VEGFA* (**d**), *NKX6.3* (**e**), *CTNND2* (**f**), *RGS6* (**g**), *FEM1B* (**h**), *SEC16A* (**i**) and *SORBS1* (**j**). The DMRs were analysed and visualized by dmrseq[25] including ten permutations. **k**, The T2D-associated DMRs in beta cells are enriched for putative binding motifs of TF such as FOXA3, FOXA2, HNF1B, OCT4, FOXA1, PAX7, FOXP1, BACH2 and PDX1, as analysed by HOMER (http://homer.ucsd.edu/homer/motif/, *P* < 0.05). **l**, Venn diagram and table showing T2D-associated DMRs in beta cells overlapping with binding sites for five islet-specific TFs: FOXA2, MAFB, NKX2.2, NKX6.1 and PDX1 (ref. 24).

Overlaps are enriched with *P* ≤ 0.001 based on region-based permutation tests. For individual *P* values, see Supplementary Table 8f. **m**, 153 of 849 (18%) T2D candidate genes identified by GWAS[2] overlap with genes that had T2D-associated DMRs in beta cells annotated to them (HES = 2.69, *P* = 3.13 × 10^{-85} based on a one-tailed hypergeometric test). **n**, A T2D-associated SNP[2] (rs362307; dashed line) is located in a T2D-associated beta cell DMR annotated to *HTT*. **o–q**, T2D-associated differences in the absolute DNA methylation level (%) of individual CpG sites annotated to *GCG* (**o**), *INS* (**p**) and *PDX1* (**q**) in beta cells from 11 (**o** and **q**) or 8 (**p**) non-diabetic donors and 3 donors with pre-T2D/T2D. **P* < 0.05, ***P* < 0.01 based on two-tailed Wilcoxon rank-sum test. Data are presented as the mean ± s.e.m. Women and men are represented by filled and open circles, respectively. In **c–j** and **n**, locations of the methylation loci are shown with black lines at the bottom, and average, smoothed lines based on the dmrseq analysis are shown in pink (pre-T2D/T2D) or blue (controls), including the standard deviation. The shaded area shows the DMR. Annotations to CpG island regions and genes are shown below each graph.

---

to *ADCY9*, *ZNF431*, *ZSWIM9*, *GNAL*, *ONECUT2*, *SNX1* and *TRAPPC13* with altered methylation in beta cells from donors with pre-T2D/T2D versus controls (Extended Data Fig. 5d–j). In line with this, CpGs annotated to *ADCY9* (*n* = 5), *ZNF431* (*n* = 1), *ONECUT2* (*n* = 2) and *SNX1* (*n* = 1) show differential islet methylation in T2D[12], supporting epigenetic dysregulation of these genes (Supplementary Table 10a,f). Notably, three T2D-associated islet DEGs[26] were differentially expressed in both alpha and beta cells in this study—*GNAL*, *NTN1* and *PRKCH* (Extended Data Fig. 5a,k and Supplementary Tables 9b and 10b).

### Exploring and visualizing DNA methylation by a web-based tool, alpha-beta-methylome

We proceeded to develop a web-based tool, alpha-beta-methylome (https://alpha-beta-methylome.serve.scilifelab.se/app/alpha-beta-methylome/), a comprehensive open resource based on the human WGBS and RNA-seq data included in this study (Fig. 1a and Supplementary Table 1c), for investigators to explore and visualize the impact of age, sex and pre-T2D/T2D on autosomal DNA methylation and/or expression of any gene in alpha and beta cells, as well as differences between alpha and beta cells, with the option to select *P*-value cut-offs and DNA methylation sequencing coverage. More detailed information is available at https://alpha-beta-methylome.serve.scilifelab.se/app/alpha-beta-methylome/.

### Functional follow-up identifies T2D-associated genes that impair beta cell function

We then explored whether any of the T2D-associated DEGs presented in Supplementary Tables 9 and 10 have a functional role in alpha or beta cells. Protein-coding DEGs with FDR < 10% were selected for functional investigations, resulting in eight DEGs in beta

cells (Extended Data Fig. 6a). Among the selected T2D-associated DEGs, *ADCY9*, *FBLL1*, *ZNF431* and *ZSWIM9* were downregulated, while *GNAL*, *ONECUT2*, *SNX1* and *TRAPPC13* were upregulated in beta cells from donors with pre-T2D/T2D (Fig. 7b,c and Extended Data Fig. 6a).

A systematic PubMed search showed that little, or nothing, was known about these genes in relation to beta cell function, and *GNAL*, *TRAPPC13*, *ZNF431*, *ZSWIM9* and *FBLL1* had previously not been linked to insulin secretion or beta cell function. *SNX1* deficiency increased cAMP and insulin secretion from MIN6B1 cells in response to a glucagon-like peptide-1 analogue[48]. Reduced *Onecut2* expression in rat beta cells led to increased granuphilin expression, linked to decreased insulin secretion[49]. *Adcy9*-deficient insulinoma cells showed reduced glucose-stimulated cAMP response and insulin secretion[50].

To mimic the situation in pre-T2D/T2D, selected genes showing lower expression in beta cells from donors with pre-T2D/T2D, that is, *ADCY9*, *FBLL1*, *ZNF431* and *ZSWIM9*, were silenced using siRNA transfection in both human islets, and rat INS-1 832/13 beta cells (hereafter called INS-1 beta cells; Fig. 7c–e). We chose this cell line rather than EndoC-βH1 as, in our experiments, INS-1 beta cells behaved more like primary human beta cells than the fetal human EndoC-βH1 in gene manipulation experiments (Extended Data Fig. 6b)[12]. siRNA transfection resulted in a 40–80% reduction in mRNA expression compared to islets/beta cells transfected with a negative control siRNA (siNC; Fig. 7d,e). Unfortunately, the *ZNF431*-targeting siRNA in islets failed in silencing the expression, and *Znf431* is not expressed in INS-1 beta cells. *ZNF431* was hence not investigated further. Interestingly, silencing of *ADCY9*, *FBLL1* or *ZSWIM9* reduced glucose-stimulated insulin secretion (GSIS) in both human islets

---

**Fig. 7 | Functional follow-up of T2D-associated DEGs in human alpha and beta cells. a,b**, Volcano plots showing T2D-associated DEGs in human alpha (**a**) and beta (**b**) cells, *P* < 0.05 based on DESeq2 (ref. 82). **c**, mRNA expression of T2D-associated beta cell DEGs selected for functional follow-up based on *q* < 0.1 and DESeq2 (ref. 82). Data from 15 non-diabetic donors and 7 donors with T2D. Women and men are represented by filled and open circles, respectively. Data are presented as the mean ± s.e.m. **d**, siRNA-mediated knockdown of *ADCY9*, *FBLL1* and *ZSWIM9* in human islets from five donors. **e**, Silencing of *Adcy9*, *Fbll1* and *Zswim9* using siRNA in rat INS-1 832/13 beta cells. *n* = 4 (*Fbll1*) or 6 (*Adcy9* and *Zswim9*) biological replicates. **f**, Insulin secretion from human islets stimulated with 1 mM and 20 mM glucose. Measurements were performed on islets from the same five donors as in **d**. Data are presented as the mean ± s.e.m. **g**, Insulin secretion from INS-1 832/13 beta cells during 1 h of stimulation with low (2.8 mM) or high (16.7 mM) glucose. *n* = 6 biological replicates. **h**, Depolarization-induced insulin secretion (35 mM K⁺) from INS-1 832/13 beta cells. *n* = 6 biological replicates from the same experiments as **e**. **i**, Overexpression of *GNAL* (OE-*GNAL*), *ONECUT2* (OE-*ONECUT2*), *SNX1* (OE-*SNX1*) or *TRAPPC13* (OE-*TRAPPC13*)

in human islets from five (*SNX1*) or six (*GNAL*, *ONECUT2* and *TRAPPC13*) donors. **j**, OE-*GNAL*, OE-*ONECUT2*, OE-*SNX1* and OE-*TRAPPC13* in INS-1 832/13 beta cells. *n* = 5 (*ONECUT2*) or 6 (*GNAL*, *SNX1* and *TRAPPC13*) biological replicates. **k**, Insulin secretion from human islets stimulated with 1 mM or 20 mM glucose for 1 h. Measurements were performed on islets from the same donors as in **i**. **l,m**, Insulin secretion from INS-1 832/13 beta cells during 1 h of stimulation with 2.8 mM or 16.7 mM glucose in absolute values (**l**) and as fold change, that is, secretion at 16.7 mM divided by secretion at 2.8 mM glucose (**m**). *n* = 6 biological replicates. **n**, Depolarization-induced insulin secretion by stimulation with 35 mM K⁺ in INS-1 832/13 beta cells. *n* = 3 biological replicates. In **d–n**, data are presented as the mean ± s.e.m. and **P* < 0.05, ***P* < 0.01, ****P* < 0.001 versus siNC or GFP, based on two-tailed (**d–n**) paired Student's *t*-tests. The same colour represents paired samples included in the same experiment, for human islets from the same donor and for the INS-1 832/13 beta cells from the same cell passage. The *q* values for **b,c** were corrected for multiple testing, while the other *P* values throughout were not.

and INS-1 beta cells (Fig. 7f,g) compared to siNC, with *ADCY9* deficiency showing the strongest effect. Both *Adcy9*-deficient and *Zswim9*-deficient INS-1 beta cells showed reduced insulin secretion in response to K⁺ stimulation relative to siNC (Fig. 7h), indicative of

issues downstream of closing ATP-sensitive potassium channels in the insulin secretion pathway.

Next, the selected genes that exhibited higher beta cell expression from donors with pre-T2D/T2D, *GNAL*, *ONECUT2*, *SNX1* and

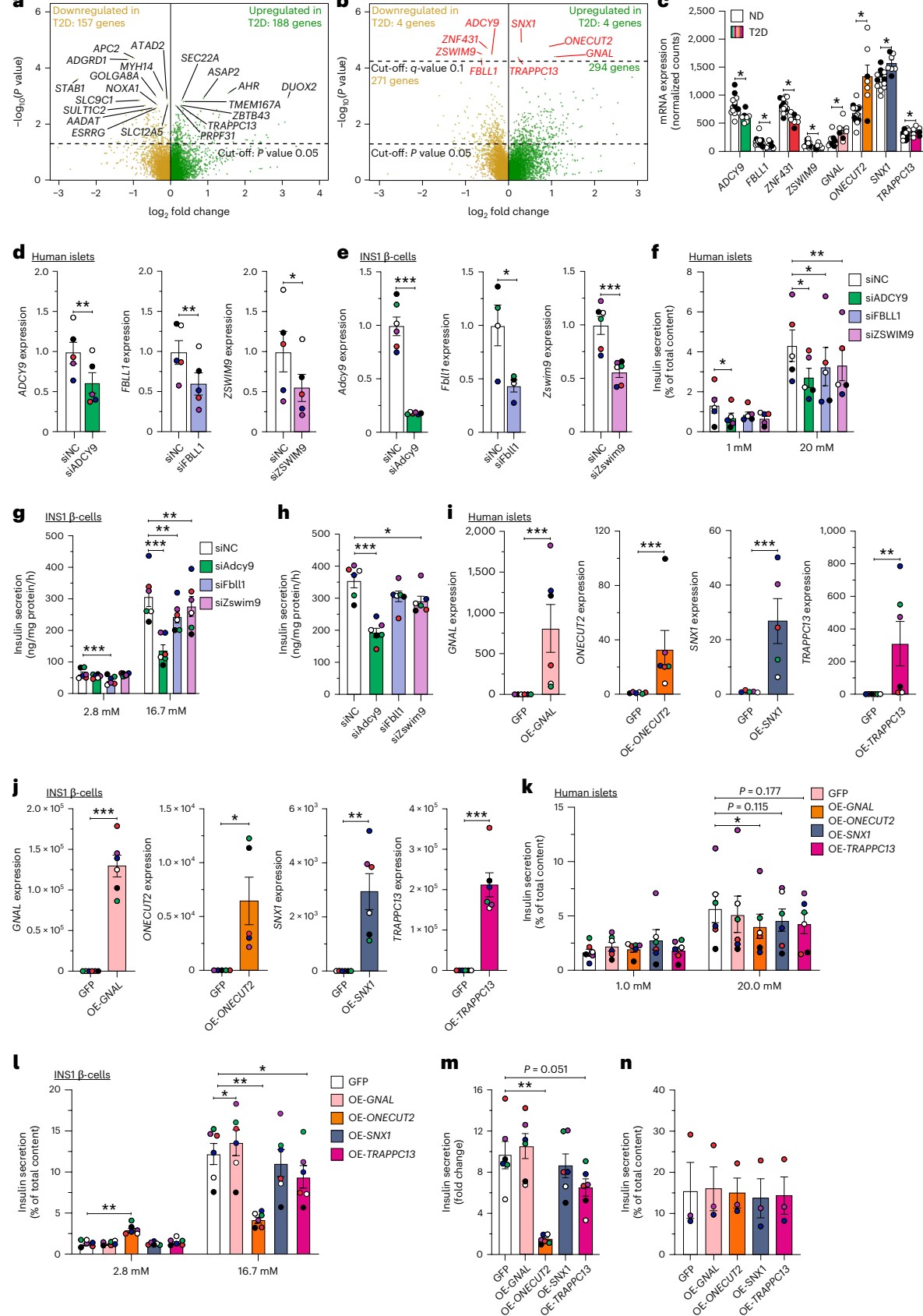

*TRAPPC13*, were overexpressed using lentiviral transduction in human islets and INS-1 beta cells (Fig. 7c,i,j). Interestingly, only overexpression of *ONECUT2* (OE-*ONECUT2*) significantly reduced GSIS in human islets (Fig. 7k), while OE-*ONECUT2* or overexpression of *TRAPPC13* (OE-*TRAPPC13*) significantly reduced GSIS in INS-1 beta cells (Fig. 7l). It is worth mentioning that in INS-1 beta cells, OE-*ONECUT2* caused a severe reduction in GSIS and elevated basal secretion, similar to what is observed in individuals with T2D[51]. These changes resulted in reduced fold change of insulin secretion, that is, secretion at high glucose divided by secretion at low glucose, for OE-*ONECUT2* and OE-*TRAPPC13* (Fig. 7m). Overexpression of neither candidate affected $K^+$-stimulated insulin secretion in INS-1 beta cells (Fig. 7n), suggesting that OE-*ONECUT2* and OE-*TRAPPC13* impair insulin secretion through mechanisms acting proximal to closure of ATP-sensitive potassium channels, for example in mitochondrial function.

### OE-*ONECUT2* causes reduced mitochondrial activity and Adcy9 deficiency impairs stimulus–secretion coupling

Because manipulation of *ADCY9* and *ONECUT2* had the strongest effects on insulin secretion (Fig. 7d–m), we decided to dissect the molecular mechanisms of these perturbances in INS-1 beta cells. We confirmed the *Adcy9* deficiency and OE-*ONECUT2* at protein levels by western blot analysis (Extended Data Fig. 6c,d). To determine whether *Adcy9* deficiency and/or OE-*ONECUT2* affect mitochondrial metabolism, we measured oxygen consumption rates (OCRs) using the Seahorse analyser. OE-*ONECUT2* reduced mitochondrial respiration with lower basal and glucose-stimulated respiration, ATP production-coupled respiration, proton leak, maximal mitochondrial respiration and spare respiratory capacity in INS-1 beta cells, versus GFP (Fig. 8a). *Adcy9* deficiency did not affect the OCR (Extended Data Fig. 6e). In line, OE-*ONECUT2* caused a small significant reduction in ATP5A in complex 5 of the electron transport chain (Extended Data Fig. 6f). Using PercevalHR to assess ATP:ADP in live single cells[52], we found impaired cytosolic ATP levels in OE-*ONECUT2* INS-1 beta cells (Fig. 8b), demonstrating reduced mitochondrial activity in these beta cells. These data support that OE-*ONECUT2* dampens mitochondrial activity, while *Adcy9* deficiency impairs stimulus–secretion coupling in beta cells. The latter fits well with the extensive literature on the role of cAMP in enhancing insulin granule exocytosis in beta cells[53].

To further characterize the mechanisms underlying the dysfunction caused by *ADCY9* deficiency and OE-*ONECUT2*, we performed global transcriptomic analyses in human islets. We found 57 and 114 DEGs in *ADCY9*-deficient and OE-*ONECUT2* islets, respectively ($q < 0.1$; Supplementary Tables 11a and 12a). Interestingly, *NR4A3* and *PCSK2* were DEGs in both *ADCY9*-deficient islets and in T2D beta cells (Supplementary Tables 10a and 11a). In addition to *ONECUT2*, also *SYNPO2*, *CFB*, *VEGFB* and *PER3* were DEGs in both OE-*ONECUT2* islets and T2D beta cells (Supplementary Tables 10a and 12a). *HDAC9* and *AZIN1* are T2D-associated genes in GWAS[2], and DEGs in *ADCY9*-deficient (Supplementary Table 11a) and OE-*ONECUT2* islets

(Supplementary Table 12a), respectively. In line with the impaired ATP production in OE-*ONECUT2* INS-1 beta cells, OE-*ONECUT2* human islets exhibited reduced expression of genes involved in oxidative phosphorylation (*NDUFA1*, *NDUFA7* and *COX10*; Extended Data Fig. 6g) and glycolysis (*GPI*, *ALDOA*, *ENO1* and *ENO2*; Supplementary Table 12a). Using gene-set enrichment analysis (GSEA), we tested whether sets of related genes had altered expression in *ADCY9*-deficient or OE-*ONECUT2* islets. GSEA yielded 3 and 50 gene sets with downregulated expression, and two and one gene sets with upregulated expression in *ADCY9*-deficient and OE-*ONECUT2* islets, respectively ($q < 0.05$; Supplementary Tables 11b and 12b). Interestingly, in OE-*ONECUT2* islets, downregulated gene sets include cellular response to carbohydrates and glucose, and regulation of peptide secretion and insulin secretion (Fig. 8c,d). Individual genes contributing to enrichment for 'regulation of insulin secretion' in OE-*ONECUT2* islets are presented in Fig. 8d and Extended Data Fig. 6h. In *ADCY9*-deficient islets, upregulated gene sets were linked to apoptosis (Extended Data Fig. 6i and Supplementary Table 11b), and cAMP signalling and insulin secretion were among enriched KEGG pathways (Supplementary Table 11c).

Next, because ONECUT2 is a TF, we used weighted gene correlation network analysis (WGCNA)[54] on 114 DEGs from OE-*ONECUT2* in human islets ($q < 0.1$; Supplementary Table 12a) to test whether *ONECUT2* is part of a dysregulated network. WGCNA revealed a network with *ONECUT2* and 107 other genes, where *ONECUT2* had stronger connection to 30 DEGs (Fig. 8e and Supplementary Table 12c). Four genes (*ONECUT2*, *SYNPO2*, *PER3* and *VEGFB*) were both in the WGCNA *ONECUT2* network and among T2D-associated beta cell DEGs (Supplementary Table 10a).

We proceeded exploring *Onecut2* in a rodent diabetes model, the GK rat. We recently demonstrated mitochondrial dysfunction in GK islets[12]. In line with beta cells from donors with pre-T2D/T2D showing elevated *ONECUT2* expression (Fig. 7c), islets from GK rats had increased *Onecut2* expression versus non-diabetic Wistar rats (Fig. 8f). In line with previous studies, islet *Ins1* expression was decreased ($P_{nominal} = 0.12$) and blood glucose increased in GK versus Wistar rats (Fig. 8g,h).

Finally, we showed that silencing ONECUT2 in EndoC-βH1 beta cells exposed to high glucose plus palmitate (HGP; 19 mM glucose plus 1 mM palmitate), mimicking a diabetogenic environment, could rescue the perturbed insulin secretion seen in beta cells exposed to HGP (Extended Data Fig. 6j,k).

Our data support ONECUT2 as a key transcription factor in beta cells, controlling expression of genes involved in mitochondrial function, glucose homeostasis and insulin secretion.

## Discussion

Here, we provide a unique resource, alpha-beta-methylome (https://alpha-beta-methylome.serve.scilifelab.se/app/alpha-beta-methylome/), where the community can explore associations between T2D, age and sex, and cell-type specific methylation and gene expression

**Fig. 8 | Dysregulation of *ONECUT2* leads to perturbed mitochondrial function, reduced insulin secretion and dysregulation of many genes in beta cells.** **a**, Reduced OCR in INS-1 832/13 beta cells overexpressing *ONECUT2* (OE-*ONECUT2*). The left graph shows the oxygen consumption trace, and the right graph the quantification of derived respiratory parameters. $n = 4$ independent experiments. *$P < 0.05$ based on two-tailed paired Student's *t*-tests. **b**, Reduced ATP production in OE-*ONECUT2* INS-1 832/13 beta cells, $n = 73$ (GFP) and 116 (OE-*ONECUT2*) cells from six different experiments. ***$P = 4.61 \times 10^{-14}$ based on a two-tailed Student's *t*-test (right). **c**, emapplot visualizing results of the GSEA of the transcriptome data generated in OE-*ONECUT2* versus control (GFP) human islets of six non-diabetic donors presented in Supplementary Table 12. The R packages clusterProfiler and enrichplot were used to generate the emapplot, and 23 gene sets are preselected. GSEA was corrected for multiple testing. **d**, The genes contributing to the enrichment for the downregulated gene set 'regulation

of insulin secretion' in the GSEA of OE-*ONECUT2* human islets, as well as selected individual downregulated genes in the gene set: *FOXO1* (*$P = 0.026$), *G6PC2* (*$P = 0.0152$), *GCK* (**$P = 0.0087$), *GIPR* (**$P = 0.0022$), *GPR119* (*$P = 0.0441$), *PDX1* (*$P = 0.0441$) and *SLC2A2* (*$P = 0.0441$). *P* values are based on two-tailed Mann–Whitney *U*-tests and presenting unlogged data. $n = 6$ biological replicates of cells expressing *GFP* or *ONECUT2*. **e**, WGCNA based on weighted correlations among OE-*ONECUT2* DEGs with $q < 0.1$ showed that *ONECUT2* is part of an expression cluster containing 114 DEGs (in network) and 108 are shown in the figure (due to Cytoscape cut-off 0.55), with direct connection to 30 DEGs. **f–h**, Male GK rats, a T2D animal model, showed higher *Onecut2* (*$P = 0.0458$; **f**), and nominally lower *Ins* ($P = 0.12$; **g**) expression in islets, as well as higher blood glucose levels (***$P = 6.11 \times 10^{-7}$; **h**) compared with male control Wistar rats ($n = 5$) based on two-tailed Student's *t*-tests. Data in **a**, **b**, **d** and **f–h** are presented as the mean ± s.e.m. *P* values presented in **a**, **b**, **d** and **f–h** were not corrected for multiple testing.

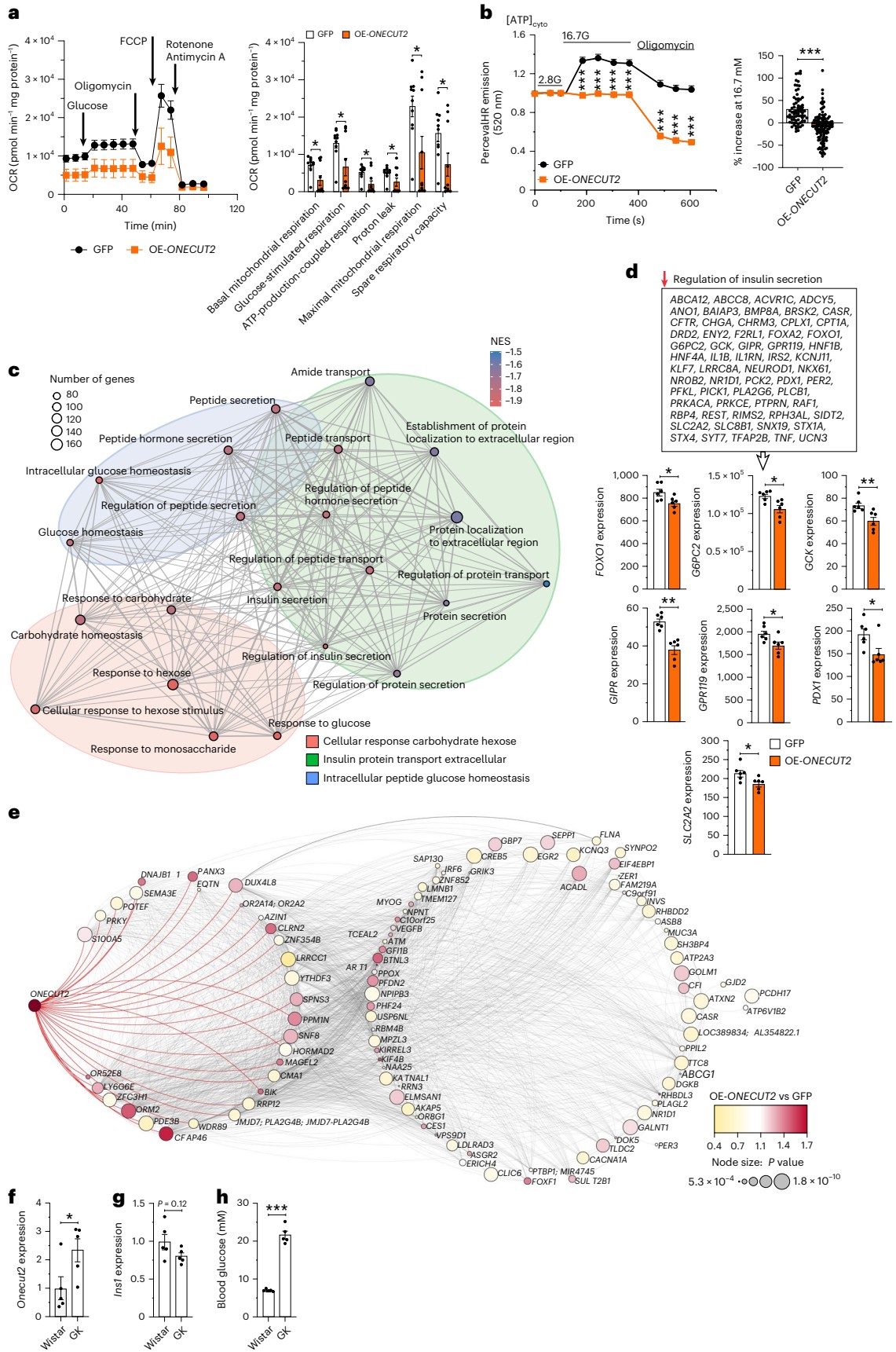

in human alpha and beta cells. By comprehensive analyses, we discovered regulatory regions which were unmethylated in one cell type, while they were highly methylated in the other, near genes with central roles in alpha and/or beta cells, including *INS/INS-IGF2*, *PDX1/PLUT1*, *MAFA*, *PCSK1*, *GCG* and *ARX*. The cell-specific epigenetic pattern was clearly linked to gene expression, and half of genes with DMRs were DEGs in alpha versus beta cells. Intriguingly, by epigenetic editing using CRISPR–dCas–DNMT3A[33], we increased *INS* promoter methylation, causing decreased *INS* expression and insulin levels in beta cells, providing evidence that methylation directly regulates transcription of this important gene. By integrating beta cell methylomes and transcriptomes from people with or without T2D, with comprehensive functional analyses in human islets and beta cells, we discovered that *ONECUT2* has a key role in beta cell function and T2D pathogenesis.

TFs establish and maintain alpha and beta cell identity. Dysregulation of TFs contributes to islet dysfunction and diabetes[55,56]. By dissecting methylomes, we discovered DMRs with large methylation differences in alpha versus beta cells, annotated to such TFs. Methylation in these DMRs likely controls TF expression and thereby cell fate. For example, *ARX*, essential for alpha cells, is hypomethylated and highly expressed in alpha cells, while it is hypermethylated and non-expressed in beta cells. *MAFA* and *PDX1*, essential for beta cells, were hypomethylated and highly expressed in the beta cells. We found cell-specific DMRs for *NEUROG3*, *KLF4* and *PAX6*, important for alpha and beta cells[56]. Long non-coding RNAs control TFs in alpha and beta cells. We identified DMRs annotated to *PLUT*, regulating *PDX1* and *PAUPAR*, regulating *PAX6* (refs. [57,58]). We found cell-specific DMRs in *HNF1A*, *HNF1B* and *HNF4A*, TFs contributing to maturity-onset diabetes of the young, and in *PAX5*, an upregulated TF in islets from people with T2D[26]. By using our web tool, alpha-beta-methylome, we found T2D-associated methylation changes annotated to these TFs and long non-coding RNAs. In line with this, we previously discovered hypermethylation of *PDX1* in islets of donors with T2D[8,11]. Additionally, DMRs identified in this study overlap with sequences bound by FOXA2, MAFB, NKX2.2, NKX6.1 and PDX1.

T2D-associated SNPs impact islet function[2]. Previously, we discovered associations between genetics and methylation as well as the chromatin structure in human islets[30,31,59]. Here, we found that numerous genes linked to T2D SNPs also have T2D-associated DMRs in alpha or beta cells, and/or show differential methylation between alpha and beta cells, for example, *INS*, *TCF7L2* (top T2D gene)[2] and *GLP1R* (treatment target)[60], as well as TFs mentioned above. Additionally, T2D SNPs are located within DMRs, further supporting interactions between genetics and epigenetics in T2D. We also found that 45% and 22% of T2D-associated genes/loci identified by GWAS overlap the alpha versus beta cell DMRs and DEGs, respectively. The reason for the higher overlap with genes annotated to DMRs than to DEGs could be because DNA methylation is more stable than gene expression, where expression of some genes represents more of a 'snapshot' at the time samples were collected. However, this needs to be further investigated.

We add *ONECUT2* 'as a piece of the puzzle' of genes causing hampered mitochondrial activity and reduced GSIS in islets/beta cells, contributing to T2D (Extended Data Fig. 6l)[12,26,61,62]. When analysing ~300,000 beta cells per donor, *ONECUT2* expression was increased ~2-fold in people with pre-T2D/T2D, and *ONECUT2* expression was higher in beta versus alpha cells. OE-*ONECUT2* in human islets and INS-1 beta cells, mimicking T2D, caused reduced GSIS, OCR, ATP production and expression of gene sets regulating glucose metabolism and insulin secretion. The latter includes reduced expression of *GCK*, which encodes glucokinase, the rate-limiting enzyme in glycolysis. This could be an important change, leading to reduced production of pyruvate to fuel the mitochondrial tricarboxylic acid cycle, oxidative phosphorylation and ATP production[63]. Rodent models showed downregulated

*Onecut2* when overexpressing *miR-29a/miR-29b/miR-29c*[64], and links between insulin resistance and downregulated ductal *Onecut2/ONECUT2* (ref. [65]), supporting differences between mice and men[66]. In 10x Genomics' scRNA-seq data from non-diabetics (5,000–10,000 islet cells), *ONECUT2* is predominantly expressed in ductal cells and to a lesser degree in beta cells, with limited expression in other cell types[67]. While our data strongly implicate *ONECUT2* in T2D pathogenesis, one should keep in mind that the overexpression of ONECUT2 is stronger in our in vitro model than in T2D beta cells in vivo.

Interestingly, ONECUT2 and ONECUT1 have nearly identical DNA binding sites, but data from our study and published studies suggest different roles for these two TFs[68]. Mutations and common variants in *ONECUT1* contribute to neonatal diabetes and T2D, respectively, and evidence supports that ONECUT1 impacts pancreas development[69–71]. However, *ONECUT1* expression was not detected in beta cells in our study, nor in fetal or adult beta cells sorted by Blodgett et al.[27]. Furthermore, there was a 10-fold higher expression of *ONECUT2* in adult compared to fetal beta cells[27], and *ONECUT1* showed very low expression compared to *ONECUT2* in beta cells in our human islet single-cell RNA-seq data[67].

This study focused on dissecting cell-specific and pre-T2D/T2D-associated DNA methylation in alpha and beta cells. However, since we previously showed that age and sex impact DNA methylation, gene expression and insulin secretion in human islets[72,73], we included analyses of age and sex associations on methylation and gene expression in our web tool, alpha-beta-methylome. Interestingly, sex-associated methylation and gene expression differences previously found in islets of, for example, *BCL11A*[73], were also identified in beta cells by exploring alpha-beta-methylome. *Bcl11a*-silencing reduced GSIS in beta cells, mimicking lower *BCL11A* expression and insulin secretion found in islets from males[73]. Additionally, age-associated beta cell methylation of, for example, *KLF14*, *FHL2* and *ZNF518B*, was identified using our web tool. These genes also showed age-associated methylation in islets[72]. We also explored whether results generated by Manduchi et al.[28] could be validated by using our web tool and, for example, we found age-associated hypomethylation of specific CpG sites in beta cells of genes involved in insulin secretion, such as *FFAR1*, *RAPGEF4*, *RAB11FIP5*, *SMAD2*, *MYO5A*, *PDX1* and *PTPRN*. It should be noted that while our web tool performs a linear model between age and DNA methylation of individual CpG sites in beta cells from donors with an age range between 36 and 81 years, Manduchi et al. measured the average methylation differences between their older (39–58 years) and younger (18–35 years) donors.

This study provides a valuable resource for islet research by identifying genomic regions where DNA methylation seems to regulate cell-specific gene expression in human alpha and beta cells. For example, we discovered a region annotated to the promoter of *SLC2A2*, encoding GLUT2, with hypomethylation in beta cells compared to alpha cells. In line with this, we found *SLC2A2* expression in beta but not alpha cells. GLUT2 is an important glucose transporter in rodent islets[74], but its importance in human islets has been questioned[75]. Nevertheless, Hattersley's group found *SLC2A2* mutations causing neonatal diabetes, and they proposed that GLUT2 affects insulin secretion in humans[76]. The epigenetic and expression data of *SLC2A2* in human beta cells in the present study together with data from our previous study, where we found reduced *SLC2A2* expression in islets from donors with T2D and that *SLC2A2* knockdown reduced insulin secretion in human islets[26], support a role for GLUT2 in mature beta cells.

We used different cut-offs for unexpressed genes in our study. We first used <2 mean normalized counts to categorize genes as not expressed, and we categorized the remaining genes into low, medium and high expression. In the second analysis, we used ≥20 mean normalized counts in ≥80% of samples for a gene to be considered expressed in alpha and beta cells. The reason for using different cut-offs was that in the second analysis we wanted to include expressed genes that could

have a biological role in the cells, while in the first analysis our goal was to investigate the degree of DNA methylation in relation to four different levels of expression. However, one should keep in mind that the cut-offs are based on our 'best guess'.

This study has strengths and limitations. By using WGBS, we covered most of the methylome, compared with DNA methylation arrays, covering ~5%. We provide alpha and beta cell methylomes and transcriptomes from the same donors. As the number of samples is limited, partly because MARIS requires many islets and WGBS is challenging, future studies should include additional donors. While this study provides new data, some cell-specific findings could be 'validated' in larger cohorts of human islets, for example sex and age associations described above[72,73], and T2D associations described in the results, for example, 65 T2D-associated beta cell DMRs overlap 91 CpGs exhibiting differential islet methylation in T2D[12], supporting their robustness. Functional validations and results from pathway analyses further support the validity of our data. Interestingly, 75.5% of the T2D-associated beta cell DMRs identified in this study were hypomethylated, which is in line with our previous study where 77% of T2D-associated methylation sites had lower methylation in whole islets from donors with T2D[12], and the recent study by Manduchi et al. supporting hypomethylation in beta cells from donors with T2D[28]. Additionally, genes discovered to have alpha versus beta cell DMRs in our study significantly overlapped with the ones found by Manduchi et al., and both our study and theirs found enriched gene pathways affecting T2D and secretion. A limitation is that only a few T2D-associated alpha cell DEGs could be validated in two other studies[26,46]. However, one needs to keep in mind that Walker et al. used a gentler cut-off for inclusion of genes compared to our study[46]. Additionally, since T2D is a heterogeneous and polygenic disease, diverse mechanisms and different genes contribute to the disease in different groups of people[2,77]. This may affect the reproducibility between some studies, despite technical soundness.

Together, we provide comprehensive cell-specific methylomes and transcriptomes in alpha and beta cells from people with or without T2D and a web tool allowing the exploration of epigenetic dysregulations contributing to impaired insulin secretion and potentially T2D.

## Methods

### Ethics statement

Written informed consent to donate organs for biomedical research was obtained from the pancreatic donors or their relatives, and all procedures were approved by the ethics committees at Uppsala University and Lund University (permit nos. 2007-05 and 2011-263) following the Helsinki declaration. Donors were not compensated for their participation. Animal experiments were performed with permission from the Animal Ethics Committee of Lund University (permit no. 5.8.18-04115/2021) in accordance with the legal requirements of the European Community (86/609/EEC).

### Human pancreatic islets

Pancreases were obtained from multi-organ donors, and pancreatic islets were isolated by collagenase digestion and density gradient purification at the Nordic Network for Clinical Islet Transplantation in Uppsala, Sweden. The findings are based on inclusion of both sexes, determined by self-reporting and concordant genetic testing. Sex is considered as a covariate in genome-wide analyses as described below. Supplementary Table 1a presents the clinical characteristics for all 24 donors included in this study, while Supplementary Table 1b presents characteristics for the donors included in each individual analysis of this study. These analyses include (i) pairwise analyses, that is, comparing data in alpha versus beta cells, in donors not diagnosed with T2D and with data available for both alpha and beta cells, and (ii) comparing data in alpha or beta cells from donors with pre-T2D/T2D, based on HbA1c ≥ 6% (≥42 mmol mol$^{-1}$) and/or T2D diagnosis, with data in alpha or beta cells from controls, including donors with

HbA1c < 6% (<42 mmol mol$^{-1}$). Additionally, Supplementary Table 1c presents the characteristics of the donors included in each possible analysis using the web-based tool alpha-beta-methylome (https://alpha-beta-methylome.serve.scilifelab.se/app/alpha-beta-methylome/), which also includes (iii) associations between age and DNA methylation or gene expression using linear regression models in alpha or beta cells from non-diabetic donors, and (iv) sex associations by comparing data in alpha or beta cells from male versus female non-diabetic donors.

### MARIS staining and FACS sorting of human alpha and beta cells

MARIS[23] was used to sort alpha and beta cells from islets from 24 donors. Briefly, islets were dispersed and fixed and then stained with antibodies for 30 min with guinea pig anti-insulin (A0564, 10119708; 1:600 dilution, Dako) and mouse anti-glucagon (MAB1249, HSQ0416111; 1:200 dilution, R&D Systems). Cells were then washed twice in DEPC-treated PBS containing 10% BSA, 1% saponin and RNase inhibitor before secondary staining with donkey anti-guinea pig Alexa Fluor 647 (706-605-148, 129003; 1:400 dilution, Jackson ImmunoResearch) and donkey anti-mouse Alexa Fluor 488 (A-21202, 1644644; 1:400 dilution, Invitrogen). The sorting of alpha and beta cells was performed on an Aria Fusion (BD Biosciences), achieving purity of 92.0% ± 7.6% for alpha cells and 96.5% ± 4.0% for beta cells. We obtained 322,627 ± 73,055 alpha cells and 311,550 ± 61,250 beta cells from each donor. When the number of sorted cells from a donor was deemed sufficient, they were split into two portions for extraction of RNA and DNA, respectively. Sample purity was also determined by deconvolution according Loyfer et al.[20] using *GCG* and *INS* as marker genes for alpha and beta cells, respectively, and either the gene expression (purity of alpha cells = 99.3% ± 0.3% and beta cells = 99.5% ± 0.1%) or DNA methylation (purity of alpha cells = 94.2% ± 0.8% and beta cells = 99.1% ± 0.4%) data generated by RNA-seq or WGBS (see below). Purity for individual samples was included in Supplementary Table 1a.

### RNA and DNA isolation from human alpha and beta cells

After sorting, the alpha and beta cells were pelleted by centrifugation at 3,000*g* for 6 min at 4 °C. Total RNA and DNA were then extracted from the cells using the RecoverAll Total Nucleic Acid Isolation kit (Ambion, Life Technologies) according to the recommended protocol from the manufacturer, except that we incubated the RNA samples in digestive buffer for 3 h at 50 °C and then for 15 min at 80 °C instead of 15 min at 50 °C and 15 min at 80 °C, according to the MARIS protocol. The quality and integrity of the RNA and DNA samples were analysed using the Agilent ScreenTape and Fragment Analyzer systems (Agilent Technologies). Next, RNA and DNA concentrations were measured by Qubit/Quant-iT assay (Thermo Fisher Scientific) to verify sufficient quantity and quality for library preparations.

### RNA-seq

Sequencing libraries were generated from 1 µg RNA from alpha and beta cells using the TruSeq RNA Library Preparation Kit followed by RNA-seq on a NextSeq 500 (Illumina). For preparation and analysis of data, quality and adaptor trimming was done using Trim Galore (https://www.bioinformatics.babraham.ac.uk/projects/trim_galore/), and then Salmon was used for quantification of transcript expression[78]. Quality control was performed using fastQC (https://www.bioinformatics.babraham.ac.uk/projects/fastqc/) and multiQC tools[79]. Downstream analyses were done using R[80]. The R package tximport[81] was used to import transcript-level abundance, estimated counts and transcript lengths into R using Gencode annotation v32. Differential gene expression analysis was done using DESeq2 (ref. 82) which applies Wald's test. For the analysis of DEGs and linear models, we considered a transcript to be expressed in a particular cell type if it has at least 20 reads in 40% of the samples. Correction for age, sex, body mass index (BMI) and days in culture was included in the analyses. Gene symbols

referring to human mRNA expression are capitalized and italicized, and mRNA expression is referred to as 'gene expression'.

## WGBS and data analysis

Genomic DNA (2.2–50 ng) was treated with sodium bisulfite using the EZ DNA Methylation-Gold D5005 kit (Zymo Research) according to the manufacturer's protocol. Sequencing libraries were prepared using the SPlinted Ligation Adapter Tagging (SPLAT) protocol[83]. Cluster generation and paired-end 150-bp read length sequencing were performed using S4 flow cells with a NovaSeq 6000 and v1.5 sequencing chemistry (150 cycles, Illumina). The number of reads, alignment statistics and sequencing depth are reported in Supplementary Table 2. The sequencing data were processed using R[80] and MultiQC[79]. Paired-end reads were trimmed using Trim Galore[84]. Genome indexing was done using Genome Reference Consortium Human Build 38 (GRCh38/hg38). Gene annotations were retrieved using the R package TxDb.Hsapiens. UCSC.hg38.knownGene (version 3.4.6. 2019). Sequencing reads were aligned, and count data were generated using Bismark[85]. Extended Data Figs. 7 and 8 present heat maps and dendrograms of the WGBS samples, respectively.

To identify DMRs, we used dmrseq[25], which uses transformed methylation proportions, and fits a linear regression model using generalized least squares with a nested autoregressive correlated error structure. The dmrseq package 'weighs' samples based on coverage and methylation state, meaning a CpG with 10 of 10 reads methylated is not treated the same as a CpG with 50 of 50 reads methylated, that is, higher coverage allows for more certainty of the methylation state. For the DMR calling, we used the following settings: >2 coverage in 70% of samples, a minimum DMR length of three CpGs and ten permutations. The $t$-statistics from Wald's test were used to evaluate the strength of the covariate's effect within each region, and statistical significance was determined via permutations using a pooled null distribution. The permutations involved calculating an initial test statistic from the observed data, then repeatedly shuffling the data and recalculating the statistic to build a reference distribution. The final $P$ value is the proportion of shuffled results that are as extreme or more extreme than the original statistic. Additionally, dmrseq's cut-off for the scalar value that represents the absolute value for the cut-off of the single CpG coefficient that is used to discover candidate regions was set to 0.20 for the analysis of alpha versus beta cells and to 0.05 for the analysis of pre-T2D/T2D versus controls. DMR annotations to gene and CpG island regions were done using hg38 and annotatr[86]. The analysis of DMRs on autosomal chromosomes included data from both males and females, while X-chromosome and Y-chromosome DMRs were analysed in males and females separately.

## HOMER TF motif discovery

HOMER (v.4.11, http://homer.ucsd.edu/homer/motif/) was run with findMotifsGenome.pl, to search for enrichments of TF binding motifs in identified DMRs.

## Culturing cell lines

The human EndoC-βH1 beta cell line (female origin, EndoCells) was seeded in Matrigel and fibronectin-coated (100 μg ml$^{-1}$ and 2 μg ml$^{-1}$, respectively; Sigma-Aldrich) culture vessels in DMEM containing 5.6 mM glucose, 2% BSA fraction V, 10 mM nicotinamide, 50 μM 2-mercaptoethanol, 5.5 μg ml$^{-1}$ transferrin, 6.7 ng ml$^{-1}$ sodium selenite, 100 units per ml penicillin and 100 μg ml$^{-1}$ streptomycin.

The rat INS-1 832/13 beta cell line (male origin, shared by C. Newgard, Duke Molecular Physiology Institute) was cultured in RPMI 1640 medium with 11.1 mM D-glucose supplemented with 10% FBS, 2 mM L-glutamine, 1 mM sodium pyruvate, 50 μM β-mercaptoethanol, 5 ml penicillin–streptomycin (10,000 U/10 mg ml$^{-1}$) and 10 mM Hepes (HyClone).

All cells were incubated in a humidified atmosphere with 5% CO$_2$ at 37 °C.

## Epigenetic editing, pyrosequencing, MethylationEPIC v2.0 array analyses, real-time qPCR and insulin and glucagon ELISA

Three different sets of epigenetic editing experiments were done in human EndoC-βH1 beta cells, which were seeded (180,000 cells per well) in 48-well plates 48 h before transduction. Next, the cells were transduced with all-in-one lentiviral constructs[33] at a multiplicity of infection (MOI) of 10 in the growth medium without antibiotics (penicillin–streptomycin) for 6 h. The constructs contain inactivated Cas9 (dCas9) fused to the DNA methyltransferase DNMT3A, and either *INS*-DMR-gRNA, a gRNA (sequence TGAGACTATAAAGCCAGCGGGGG) targeting an *INS* DMR (Figs. 2e and 4a,b), or *TH*-DMR-gRNA, a gRNA (sequence GCAGCTGTACTGGTTCACGG) targeting *TH* (Fig. 4c) as well as a potential *INS* enhancer (Fig. 4a,c). In the third set of editing experiments, EndoC-βH1 beta cells were transduced with a construct containing *GCG*-DMR-gRNA, a gRNA (sequence CATGCGTGATTGAAAGTAGA*GCG*) targeting the DMR presented in Figs. 2g and 4d, dCas9 fused to TET1, an enzyme involved in demethylating DNA. The virus-containing medium was then removed and replaced with complete growth medium. Forty-eight hours after transduction, cells were transferred into 24-well plates, and after an additional 48 h, cells were put into selection medium, including 4 μg ml$^{-1}$ puromycin, for 7 days. The selection medium was renewed after 3 days. After puromycin selection, cells from the same well were collected for RNA and DNA extraction, and cells from different wells were collected for protein extraction.

DNA and RNA were then prepared from the EndoC-βH1 beta cells using the Allprep RNA/DNA Mini Kit (80204, QIAGEN, Hilden, Germany) according to the manufacturer's instructions. The DNA and RNA concentrations were measured using a NanoDrop (ND-1000; Thermo Fisher Scientific).

Next, we performed bisulfite conversion using 200 ng of DNA using the EpiTect Bisulfite Kit (59104, QIAGEN) according to the manufacturer´s instructions. Using the PyroMark Assay design software 2.0 (QIAGEN), the following pyrosequencing assays were designed: *INS*-forward primer GATTTTTTTATTTTAGGTTTTAATGGG, *INS*-reverse primer ACAAACCTACTTAATAACCTCTTCTAATA, *INS*-sequencing primer TTTATTTTAGGTTTTAATGGGT, *TH*-forward primer GGGGTGATTTTATTTAAGGAATTTATTTAG, *TH*-reverse primer CCTAACCCACCTAAACTTATCCTT, *TH*-sequencing primer ATTGTTTGTTTGAGGAG, *GCG*-forward primer GGATATGTATAAAATAGGGATGGTTATGG, *GCG*-reverse primer AAAAAAACAACTTAAAAATCCACCTTCTA and *GCG*-sequencing primer AGGGATGGTTATGGG. Pyrosequencing was then run on a PyroMark Q48 autoprep instrument (QIAGEN). Notably, several pyrosequencing assays covering the genomic regions targeted by and surrounding the three investigated gRNAs failed, potentially due to CG-rich regions. Subsequently, we were unable to analyse methylation of all CpG sites in these regions by using pyrosequencing.

We proceeded to use the Infinium MethylationEPIC v2.0 BeadChip array (Illumina) to study DNA methylation of CpG sites covered by EPIC v2.0 probes in the *INS*, *TH* and *GCG* loci in samples from the epigenetic editing experiments. These arrays were analysed at Eurofins, Denmark. DNA was treated with bisulfite using the EZ-96 DNA Methylation kit (Zymo Research). Preprocessing and quality control of the methylation data were performed using R Statistical Software version 4.3.3 (https://www.R-project.org/). Probes on the array that target SNPs, non-CpG sites, cross-reactive probes[87], Y-chromosome-specific probes, probes targeting single-nucleotide variants of common cancer driver mutations and probes with a detection $P$ value > 0.01 in more than 10% of the samples were removed. The intensities from the remaining probes were then corrected for background and dye bias using the NOOB function from the R package minfi (version 1.48.0)[88], and intra-sample normalization performed with the BMIQ function included in the R package wateRmelon (version 2.8.0)[89]. To minimize batch effects, all paired editing samples (control and gRNA) were placed on the same array. Paired $t$-tests were used to analyse the methylation data covered by EPIC

v2.0 probes from 10 kb upstream of the TSS to 10 kb downstream of the gene end of *INS*, *TH* and *GCG*. Here, EPIC v2.0 array covered 56 sites for *INS*, 62 for *TH* and 14 for *GCG*. *M* values were used for all bioinformatic and statistical analyses, and they were then converted to beta values (1–100% methylation) for presenting the results.

cDNA was generated with the RevertAid First Strand synthesis kit (K1622, Thermo Fisher Scientific). Gene expression was analysed by real-time qPCR in triplicates on a 384-well plate with TaqMan assays (Thermo Fisher Scientific) for *INS* (HS00355773_m1), *TH* (Hs00165941_m1), *GCG* (Hs01031536_m1) and *PPIA* (Hs04194521_s1, endogenous control), in a QuantStudio 7 Flex Real Time PCR system (Thermo Fisher Scientific) under default cycling parameters. Data collection and most analysis for epigenetic editing experiments were not performed blinded to the conditions of the experiment because samples needed to be loaded on plates and discs in a certain order. However, EPIC array experiments performed by Eurofins were blinded. No data points were excluded from the analyses.

Total protein from wells with edited EndoC-βH1 beta cells was extracted by using 200 µl RIPA buffer: 0.1% sodium dodecyl sulfate (SDS), 150 nM NaCl, 1% Triton X-100, 50 mM Tris-Cl, pH 8 and ethylenediaminetetraacetic acid (EDTA)-free protease inhibitor (Roche). The protein amount/concentration was analysed using the BCA Protein Assay kit (Thermo Fisher Scientific). Insulin and glucagon levels in beta cells were then analysed by ELISA (insulin: 10-1113-01; glucagon: 10-1271-01, Mercodia). BCA and ELISA plates were read on a CLARIOstar microplate reader (BMG Labtech).

### Gene silencing, overexpression and insulin secretion in human pancreatic islets

Handpicked human islets (100–200) were placed in culture dishes containing RPMI 1640 media (11879020, Thermo Fisher Scientific) with 5 mM glucose, 10% FBS and 200 mM L-glutamine. Human islets were then transfected in a final volume of 2.5 ml per dish containing serum free Opti-MEM, 6.25 µl of Lipofectamine RNAiMAX, 50 nM of Silencer Select Pre-Designed siRNA against *ADCY9* (s1037), *FBLL1* (s196099), *ZNF431* (s46878) *ZSWIM9* (s51624) or negative control no. 2 siRNA (siNC, 4390846; all from Thermo Fisher Scientific). In line with previous studies[12,26], a second transfection was performed 24 h after the first, and all functional experiments were performed 72 h after the first transfection. For overexpression experiments, lentiviral plasmids containing the human phosphoglycerate kinase 1 promoter in front of cDNA of human *GNAL*, *ONECUT2*, *SNX1* or *TRAPPC13*, with and without His-tag, followed by an internal ribosomal entry site and the cDNA for *GFP*, were cloned by GenScript. Lentiviral vectors were produced by Lund University Cell and Gene Therapy core facility. Lentivirus conferring gene expression of only GFP was used as control. Around 100–200 islets were transduced with these lentiviruses for 72 h with a MOI of 5. Gene symbols referring to human mRNA expression are capitalized and italicized, whereas when referring to human protein levels, protein symbols are capitalized but not italicized.

For insulin secretion, six batches of 6–12 human islets per donor were pre-incubated for 30 min in KREBS buffer (2.5 mM CaCl₂, 4.7 mM KCl, 120 mM NaCl, 25 mM NaHCO₃, 1.2 mM KH₂PO₄, 1.2 mM MgSO₄ and 10 mM HEPES) with 1 mM glucose before stimulation for 1 h in KREBS with either 1 mM or 20 mM glucose (*n* = 3 for each condition). After glucose stimulation, islets were dissolved in 100 µl RIPA buffer (50 mM TRIS-HCl, 150 mM NaCl, 0.5 mM sodium deoxycholate, 2 mM EDTA, 50 mM NaF, 1% Triton-X, 0.1% SDS). Insulin was measured by ELISA (10-1113-01, Mercodia), and secreted insulin was normalized to the total insulin content in the islets.

### Gene silencing and overexpression in INS-1 beta cells and EndoC-βH1 beta cells

The INS-1 832/13 beta cells were seeded (250,000 cells per well) in 24-well plates containing 1 ml medium without antibiotics a day before transfection. For gene silencing, cells were transfected the following day, in a total volume of 600 µl of serum free Opti-MEM and 1.5 µl of Lipofectamine RNAiMAX with 25 nM Silencer Select Pre-Designed siRNA against *Adcy9* (s235414), *Fbll1* (s170925) or *Zswim9* (s1587760), or silencer select negative control no. 2 siRNA (4390846) as negative control. All reagents were from Thermo Fisher Scientific. Cells were assayed 72 h after transfection, when reaching 90–100% confluence. For overexpression experiments in INS-1 beta cells, the same lentiviral vectors as described for human islets were used. Cells (79,000 cells per well in 48-well plates) were transduced for 24 h with a MOI of 1.25 and all experiments were performed 72 h after initiation of transduction. EndoC-βH1 beta cells were transfected as described above with *RHOT1*-targeting siRNA (s30650) or the negative control.

### Insulin secretion in INS-1 beta cells and EndoC-βH1 beta cells

Confluent plates of transfected or transduced INS-1 832/13 beta cells or EndoC-βH1 beta cells, the latter cultured overnight in low-glucose media before experiments, were washed twice carefully with pre-warmed Secretion Assay Buffer (SAB), pH 7.2 (1.16 mM MgSO₄, 4.7 mM KCl, 1.2 mM KH₂PO₄, 114 mM NaCl, 2.5 mM CaCl₂, 25.5 mM NaHCO₃, 20 mM HEPES and 0.2% bovine serum albumin) containing 2.8 mM glucose. Cells were then pre-incubated in fresh SAB with 2.8 mM glucose for 2 h. Next, the cells were stimulated for 1 h in SAB with 2.8 mM glucose, 16.7 mM glucose or 2.8 mM glucose with 35 mM KCl at 37 °C. Insulin levels were measured by ELISA (10-1145-01 (rat) or 10-1113-01 (human), Mercodia) and normalized to total protein or total insulin content in each well. Total protein from each well was extracted by using 100–200 µl RIPA buffer: 0.1% SDS, 150 nM NaCl, 1% Triton X-100, 50 mM Tris-Cl, pH 8 and EDTA-free protease inhibitor (Roche). The protein content was analysed by BCA assay (Thermo Fisher Scientific). Plates were read on a CLARIOstar microplate reader.

### Real-time qPCR in human islets, INS-1 beta cells, rat islets and EndoC-βH1 beta cells

Total RNA was extracted with the miRNeasy isolation kit (QIAGEN) according to the manufacturer's instructions. The equipment and reagents mentioned below were purchased from Thermo Fisher Scientific. RNA concentrations were measured using NanoDrop (ND-1000). cDNA was generated with the RevertAid First Strand cDNA synthesis kit. qPCR was performed in triplicates on 384-well plates in a QuantStudio 7 Flex Real Time PCR system under default cycling parameters. TaqMan assays were used to measure expression levels of human *ADCY9* (Hs00181599_m1), human *FBLL1* (Hs01584505_s1), *ZNF431* (Hs00819968_m1), human *ZSWIM9* (Hs00736775_m1), human *ONECUT2* (Hs00191477_m1), human *RHOT1* (Hs00430256_m1), human *SNX1* (Hs00162052_m1), human *TRAPPC13* (Hs01556088_m1), human *GNAL* (Hs00181836_m1), rat *Adcy9* (Rn01424753_m1), rat *Fbll1* (Rn06322963_s1), rat *Zswim9* (Rn01753793_m1), rat *Onecut2* (Rn01265320_m1), rat *Snx1* (Rn01418446_m1), rat *Trappc13* (Rn01493919_m1), rat *Gnal* (Rn01491898_m1) and rat *Ins1* (Rn02121433_g1). Human *PPIA* (Hs04194521_s1) and human *HPRT1* (Hs02800695_m1), as well as rat *Ppia* (Rn00690933_m1) and *Hprt1* (Rn01527840_m1), were used as endogenous controls for normalizing mRNA expression. Data are presented as relative quantification, describing the difference in expression of the gene compared with a control group. Threshold levels of all Ct values were automatically set, and the gene expression levels were normalized using the geometric means of the two endogenous controls, *PPIA* and *HPRT1*. Relative expressions were calculated with the ΔΔCt method.

### Western blot analysis

To extract protein, cells were first lysed in ice-cold RIPA buffer (50 mM Tris-HCl pH 7.4–7.6, 150 mM NaCl, 2 mM EDTA, 1% Triton X-100, 0.5% sodium deoxycholate and 0.1% SDS) containing protease inhibitor cocktail (Sigma-Aldrich). Lysates were spun at 11,000*g* for 3 min to pellet debris and protein concentration of the supernatant was measured

by BCA assay (Thermo Fisher Scientific). Next, 10 µg of protein was mixed with 5x sample buffer containing 10% β-mercaptoethanol and boiled before electrophoretic separation on 4–15% TGX Stain-Free gels (Bio-Rad). The gels were then activated with ultraviolet light for 1 min to visualize total protein on the blotted membrane. Protein was transferred to PVDF membrane with a TransBlot Turbo Transfer System (Bio-Rad). Membranes were blocked with 5% milk and 1% BSA in a buffer consisting of 150 mM NaCl, 20 mM Tris-HCl, pH 7.5, and 0.1% Tween for 1 h. The blots were probed with antibodies against Anti-6X His-tag antibody [EPR20547] - ChIP grade (ab213204, 1000344-30, clone EPR20547; 1:500 dilution, Abcam), ADCY9/AC9 (ab191423, GR269903-5, clone EPR16188; 1:500 dilution, Abcam) or total OXPHOS rodent antibody cocktail (ab110413-MS604, lot 2101021674; 1:250 dilution, Abcam) and incubated overnight at 4 °C. Secondary antibodies were horseradish peroxidase-conjugated goat anti-rabbit (7074, lot 32; 1:10,000 dilution, Cell Signaling Technology) or goat anti-mouse (1706516, 1:2,000 dilution, Bio-Rad). Clarity Western ECL Substrate was used for visualization of proteins with a ChemiDoc XRS+ System (Bio-Rad). The signal intensity of each protein band was measured with Image Lab software (version 5.2.1; Bio-Rad) and normalized to the total amount of protein loaded in the lane.

### OCR measurements
Mitochondrial OCRs were evaluated with an XFe24 extracellular flux analyser (Agilent). Details of this analysis are described in Supplementary File 1.

### ATP:ADP ratio measurement in live single cells using PercevalHR
The ATP/ADP ratio was measured with the fluorescent biosensor PercevalHR. A total of 70,000 INS-1 beta cells were seeded on poly-D-lysine (1 mg ml$^{-1}$)-coated Lab-Tek chambered cover glass (Thermo Fisher Scientific), followed by transfection with Lipofectamine 3000 (Thermo Fisher Scientific) with 1 µg of PercevalHR plasmid DNA (Addgene ID: 21737) per well together with lentivirus transduction containing either the cDNA of *GFP* or *ONECUT2*. Seventy-two hours later, the cells were pre-incubated in experimental buffer (pH 7.4): 3.6 mM KCl, 1.3 mM CaCl$_2$, 0.5 mM MgSO$_4$, 0.5 mM Na$_2$HPO$_4$, 10 mM HEPES, 5 mM NaHCO$_3$ and 135 mM NaCl, supplemented with 2.8 mM glucose, for 90 min at 37 °C. Cover glass with adhered cells was mounted on the stage of a Nikon ECLIPSE Ti2 microscope (Bergman Labora) equipped with a confocal unit. PercevalHR was excited with laser light at 488 nm and emission was detected at 520 nm at 2.8 mM glucose and after addition of 16.7 mM glucose and 5 µM oligomycin sequentially.

### Gene expression microarray of *ADCY9*-deficient or *ONECUT2*-overexpressing islets
Total RNA was extracted from human islets after silencing *ADCY9* using siADCY9 (*n* = 5 donors) or *ONECUT2* lentivirus overexpression (OE-*ONECUT2*, *n* = 6 donors) using the miRNeasy isolation kit (QIAGEN) according to the manufacturer's recommendations. Details of the analyses of the microarray data are described in Supplementary File 2.

### Rat model
Diabetic GK rats[12], developed by selective breeding of Wistar rats, and control Wistar rats from Janvier labs in France were kept in standard controlled housing conditions; 21–22 °C, 55–65% humidity and 12-h light–dark cycle, and were given standard chow (SAFE A40) and water ad libitum. Animals were used at 11–13 weeks of age. Only male rats were included to reduce the number of animals used. Blood glucose levels were measured from the tail vein using a glucometer (ACCU-CHEK Aviva, Roche). Pancreatic islets were isolated by collagenase digestion and handpicked in cold Hank's buffer with 1 mg ml$^{-1}$ bovine serum albumin. Islets (75–200) were washed in ice-cold PBS, lysed in Qiazol (QIAGEN), transferred to microcentrifuge tubes, and homogenized by

vortexing tubes for 1 min before storage at −20 °C until RNA extraction using the MiRNeasy Mini Kit protocol (217004, QIAGEN). RNA concentration was measured using NanoDrop (ND-1000 Spectrophotometer).

### Rescue experiment in EndoC-βH1 beta cells exposed to HGP
Details are described in Supplementary File 3.

### Statistical methods and data collection
Clinical characteristics between individuals with pre-T2D/T2D and non-diabetic controls were analysed using two-sample, two-tailed *t*-tests. DMR calling in alpha versus beta cells was done using dmrseq[25] (details are found above) including donors with data available in both cell types. DMR calling in donors with pre-T2D/T2D versus controls in alpha and beta cells was done in a similar way; however, here the analyses were adjusted for age, sex, BMI and days in culture of islets before MARIS. DEGs in alpha versus beta cells were analysed in paired samples using DESeq2 (ref. 82). DEGs in donors with pre-T2D/T2D versus controls in alpha and beta cells were identified in a similar way, and here the analyses were adjusted for age, sex, BMI and days in culture of islets. FDR analysis was used to account for multiple testing, and FDR below 5% and 10% (*q* < 0.05 and *q* < 0.1) were applied. No statistical methods were used to pre-determine sample sizes, but they were based on results from our previous study where we analysed DNA methylation and gene expression in whole human islets[11]. T2D-associated differences in DNA methylation of individual CpGs were analysed using a non-parametric Wilcoxon rank-sum test. Student *t*-tests (paired and two-tailed) were used in silencing experiments comparing siRNAs of target genes with control (siNC cells), as well as the overexpression experiments in human islets and INS-1 beta cells. Student's *t*-tests (unpaired, two-tailed) were used for analyses of GK versus Wistar rats. Paired analysis of variance was used for analyses of the siONECUT2/HGP rescue experiment in EndoC-βH1 beta cells, with post hoc tests based on paired *t*-tests. Tests were two-tailed unless stated otherwise.

To show if overlapped sets display statistically significant enrichment, we used gene-set enrichment hypergeometric tests for gene overlaps and region-based permutation tests for genomic region overlaps. Details of these analyses are described in Supplementary File 4.

While the non-parametric tests used do not require normal distribution, parametric tests were used for normally distributed data. For EPIC v2.0 array data, *M* values were used for bioinformatic and statistical analyses. The collection and analysis of genome-wide data were performed blinded to the conditions of the experiment. Data collection and analysis for the remaining experiments were not performed blind to the conditions of the experiments. No data points were excluded from analysis unless they were technical outliers.

### Figure preparation
Graphs in Figs. 1–8 and Extended Data Figs. 1–8 were created using GraphPad Prism and arranged in CorelDraw version 24.3.05.571. Fig. 1a and Extended Data Fig. 6l were created in BioRender; Ofori, J. https://biorender.com/h6zumch and https://biorender.com/1w54ey4 (2026). Euler diagrams for Figs. 2k, 5j and 6l were created using eulerr[90]. Fig. 8e was generated using Cytoscape (v.3.10.3).

### Reporting summary
Further information on research design is available in the Nature Portfolio Reporting Summary linked to this article.

## Data availability
The WGBS and RNA-seq data generated from human alpha and beta cells are deposited in Zenodo (https://doi.org/10.5281/zenodo.18656853)[91], and in the LUDC repository (https://www.ludc.lu.se/resources/repository; accession nos. LUDC2025.08.1 for the alpha cell WGBS data, LUDC2025.08.3 for the beta cell WGBS data, LUDC2025.08.4 for the alpha cell RNA-seq data and LUDC2025.08.5

for the beta cell RNA-seq data). Data can be requested through the repository portal and individual-level data from the human pancreatic islets are not publicly available due to ethical and legal restrictions related to the Swedish Biobanks in Medical Care Act, the Personal Data Act and European Union's General Data Protection Regulation and Data Protection Act. The human alpha and beta cell RNA-seq data used for validation are available in the Gene Expression Omnibus under accession number GSE67543. The gene expression microarray data from *ADCY9*-deficient or *ONECUT2*-overexpressing human islets are available in the Gene Expression Omnibus under accession numbers GSE319105 and GSE319109, respectively). alpha-beta-methylome, an open resource based on the WGBS and RNA-seq data included in this study, is available at https://alpha-beta-methylome.serve.scilifelab.se/app/alpha-beta-methylome/, and allows users to explore and visualize the impact of age, sex and pre-T2D/T2D on autosomal DNA methylation and/or expression of any gene in alpha and beta cells, as well as differences between alpha and beta cells. Source data are provided with this paper.

## Code availability

The computer code used to generate the results described in 'WGBS and data analysis' and 'Statistical methods and data collection' is deposited in Zenodo (https://doi.org/10.5281/zenodo.18681937)[92]. Other details are available from the corresponding author on on request.

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

## Acknowledgements

We thank the Nordic Network for Clinical Islet Transplantation (JDRF award 31-2008-413) for providing material for the study; the Human Tissue Laboratory at LUDC; T. Dayeh for experimental support; O. Korsgren, Uppsala University for support with the islet isolation infrastructure; C. Wollheim, University of Geneva for valuable discussions; Lund University Cell and Gene Therapy core facility for the support in virus production; and the Bioinformatics and Expression Analysis (supported by the Board of Research at the Karolinska Institute and the Research Committee at Karolinska Hospital) for microarray analysis. WGBS was performed by the SNP&SEQ Technology Platform in Uppsala, which is part of the National Genomics Infrastructure Sweden and Science for Life Laboratory, supported by the Swedish Research Council and the Knut and Alice Wallenberg Foundation. Research grants supporting the study were obtained from the Kungliga Fysiografiska Sällskapet i Lund (the Birgit and Hellmuth Hertz Foundation, to J.K.O.), Novo Nordisk foundation (to C.L.), the Swedish Foundation for Strategic Research (IRC15-0067, to C.L. and L.E.), the Swedish Research Council (2016-02486, 2018-02567, 2019-01406, 2014-2775, 2018-02435, 2021-00628 and 2024-03597, to C.L.), the Region Skåne (Avtal om Läkarutbildning och Forskning (ALF), to C.L.), Strategic Research Area Exodiab (2009-1039, to C.L. and L.E.), the European Research Council (ERC-Paintbox, to C.L.), the Diabetes Foundation (to C.L. and L.E.), as well as Syskonen Svensson, Magnus Bergvall, Hjelt (to S.R.), Åke Wiberg, Påhlsson Foundations, Sigurd och Elsa Goljes memory Foundation (to S.R.) and Lars Hierta's memory Foundation (to S.R.). The flow cytometry study was performed at the LUDC-Flow Cytometry Core Facility, supported by the Swedish Research Council, Strategic Research Area Exodiab (2009-1039), by the Swedish Foundation for Strategic Research (IRC15-0067), by Region Skåne/(ALF) and by the Infrastructure Grant of Lund University, 2018. This work was also supported by the SciLifeLab & Wallenberg Data Driven Life Science Program (KAW2024.0159, to C.L.).

## Author contributions

C.L. initiated the project. J.K.O., S.R., A.L., A.P., M.M., A.K., L.S., K.K., Å.N., M.D.-P., A.W., K.B. and C.L. performed experiments and/or analyses. J.J., T.R., B.K., M.R. and L.E. designed experiments and experimental tools, and interpreted data. J.K.O., K.B. and C.L. drafted the manuscript. J.K.O. performed functional work in human islets, rat islets and INS-1 beta cells. S.R. performed all epigenetic editing experiments and contributed to writing the manuscript. A.L. developed the web tool and pipeline for EPIC v2.0 data and performed bioinformatic analyses. A.P. developed bioinformatic pipelines and performed bioinformatic analyses. All authors designed some experiments and/or analyses of data. All authors read and edited the manuscript.

## Funding

## Competing interests

The authors declare no competing interests.

## Additional information

**Extended data** is available for this paper at https://doi.org/10.1038/s42255-026-01498-9.

**Correspondence and requests for materials** should be addressed to Charlotte Ling.

[1]Epigenetics and Diabetes Unit, Department of Clinical Sciences in Malmö, Lund University, Scania University Hospital, Malmö, Sweden. [2]SciLifeLab, Lund University, Lund, Sweden. [3]Center for Evolutionary Hologenomics, Globe Institute, University of Copenhagen, Copenhagen, Denmark. [4]Exodiab, Lund University Diabetes Centre, Lund University, Malmö, Sweden. [5]Department of Advanced Medical and Surgical Sciences, University of Campania 'Luigi Vanvitelli', Naples, Italy. [6]Institute of Pediatric Rare Diseases, Florida State University, Tallahassee, FL, USA. [7]University of Groningen, University Medical Center Groningen, Groningen, the Netherlands. [8]Islet Cell Exocytosis Unit, Department of Clinical Sciences in Malmö, Lund University Diabetes Centre, Lund University, Malmö, Sweden. [9]These authors contributed equally: Jones K. Ofori, Sabrina Ruhrmann. [10]These authors jointly supervised this work: Karl Bacos, Charlotte Ling. ✉e-mail: charlotte.ling@med.lu.se

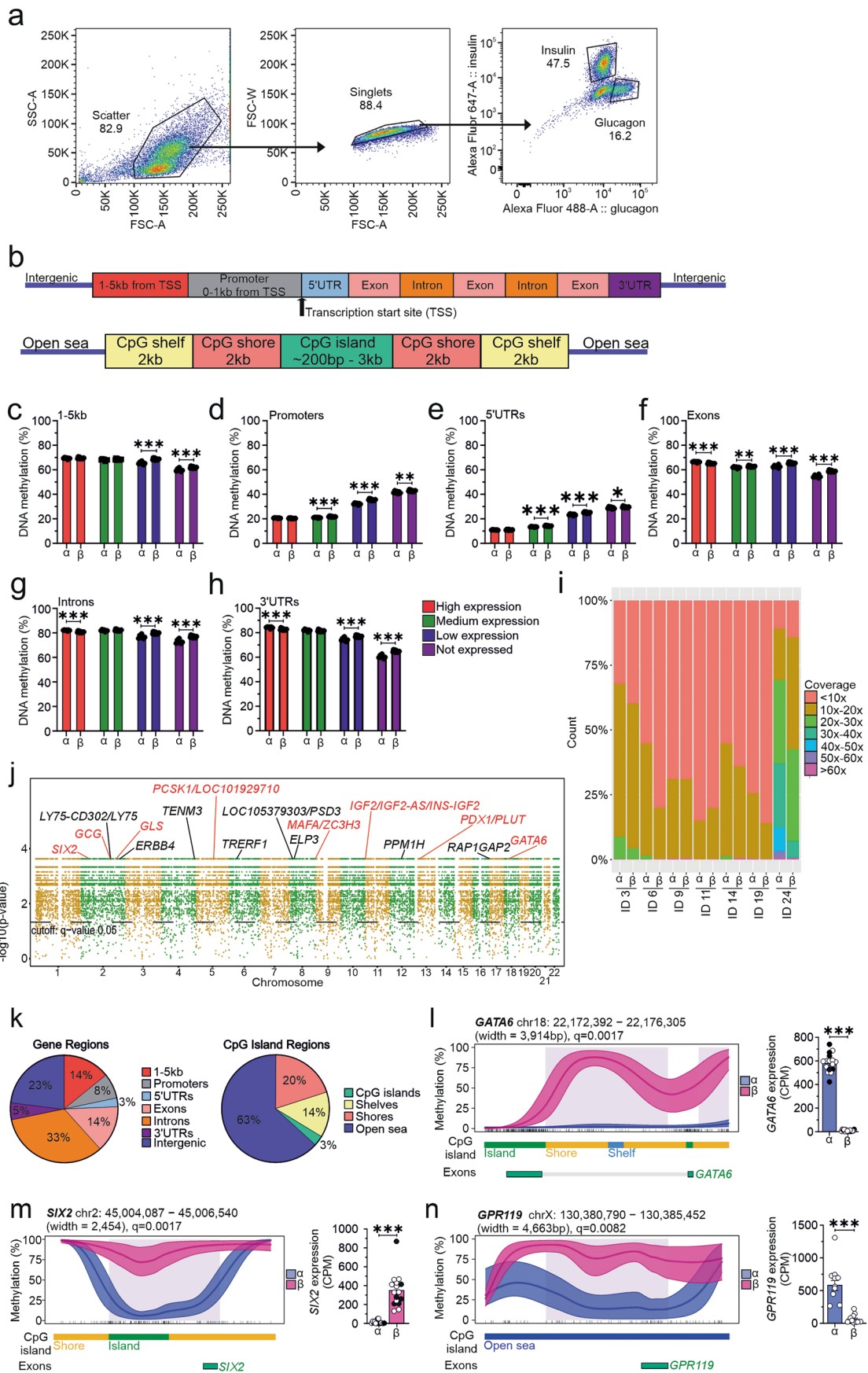

**Extended Data Fig. 1 | See next page for caption.**

**Extended Data Fig. 1 | Characterization of the DNA methylome in sorted human α- and β-cells data related to Figs. 1 and 2. a**, Gating strategy used in sorting α- and β-cells from human islets after staining with antibodies against glucagon and insulin. **b**, CpG-sites are mapped to Gene regions based on functional genome distribution, where the promoter is defined as the region between the transcription start site (TSS) and 1 kb upstream, and to CpG island regions based on CpG content. A CpG island is defined as a ≥ 200 bp stretch of DNA with a CG content of ≥50% and an observed CpG/expected CpG in excess of 0.6. Shores are 2,000 bp regions flanking CpG islands. Shelves are 2,000 bp regions outside the shores. The Open Sea is everything outside the shelves. **c-h**, Average DNA methylation levels in α- and β-cells in different gene regions; 1-5 kb from TSS, p = 8.36E-07 (low), p = 3.23E-04 (not) (**c**), Promoters, p = 2.72E-04 (medium), p = 9.73E-09 (low), p = 0.0036 (not) (**d**), 5'untranslated regions (UTRs), p = 3.83E-04 (medium), p = 3.41E-05 (low), p = 0.040 (not) (**e**), Exons, p = 5.82E-05 (high), p = 0.008 (medium), p = 4.20E-06 (low), p = 1.11E-08 (not) (**f**), Introns, p = 3.83E-05 (high), p = 2.10E-05 (low), p = 2.41E-06 (not) (**g**), and 3'UTRs, p = 2.17E-04 (high), p = 1.25E-04 (low), p = 1.08E-07 (not) (**h**) of not expressed genes and expressed genes divided into low-, medium-, and high-expressed in α- and β-cells (n = 7 (α-cells) and 11 (β-cells) donors), *p < 0.05, **p < 0.01, ***p < 0.001 based on two-tailed Student's t-tests. Data are presented as mean ± SEM. **i**, Coverage for the whole genome bisulfite sequencing (WGBS) data used for analysis of DMRs between sorted α- and β-cells from the same human islets (n = 7 donors). **j**, Manhattan plot displaying the 22,544 DMRs between α- and β-cells at a false discovery rate (FDR) less than 5% (q < 0.05 based on dmrseq[25]). **k**, Proportion of identified DMRs annotated to different gene and CpG island regions. **l, m**, Left; α- versus β-cell DMRs annotated to *GATA6* (**l**) and *SIX2* (**m**) in sorted α- and β-cell (n = 7 donors, dmrseq[25]). Right; expression of respective gene in sorted α- versus β-cells (n = 16, ***q < 0.001, based on DESeq2[82], presented as mean ± SEM, see Supplementary Table 6 for exact P values). Women and men are represented by filled and open circles, respectively. **n**, The top X-chromosome DMR (only males, n = 5) annotated to *GPR119* (left) and *GPR119* expression (right) in sorted α- and β-cell (n = 10 males, ***q = 2.93E-35 based on DESeq2[82], presented as mean ± SEM). The DMRs are analyzed and visualized by dmrseq[25]. In DMR plots presented in **l-n**, locations of the methylation loci are shown with black lines at the bottom, and average, smoothed lines based on the dmrseq analysis are shown in pink (β-cells) or blue (α-cells), including standard deviation. The shaded area shows the DMR. Annotations to CpG island regions and genes are shown below each graph. The p-/q-values for **j** and **l-n** were corrected for multiple testing.

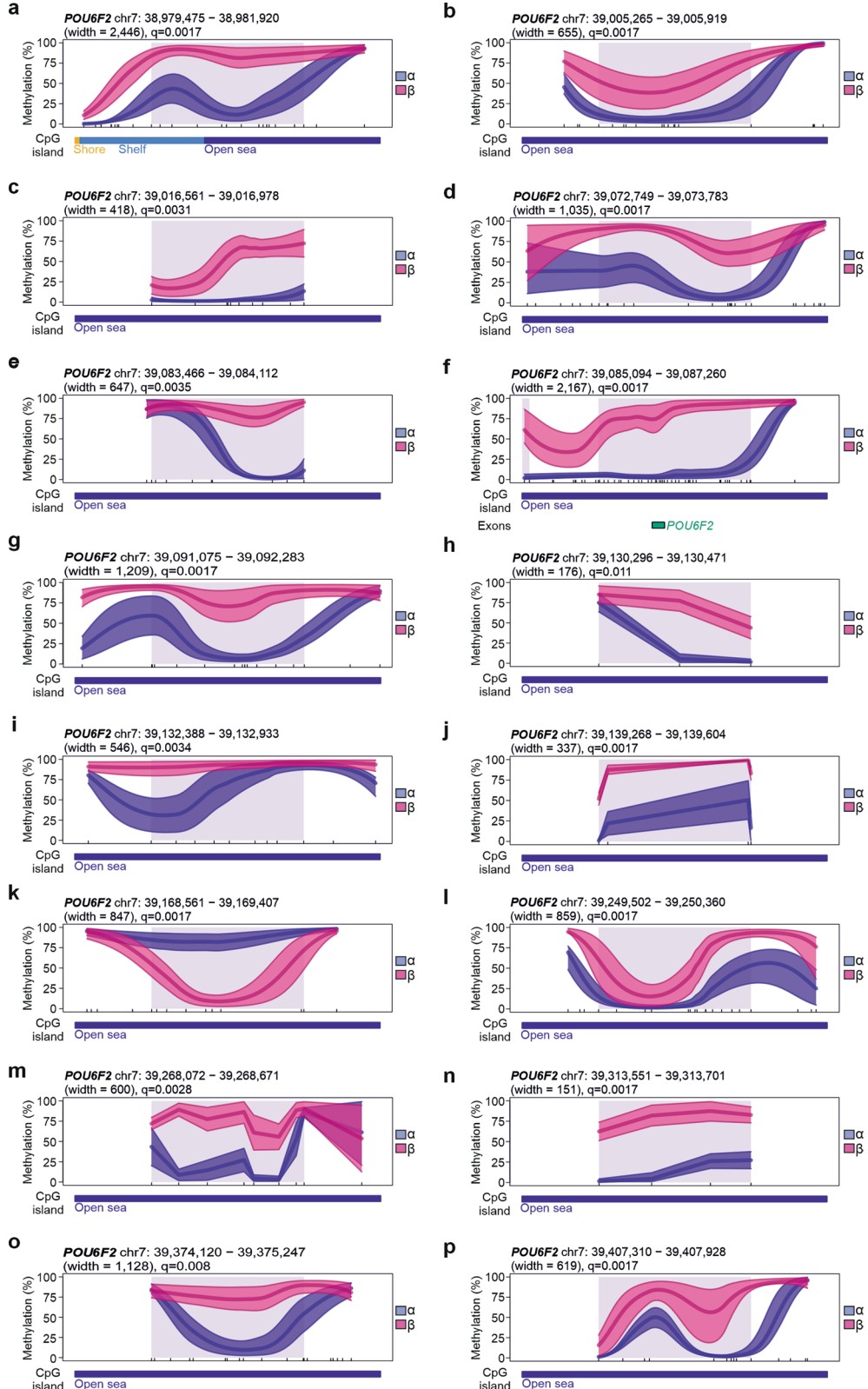

**Extended Data Fig. 2 | α- versus β-cell DMRs in *POU6F2* data related to Fig. 2. a-p**, 16 significant DMRs annotated to *POU6F2* identified in sorted α- versus β-cells (n = 7 donors) at a false discovery rate (FDR) less than 5% (q < 0.05, corrected for multiple testing). In all the DMR plots: locations of methylation loci are shown with black lines at the bottom and average, smoothed lines are shown in pink (β-cells) or blue (α-cells), including standard deviation. The shaded area shows the DMR. Annotations to CpG island regions and exons, if applicable, are shown below each graph. The DMRs were analyzed and visualized by dmrseq[25].

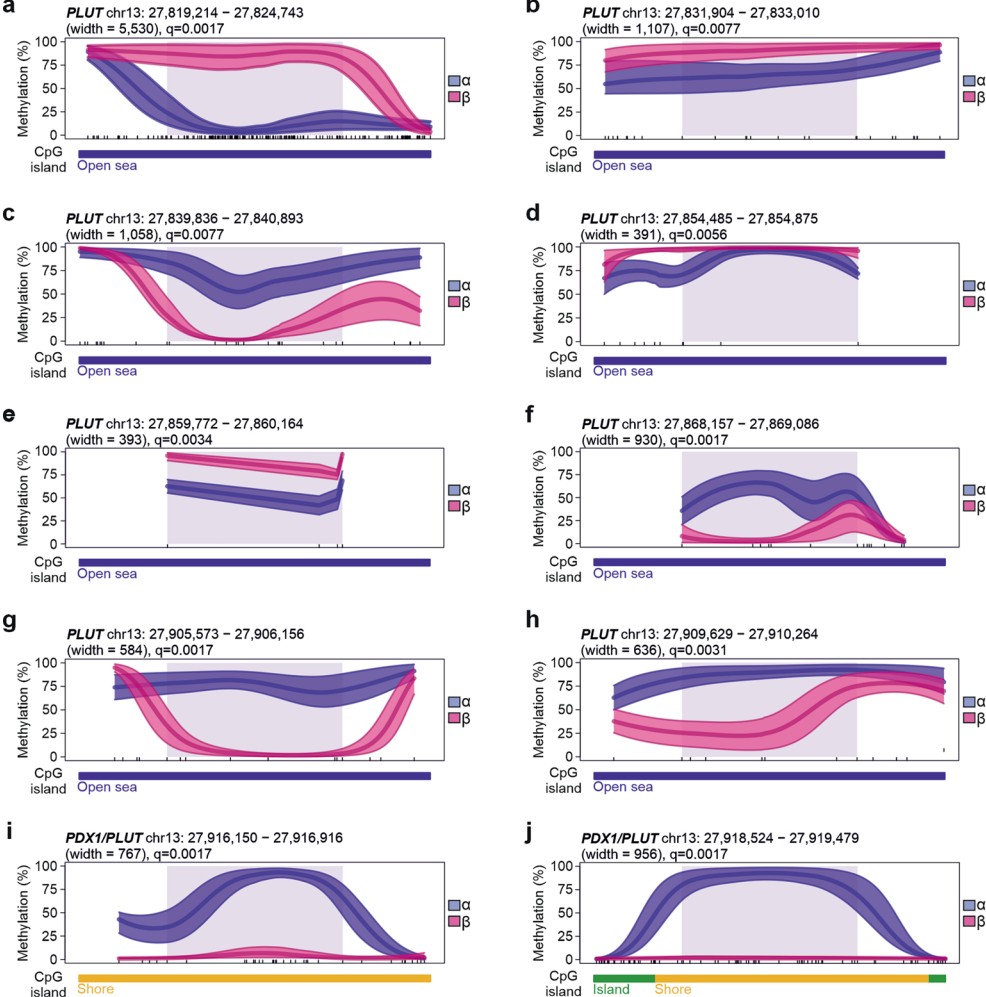

**Extended Data Fig. 3 | α- versus β-cell DMRs in *PDX1/PLUT*, data related to Fig. 2. a-j**, Significant DMRs annotated to *PDX1/PLUT* identified in sorted α- versus β-cells (n = 7 donors) at a false discovery rate (FDR) less than 5% (q < 0.05, corrected for multiple testing). In all the DMR plots: locations of methylation loci are shown with black lines at the bottom and average, smoothed lines are shown in pink (β-cells) or blue (α-cells), including standard deviation. The shaded area shows the DMR. Annotations to CpG island regions are shown below each graph. The DMRs were analyzed and visualized by dmrseq[25].

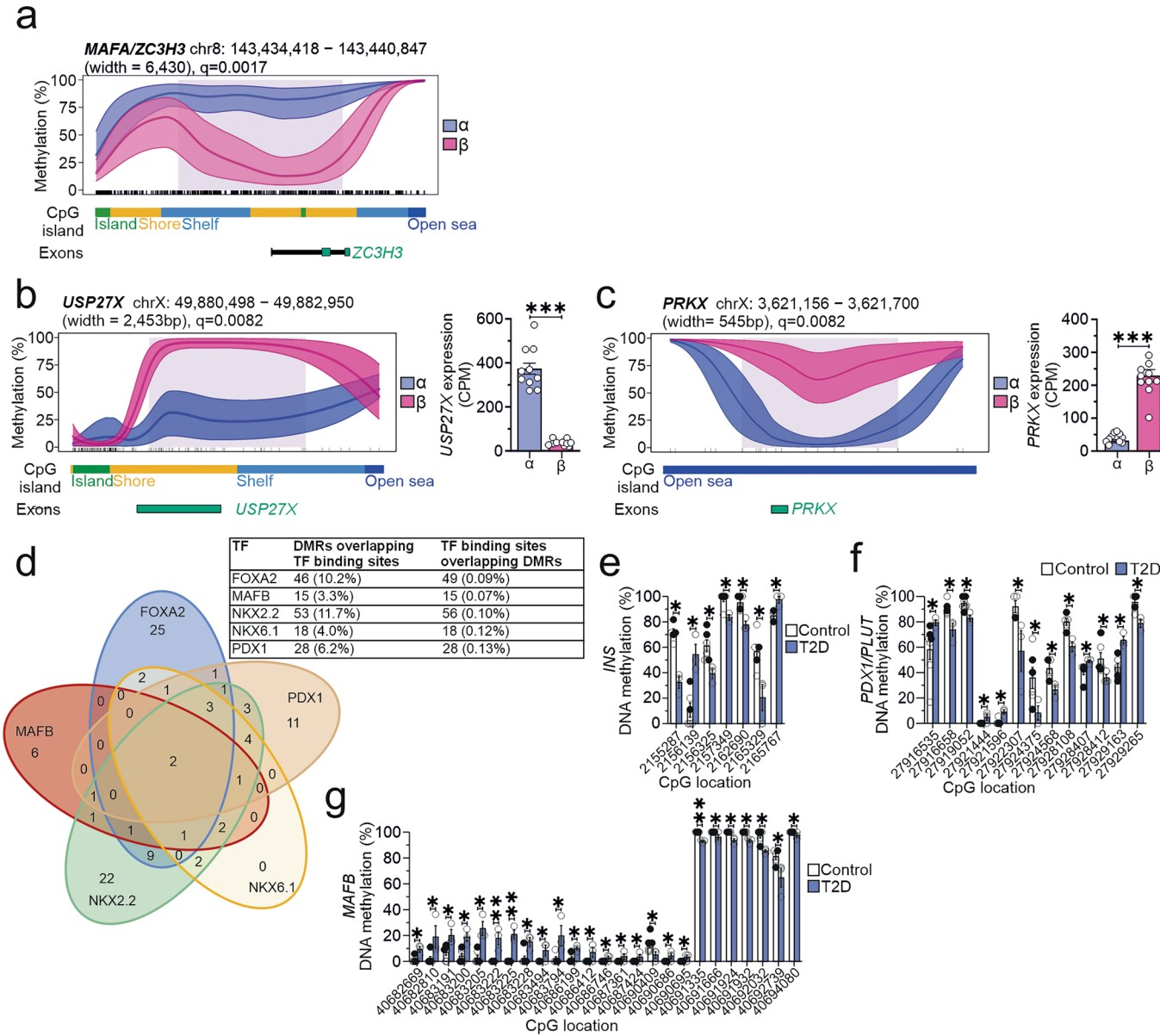

**Extended Data Fig. 4 | DMRs, TF binding and DNA methylation of individual CpG sites in α- and β-cells, data related to Figs. 2 and 5. a**, The longest α- versus β-cell DMR with most CpG sites is annotated to *MAFA/ZC3H3*. **b**, **c**, Some of the most significant α- versus β-cell DMRs on the X chromosome in males (n = 5 donors) are annotated to *USP27X* (**b**) and *PRKX* (**c**). Expression of respective gene in sorted α- and β-cells is shown on the right (n = 10 donors, data presented as mean ± SEM, ***q < 0.001, based on DESeq2[82]). In DMR plots presented in **a-c**, locations of the methylation loci are shown with black lines at the bottom, and average, smoothed lines based on the dmrseq analysis are shown in pink (β-cells) or blue (α-cells), including standard deviation. The shaded area shows the DMR. Annotations to CpG island regions and genes are shown below each graph. The p-/q-values for **a-c** were corrected for multiple testing. **d**, A Venn

diagram and table showing DMRs in α- versus β-cells on the X chromosome in males overlapping with binding sites for five islet specific transcription factors: FOXA2, MAFB, NKX2.2, NKX6.1 and PDX1 (ref. 24). Overlaps are enriched with p ≤ 0.001 based on region-based permutation tests. For individual p-values, see Supplementary Table 4D. **e-g**, pre-T2D/T2D-associated differences in the absolute DNA methylation level (%) of individual CpG sites annotated to *INS* (**e**), *PDX1/PLUT* (**f**), and *MAFB* (**g**) in α-cells. *p < 0.05, **p < 0.01 based on two-tailed Wilcoxon rank-sum test in pre-T2D/T2D (n = 3) versus non-diabetic controls (n = 7), and data are presented as mean ± SEM. For exact p-values, see Source data file for Extended Data Fig. 4. Women and men are represented by filled and open circles, respectively.

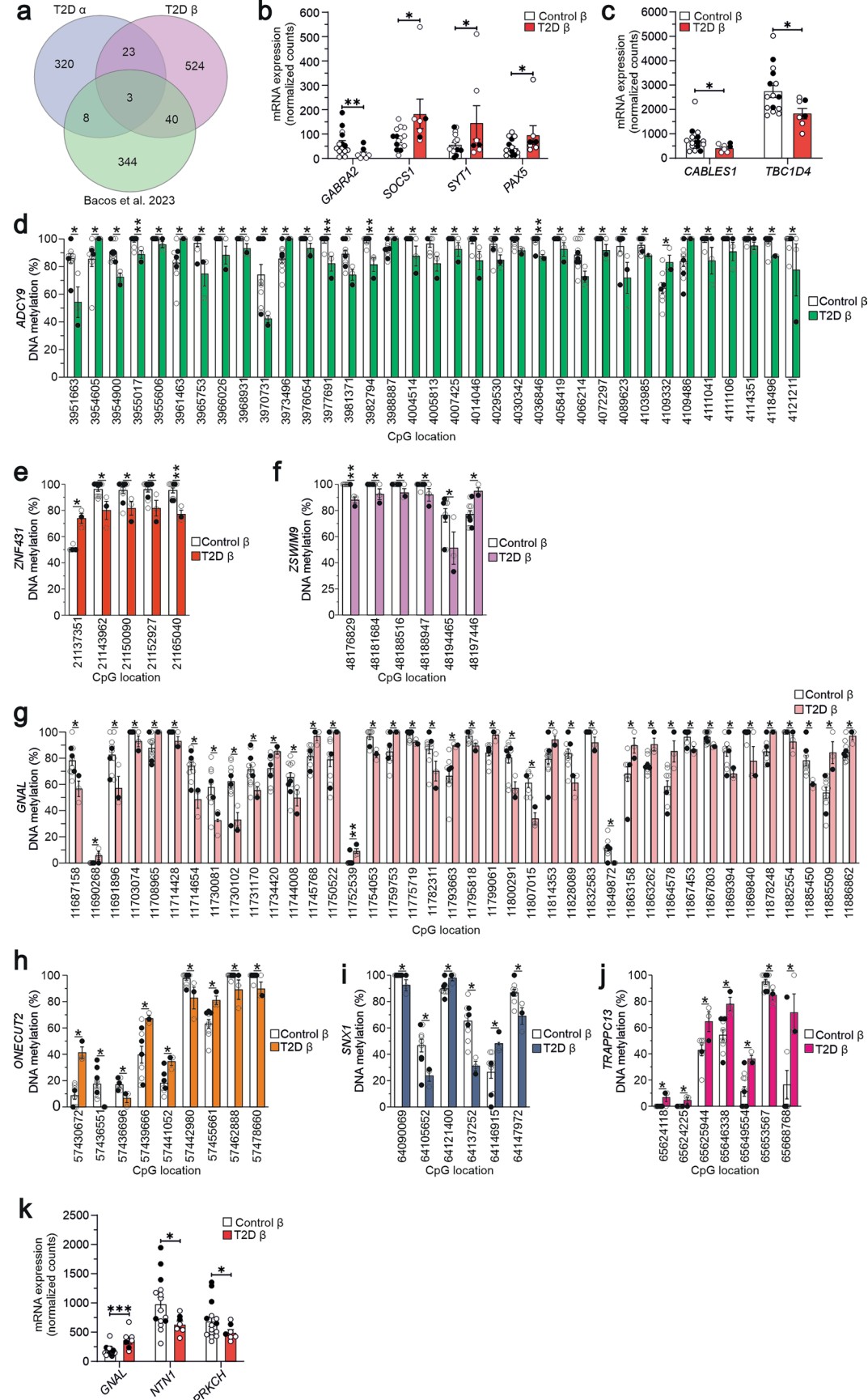

**Extended Data Fig. 5 | See next page for caption.**

**Extended Data Fig. 5 | Epigenetic regulation of T2D candidates, data related to Fig. 7. a**, Venn diagram showing T2D-associated differentially expressed genes (DEGs) in human α- and β-cells overlapping with DEGs identified in whole human islets by Bacos et al.[26] **b**, **c**, Examples of T2D-associated DEGs found in β-cells (n = 15 non-diabetic donors and 7 donors with T2D) that overlap with T2D-associated DEGs found in human islets by Bacos et al.[26]: *GABRA2*, *SOCS1*, *SYT1*, *PAX5*, *CABLES1* and *TBC1D4* (*p < 0.05, **p < 0.01 based on DESeq2[82]). **d-j** pre-T2D/T2D-associated differences in the absolute DNA methylation level (%) of individual CpG sites annotated to *ADCY9* (**d**), *ZNF431* (**e**), *ZSWIM9* (**f**), *GNAL* (**g**), *ONECUT2* (**h**), *SNX1* (**i**), and *TRAPPC13* (**j**), respectively in β-cells. *p < 0.05, **p < 0.01 pre-T2D/T2D (n = 3 donors) versus non-diabetic controls (n = 11 donors) based on two-tailed Wilcoxon rank-sum test. **k**, *GNAL, NTN1*, and *PRKCH* are T2D-associated DEGs in both sorted α- and β-cells, as well as in whole islets from Bacos et al.[26]. Data shown is for β-cells from pre-T2D/T2D (n = 7 donors) versus non-diabetic controls (n = 15 donors), *p < 0.05, ***p < 0.001. In **b-k**, women and men are represented by filled and open circles, respectively. Data are presented as mean ± SEM. For exact p-values, see **Supp**. Table 10 (panels **b**, **c**, and **k**) and Source data files (panels **d-j**).

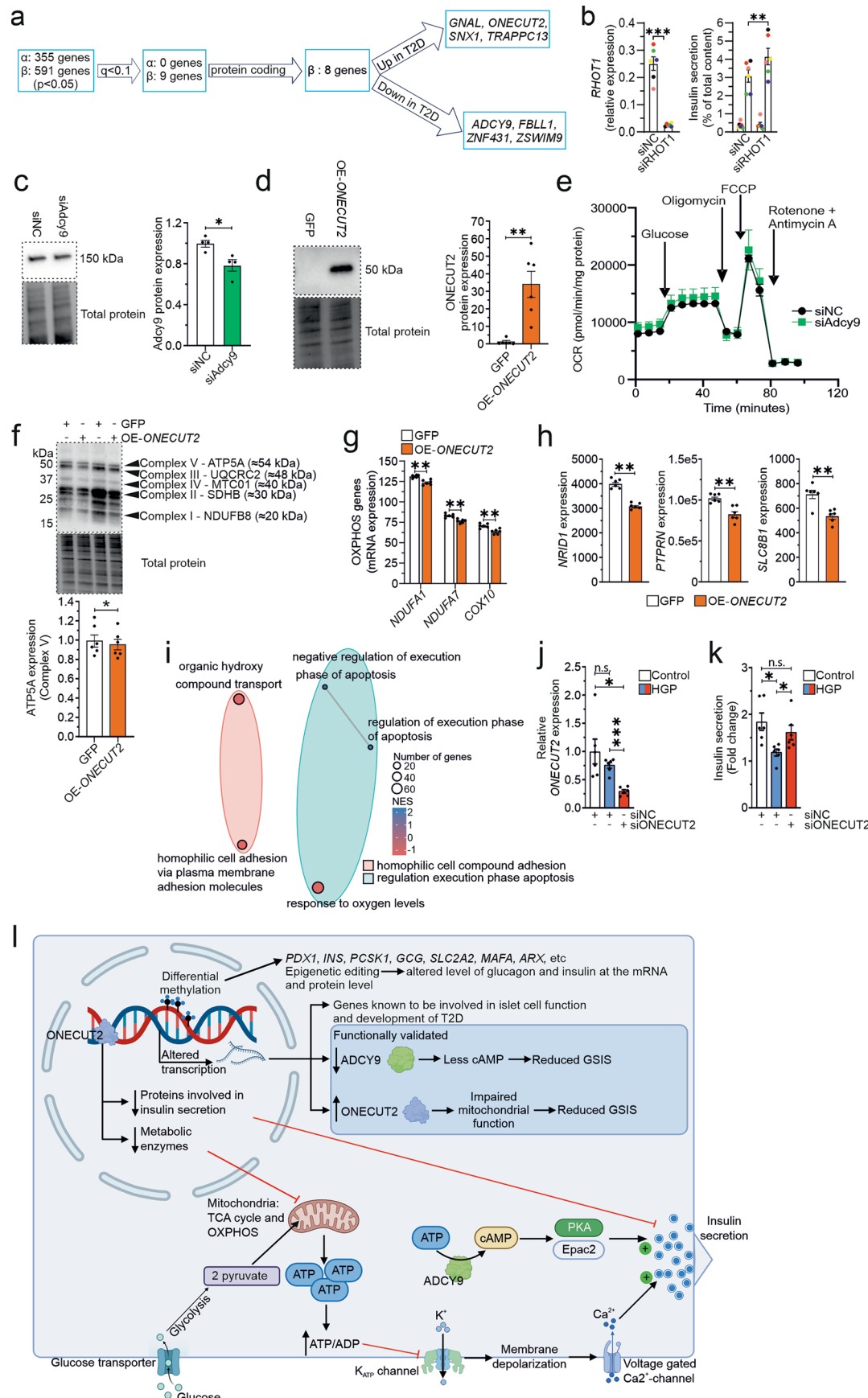

**Extended Data Fig. 6 | See next page for caption.**

**Extended Data Fig. 6 | Selection of T2D candidates, data related to Figs. 7 and 8. a**, Candidate gene selection for functional follow-up from RNA-seq data in sorted α- and β-cells from donors with pre-T2D/T2D versus controls. Flowchart shows the differentially expressed genes (DEGs) in pre-T2D/T2D versus non-diabetic controls in α- and β-cells, respectively, and selection of eight DEGs in β-cells for follow-up. **b**, Gene manipulation in EndoC-βH1 β-cells often gives different phenotypes compared to human islets and INS-1 β-cells. Figure shows that *RHOT1* knockdown (left, ***p = 0.0002) resulted in increased glucose-stimulated insulin secretion (right, **p = 0.0041) in EndoC-βH1 β-cells, which should be compared to reduced insulin secretion seen in human islets and INS-1 β-cells treated similarly (Figs. 4f and 5b in previous publication[12]). n = 6 biological replicates and paired t-tests. **c**, Reduced ADCY9 protein levels in INS-1 β-cells after siRNA transfection, n = 4 biological replicates, *p = 0.037 versus negative control (siNC) based on one-tailed paired Student's t-test. **d**, Increased ONECUT2 protein levels in INS-1 β-cells after lentiviral transduction, n = 6 biological replicates, **p = 0.0079 versus GFP (control) based on paired Student's t-test. **e**, No significant effect on oxygen consumption rate in siAdcy9 INS-1 β-cells, versus siNC, n = 4 biological replicates, p > 0.05 based on paired Student's t-test. **f**, Reduced ATP5A protein levels in OE-*ONECUT2* INS-1 β-cells, n = 6 biological replicates, *p = 0.049 versus GFP based on paired Student's t-test. **g**, Microarray showed decreased expression of *NDUFA1* (**p = 0.0022), *NDUFA7*

(**p = 0.0022), and *COX10* (**p = 0.0043) in OE-*ONECUT2* islets (n = 6 biological replicates/donors) based on Mann-Whitney U-tests, presenting unlogged data. **h**, Expression of *NRID1* (**p = 0.0022), *PTPRN* (**p = 0.0022), and *SLC8B1* (**p = 0.0087), contributing to the enrichment of the down-regulated gene set 'regulation of insulin secretion' in GSEA of OE-*ONECUT2* human islets versus GFP (Mann-Whitney U-tests, presenting unlogged data). n = 6 biological replicates/donors. **i**, emapplot visualizing GSEA results of transcriptome data in human islets transfected with siADCY9 versus siNC. n = 5 biological replicates and GSEA was corrected for multiple testing. For full data, see Supplementary Table 11. **j**, *ONECUT2* expression, and **k**, insulin secretion, presented as fold change (secretion at 16.7 mM glucose divided by secretion at 2.8 mM glucose), when silencing ONECUT2 using siRNA (siONECUT2) versus siNC in EndoC-βH1 β-cells exposed to high glucose (19 mM glucose) plus palmitate (1 mM palmitate, HGP) or control media for 72 h. n = 6 biological replicates, data analyzed by one-way ANOVA (**j**, p = 0.0293, and **k**, p = 0.0108) with paired Student's t-tests as post hoc. **j**, *p = 0.023: control+siNC versus HGP+siONECUT2, ***p = 4.38E-05: HGP+siNC versus HGP+siONECUT2, and **k**, *p = 0.017: control+siNC versus HGP+siNC, *p = 0.049: HGP+siNC versus HGP+siONECUT2. n.s.: Not significant. **l**, Cartoon summarizing key results from this study; created in BioRender; Ofori, J. https://BioRender.com/1w54ey4 (2026). Data in **b-h** and **i-j** are presented as mean ± SEM. Tests were two-tailed.

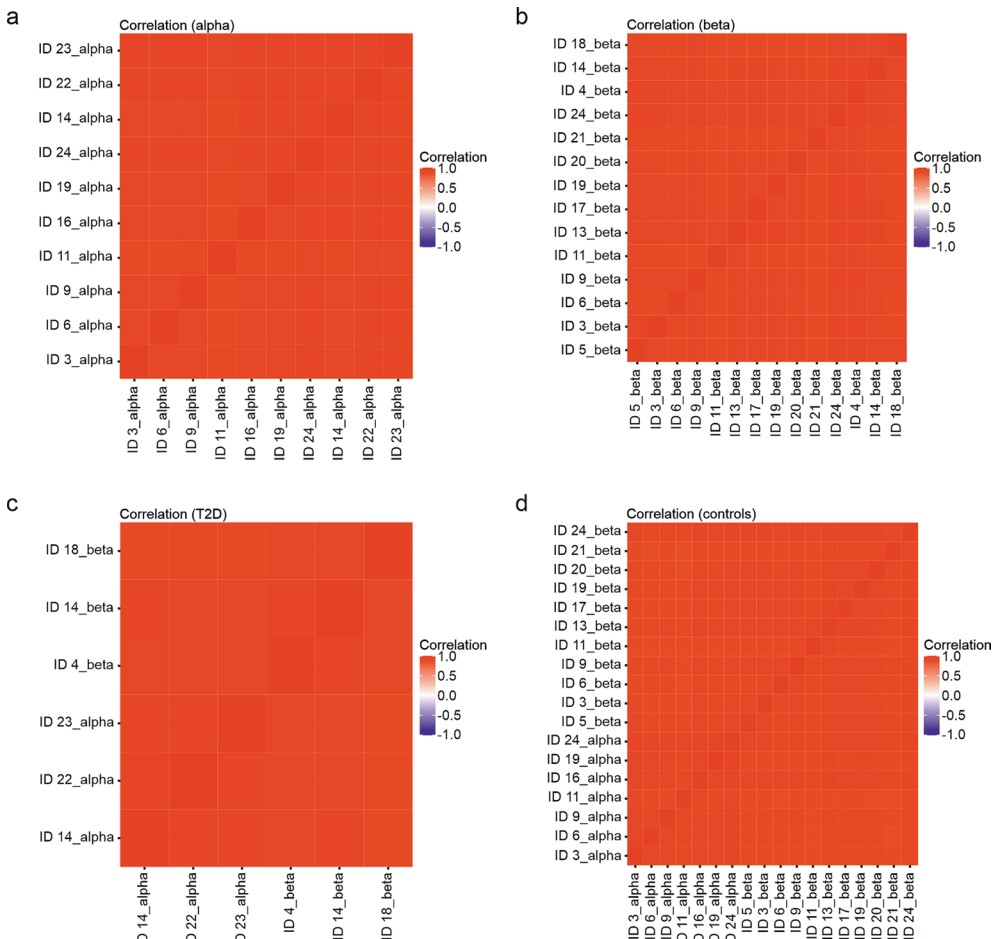

**Extended Data Fig. 7 | Heatmaps of the WGBS samples, data related to Figs. 2, 5 and 6.** Heatmaps showing correlations between all **a**, α-cell samples (n = 10 donors), **b**, β-cell samples (n = 14 donors), **c**, α- and β-cells samples from donors with T2D (n = 6 donors), and **d**, α- and β-cell samples from the non-diabetic controls (n = 18 donors) for the WGBS data.

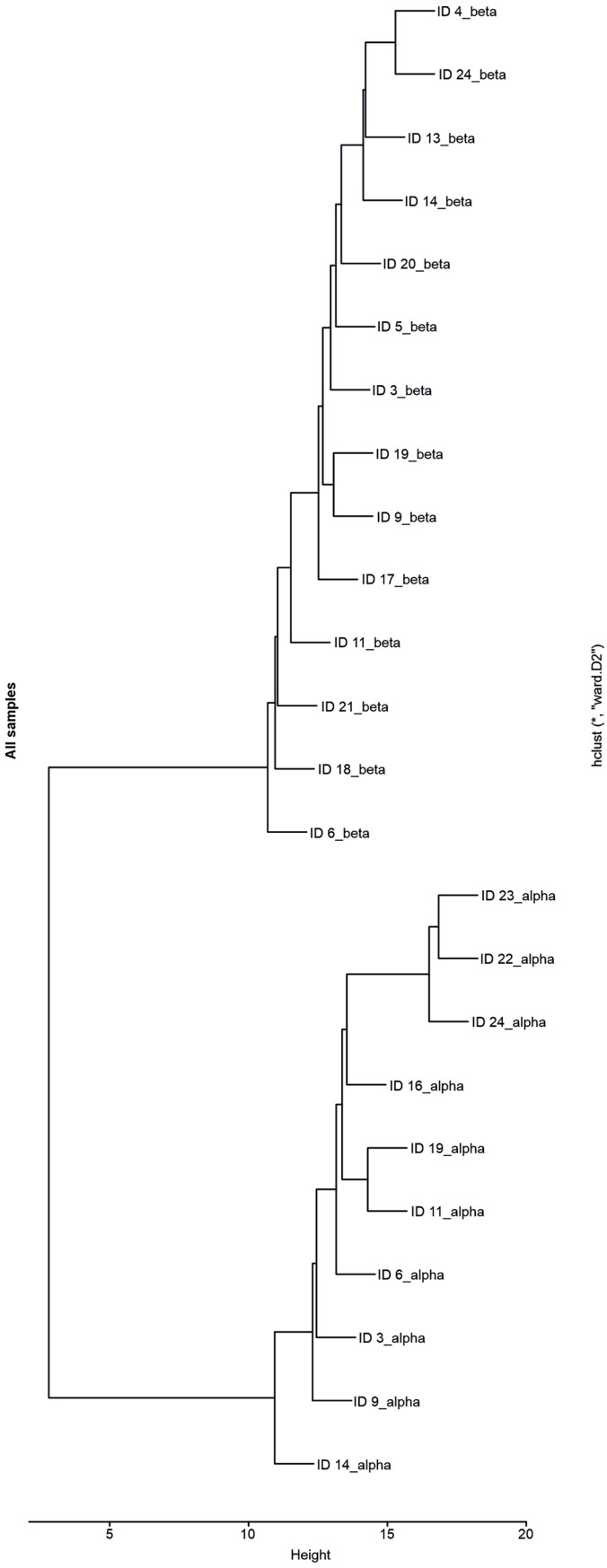

**Extended Data Fig. 8 | Dendrogram of the WGBS samples, data related to Figs. 2, 5 and 6.** Dendrogram including all α- (n = 10 donors) and β-cell (n = 14 donors) samples from the WGBS.

# Reporting Summary

## Statistics

For all statistical analyses, confirm that the following items are present in the figure legend, table legend, main text, or Methods section.

| n/a | Confirmed | |
|---|---|---|
| ☐ | ☒ | The exact sample size (*n*) for each experimental group/condition, given as a discrete number and unit of measurement |
| ☐ | ☒ | A statement on whether measurements were taken from distinct samples or whether the same sample was measured repeatedly |
| ☐ | ☒ | The statistical test(s) used AND whether they are one- or two-sided<br>*Only common tests should be described solely by name; describe more complex techniques in the Methods section.* |
| ☐ | ☒ | A description of all covariates tested |
| ☐ | ☒ | A description of any assumptions or corrections, such as tests of normality and adjustment for multiple comparisons |
| ☐ | ☒ | A full description of the statistical parameters including central tendency (e.g. means) or other basic estimates (e.g. regression coefficient) AND variation (e.g. standard deviation) or associated estimates of uncertainty (e.g. confidence intervals) |
| ☐ | ☒ | For null hypothesis testing, the test statistic (e.g. *F*, *t*, *r*) with confidence intervals, effect sizes, degrees of freedom and *P* value noted<br>*Give P values as exact values whenever suitable.* |
| ☒ | ☐ | For Bayesian analysis, information on the choice of priors and Markov chain Monte Carlo settings |
| ☐ | ☒ | For hierarchical and complex designs, identification of the appropriate level for tests and full reporting of outcomes |
| ☐ | ☒ | Estimates of effect sizes (e.g. Cohen's *d*, Pearson's *r*), indicating how they were calculated |

*Our web collection on statistics for biologists contains articles on many of the points above.*

## Software and code

Policy information about availability of computer code

| Data collection | Illumina NextSeq 500: NextSeq Control Software v. 4.0.1<br>NovaSeq 6000: NovaSeq Control Software v1.7.5 |
|---|---|

| Data analysis | WGBS: Real Time Analysis v3.3, Sequencing Analysis Viewer v2.4.7, R (v4.1.3), bioconductor-annotatr (v1.20.0), bioconductor-txdb.hsapiens.ucsc.hg38.knowngene (v3.14.0), bioconductor-org.hs.eg.db (v3.14.0), bioconductor-bsseq (v1.30.0), bioconductor-dmrseq (v1.14.0), trim-galore (v0.6.6), fastqc (v0.11.9), multiqc (v1.12), cutadapt (v3.2), bowtie2 (v2.4.2), samtools (v1.12), bismark (v0.23.0), bamtools (v2.5.1)<br>EPIC arrays: R (v.4.6.0), Minfi (v.1.57.0), wateRmelon (v2.17.0)<br>RNA-seq: R (v4.2.2) with bioconductor-genomicfeatures (v1.50.22), bioconductor-tximport (v1.26.0), bioconductor-deseq2 (v1.38.0), bioconductor-genomeinfodb (v1.34.8), trim-galore (v0.6.7), cutadapt (v3.4), salmon (v1.5.2), fastqc (v0.11.9), multiqc (v1.11)<br><br>Overlap between DMRs and other data: HOMER (v.4.11, http://homer.ucsd.edu/homer/motif/), R (v.4.4.1),Bioconductor GenomicRanges (v1.54.1 and v.1.58.0), R package VennDiagram (v.1.7.3). annotatr (v.1.32.0), regioneR (v.1.38), BSgenome.Hsapiens.UCSC.hg38.masked (v.1.4.5), phyper (v.4.4)<br><br>Expression arrays: Transcriptome Analysis Console (v4.0), R packages clusterProfiler (v.4.10.0) and enrichplot (v.1.22.0), R WGCNA package (v. 1.73), Cytoscape (v.3.10.3)<br>Functional experiments: GraphPad Prism (v10)<br>The computer code used to generate the results described in the methods section under DNA methylation analysis and Statistical methods is deposited in Zenodo (https://doi.org/10.5281/zenodo.18681936). Other details are available from the corresponding author upon on request. |
|---|---|

For manuscripts utilizing custom algorithms or software that are central to the research but not yet described in published literature, software must be made available to editors and reviewers. We strongly encourage code deposition in a community repository (e.g. GitHub). See the Nature Portfolio guidelines for submitting code & software for further information.

# Data

Policy information about availability of data

All manuscripts must include a data availability statement. This statement should provide the following information, where applicable:
- Accession codes, unique identifiers, or web links for publicly available datasets
- A description of any restrictions on data availability
- For clinical datasets or third party data, please ensure that the statement adheres to our policy

The WGBS and RNA-seq data generated from human α- and β-cells were deposited in Zenodo (https://doi.org/10.5281/zenodo.18656853) and the LUDC repository (https://www.ludc.lu.se/resources/repository, accession numbers: LUDC2025.08.1 for the α-cell WGBS data, LUDC2025.08.3 for the β-cell WGBS data, LUDC2025.08.4 for the α-cell RNA-seq data, and LUDC2025.08.5 for the β-cell RNA-seq data). Data can be requested through the repository portal and individual-level data from the human pancreatic islets are not publicly available due to ethical and legal restrictions related to the Swedish Biobanks in Medical Care Act, the Personal Data Act and European Union's General Data Protection Regulation and Data Protection Act. Source data are provided with this paper. The human α- and β-cell RNA-seq data used for validation is available in Gene Expression Omnibus GEO Accession viewer (GSE67543).

The gene expression microarray data from ADCY9-deficient or ONECUT2-overexpressing human islets are available in Gene Expression Omnibus GEO Accession viewer (accession numbers GSE319105 and GSE319109, respectively).

Data reproducibility: We have used several different methods/models to test the reproducibility of our data. For example, the global RNA-sequencing data was replicated via comparisons with published data (PMID 25931473 and 36656641), while the ONECUT2 overexpression in human beta-cells from people with T2D was replicated in islets from a rat model of diabetes. For the DNA methylation data in human alpha- and beta-cells, we could validate sex-associations based on our previous study in whole human islets (PMID 25517766), and age-associations based on our previous paper in whole islets (PMID 27029739), and T2D-associations based on our previous paper in whole human islets (PMID 38086799). Moreover, the global methylation patterns in the sorted alpha- and beta-cells cells are similar to the pattern in whole islets (PMID 28052964). For the functional analyses, we focus mainly on ADCY9 and ONECUT2. Manipulation of these two genes resulted in similar phenotypes regarding glucose-stimulated insulin secretion in both human islets and clonal beta-cells.

# Research involving human participants, their data, or biological material

Policy information about studies with human participants or human data. See also policy information about sex, gender (identity/presentation), and sexual orientation and race, ethnicity and racism.

| Reporting on sex and gender | The findings of this project are based on inclusion of both sexes, determined by self-reporting and concordant genetic testing.<br>Sex is considered as a co-variate in genome-wide analyses of pre-T2D/T2D-associations in human alpha- and beta-cells.<br>The number of males and females in all analyses is presented in Supplementary Table 1. Briefly, the LUDC Islet Sorting Cohort consists of islets from 17 men and 7 women. |
|---|---|
| Reporting on race, ethnicity, or other socially relevant groupings | All pancreatic islets are from donors from The Nordic Network for Clinical Islet Transplantation Program (www.nordicislets.org) from Scandinavia, a defined geographic region. Clustering of existing GWAS data from this population suggests that a small number of donors are of non-Scandinavian descent. There is no reporting on any other socially relevant groupings. |
| Population characteristics | Regarding the LUDC Islet Sorting Cohort: 24 donors with an age of 64.0 (36-81) years, BMI of 25.8 (18.9-32.7) kg/m2, and HbA1c of 38.0 (22-49) mmol/mol.<br>The ethical permits do not allow sharing individual level clinical data in public domains. |
| Recruitment | Donors of pancreatic islets were from the Scandinavian Transplantation Unit and included multi-organ human donors. Islets were included in this study when not used for transplantation due to clinical reasons. There may be a bias in who is willing to transplant organs for research. However, it is impossible to dissect how such bias would impact our results. |

| Ethics oversight | All procedures regarding the human pancreatic islets were approved by the Swedish Ethical Review Authority (permit numbers 2007-05 and 2011-263). The studies followed the Helsinki Declaration and written informed consent was obtained from pancreatic islet donors or their relatives |
|---|---|

Note that full information on the approval of the study protocol must also be provided in the manuscript.

# Field-specific reporting

Please select the one below that is the best fit for your research. If you are not sure, read the appropriate sections before making your selection.

☒ Life sciences        ☐ Behavioural & social sciences        ☐ Ecological, evolutionary & environmental sciences

For a reference copy of the document with all sections, see nature.com/documents/nr-reporting-summary-flat.pdf

# Life sciences study design

All studies must disclose on these points even when the disclosure is negative.

| Sample size | Sample size was based on our previous studies where we have analyzed DNA methylation and gene expression in human islets (PMID: 28052964), and performed gene knockdown and/or overexpression experiments (e.g. 36656641 and 38086799). |
|---|---|
| Data exclusions | WGBS data was excluded based on low coverage as this makes the data unreliable |
| Replication | The global RNA-sequencing data was replicated via comparisons with published data (PMID 25931473 and 36656641), while the ONECUT2 overexpression was also replicated in a rat model of diabetes.<br>For the generation of methylation data in whole human islets we have previously used several methods; EpiTYPER, WGBS, methylation arrays, and pyrosequencing. The data produced by the thee methods have always yielded data that correlate very strongly (see for example PMID 38086799 and 28052964). Moreover, the global methylation patterns in the sorted cells is very similar to the pattern in whole islets (PMID 28052964).<br>For the functional analyses, we focus mainly on ADCY9 and ONECUT2. Manipulation of these two genes resulted in similar phenotypes regarding glucose-stimulated insulin secretion in both human islets and clonal beta-cells. |
| Randomization | Islet samples were allocated into experimental groups (control, pre-T2D, or T2D) by the donor having a T2D diagnosis or not, or by their HbA1c.<br>Samples were allocated to sex based on self-reporting and concordant genetic testing. |
| Blinding | During the genome-wide DNA methylation and gene expression experiments the samples were anonymous to the technicians who ran the analyses and hence they did not know which group samples belonged to. However, during for example qPCR and Western blot analysis, the technician needed to know the order the samples were loaded on plates and gels and then blinding was not possible. |

# Reporting for specific materials, systems and methods

We require information from authors about some types of materials, experimental systems and methods used in many studies. Here, indicate whether each material, system or method listed is relevant to your study. If you are not sure if a list item applies to your research, read the appropriate section before selecting a response.

## Materials & experimental systems

| n/a | Involved in the study |
|---|---|
| ☐ | ☒ Antibodies |
| ☐ | ☒ Eukaryotic cell lines |
| ☒ | ☐ Palaeontology and archaeology |
| ☐ | ☒ Animals and other organisms |
| ☐ | ☒ Clinical data |
| ☒ | ☐ Dual use research of concern |
| ☒ | ☐ Plants |

## Methods

| n/a | Involved in the study |
|---|---|
| ☒ | ☐ ChIP-seq |
| ☐ | ☒ Flow cytometry |
| ☒ | ☐ MRI-based neuroimaging |

# Antibodies

| Antibodies used | Antibodies used for FACS: guinea pig anti-insulin (A0564, Lot 10119708, 1:600 dilution, Dako), mouse anti-glucagon (MAB1249, Lot HSQ0416111, clone number 181402, 1:200 dilution, R&D Systems), donkey anti-guinea pig Alexa-647 (706-605-148, Lot 129003, 1:400 dilution, Jackson Immunoresearch) and donkey anti-mouse Alexa-488 (A-21202, Lot 1644644, 1:400 dilution, Invitrogen).<br><br>Antibodies used for western blot: Anti-6X His tag (#ab213204, lot 1000344-30, clone number EPR20547, 1:500 dilution, Abcam), anti-ADCY9/AC9 (#ab191423, lot GR269903-5, clone number EPR16188, 1:500 dilution, Abcam), total OXPHOS rodent Ab cocktail (#ab110413-MS604, lot 2101021674, 1:250 dilution, Abcam), horseradish peroxidase (HRP)-conjugated goat anti-rabbit (#7074, lot 32, 1:10,000 dilution, Cell Signaling Technology), and goat anti-mouse (#1706516, 719 1:2,000 dilution, Bio-Rad). |
|---|---|

| Validation | All antibodies are commercially available and have been characterized by the manufacturers (and in several publications) for their reactivity in the appropriate species and for their compatibility to be used with the respective application. Conditions for blocking and antibody dilutions are stated in the method section. siRNA-silenced samples were used for validation of the anti-ADCY9 antibody (Supplementary Fig. 6c). For the overexpression experiments, we used GFP-transfected cells as validation (Supplementary Fig. 6d). For Western blot we also used molecular weight markers to identify the band(s) that migrated at the expected size of each respective protein analyzed. Validation details are also available on the manufacturers' websites. |
|---|---|

# Eukaryotic cell lines

Policy information about cell lines and Sex and Gender in Research

| Cell line source(s) | We used the rat INS-1 832/13 β-cell developed by Hohmeier, H.E. et al (PMID: 10868964) and the human EndoC-βH1 developed by Ravassard P. et al (PMID 21865645). INS-1 832/13 β-cells were developed from a male rat and EndoC-βH1 from a female fetus. |
|---|---|
| Authentication | It was verified that these cells are β-cells based on their ability to secrete insulin and respond to glucose with increased glucose-stimulated insulin secretion (only β-cells produce and secrete insulin). Additionally, the EPIC v.2.0 data from EndoC-βH1 also support β-cell origin as the degree of INS DNA methylation was low while GCG DNA methylation was high in promoter regions in cells not exposed to epigenetic editing. The cell lines were not authenticated using any additional authentication techniques. |
| Mycoplasma contamination | Cells were tested regularly for mycoplasma and were never contaminated. |
| Commonly misidentified lines (See ICLAC register) | No commonly misidentified cell lines were used in the study. |

# Animals and other research organisms

Policy information about studies involving animals; ARRIVE guidelines recommended for reporting animal research, and Sex and Gender in Research

| Laboratory animals | We used male GK rats, developed by selective breeding of Wistar rats (PMID: 32180184) and control Wistar rats from Janvier labs in France. Animals were kept in standard controlled housing conditions; 21–22 °C, 55–65% humidity and 12-h light/12-h dark cycle and given standard chow (SAFE® A40, SAFE, Rosenberg, Germany) and water ad libitum. The animals were used at 11-13 weeks of age. |
|---|---|
| Wild animals | No wild animals were used in the study. |
| Reporting on sex | Regarding animals, we only use one sex. The main reason is to reduce the number of animals included in the experiments according to 3R principle (Replacement, Reduction and Refinement). It is previously known that both male and female GK rats have reduced insulin secretion. Moreover, in all our human experiments both sexes were included. |
| Field-collected samples | No field collected samples were used in the study. |
| Ethics oversight | Animal experiments were performed with permission of the Animal Ethics Committee of Lund University (Permit number 5.8.18-04115/2021) in accordance with the legal requirements of the European Community (86/609/EEC). |

Note that full information on the approval of the study protocol must also be provided in the manuscript.

# Clinical data

Policy information about clinical studies
All manuscripts should comply with the ICMJE guidelines for publication of clinical research and a completed CONSORT checklist must be included with all submissions.

| Clinical trial registration | not relevant for this study |
|---|---|
| Study protocol | not relevant for this study |
| Data collection | not relevant for this study |
| Outcomes | not relevant for this study |

# Plants

| | |
|---|---|
| Seed stocks | not relevant for this study |
| Novel plant genotypes | not relevant for this study |
| Authentication | not relevant for this study |

# Flow Cytometry

## Plots

Confirm that:

☒ The axis labels state the marker and fluorochrome used (e.g. CD4-FITC).

☒ The axis scales are clearly visible. Include numbers along axes only for bottom left plot of group (a 'group' is an analysis of identical markers).

☒ All plots are contour plots with outliers or pseudocolor plots.

☒ A numerical value for number of cells or percentage (with statistics) is provided.

## Methodology

| | |
|---|---|
| Sample preparation | Samples were prepared according to the protocol in PMID:24594682, with minor changes:<br>Islets were centrifuge at 150 rcf(g) for 2 min at RT before being washed 3 times with PBS. Islet were then dispersed by addition of 5ml of TrypLE (#12604013, ThermoFisher Scientific) and incubattion in a water bath at 37° for 6-8 min (until the solution turns into a milky suspension). Cells were pipetted up and down a few times to break up cell clumps and complete media was then added. Cells were allowed to rest for 5 minutes before centrifugation (300g for 10min). Cells were then resuspended in media and passed through a 40μm cell strainer before being washed twice in PBC (300g for 5min).<br>Work surfaces, centrifuge, micropipettors, pipet-aid, and gloves were cleaned with RNaseZap. Cells were resuspended in fixation buffer (4% PFA, 0.1% saponin, and 2%RNase inhibitor, in PBS) and incubated under gentle agitation for 30min at 4 degrees. Cells were then centrifuged 3min at 3000g at 4°. before being washed twice in wash buffer (0.2% BSA, 0.1% saponin, 1% RNase inhibitor, in PBS). After the second wash, cells were resuspended in staining solution (1% BSA, 0.1% saponin, 5% RNase inhibitor, in PBS) with guinea pig anti-insulin (Dako, A0564, 1:600) and mouse anti-glucagon (R&D Systems, MAB1249, 1:200) and put on rotator at 4 degrees for 30 minutes. Cells were then washed twice in wash buffer before resuspension in staining buffer with donkey anti-guinea pig Alexa-647 (Jackson Immunoresearch, 706-605-148, 1:400) and donkey anti-mouse Alexa-488 (Invitrogen, 1:400). Tubes were wrapped in aluminum foil and put on rotator at 4 degrees for 30 minutes. Cells were then washed twice in wash buffer and resuspended in sorting buffer (0.5% BSA and 5% RNase inhibitor in PBS) and straining into a FACS tube with strainer cap before sorting. |
| Instrument | Aria Fusion (BD Biosciences, San Jose, CA, USA) |
| Software | FlowJo |
| Cell population abundance | We sorted 322,627±73,055 α-cells (purity 92.0±7.6%) and 311,550±61,250 β-cells (purity 96.5%±4.0) from islet preparations of 24 donors with or without T2D (cell number and purity from FloJo). The purity of the samples was confirmed by methylation and expression of GCG and INS, the main markers of α- and β-cells, respectively, in the WGBS and RNA-seq data. |
| Gating strategy | The gating strategy is displayed in Supplementary Fig 1a. |

☒ Tick this box to confirm that a figure exemplifying the gating strategy is provided in the Supplementary Information.

