## [Peer Review File · Nature Metabolism]

Cell-specific DNA methylation in human α - and β -cells regulates gene expression in type 2 diabetes

Corresponding Author: Professor Charlotte Ling

Version 0:

Reviewer comments:

Reviewer #1

(Remarks to the Author)

This manuscript explores CpG methylation throughout the genomes of alpha and beta cells in non-diabetic and T2D or pre-T2D donors. They use their dataset to identify a few critical genes that have DMRs and DEGs in diabetes and go on to test gain- and loss-of-function models of these genes in human islets or beta-cell lines. The authors further have made a public database for their data; however, although I would have liked to explore its functionality, it is not available yet. This manuscript adds a novel perspective that enriches our understanding and complements existing research in the field. Nonetheless, there are still areas that require clarification or further development to meet the standards for publication.

Major points

1. The donor characteristics table is insufficient. De-identified data should be provided for each donor, including the duration of diabetes and treatment of those with diabetes, if available. For example, see the Human Islet Checklist in the journal Diabetes. To improve clarity, the specific assays performed for each donor should be included alongside this information. Currently, the use of aggregation makes it difficult to determine which assays were conducted on the same donor. For instance, it would have been impossible to recognize from these tables that only two nondiabetic females had WGBS data available in both alpha and beta cells, as described in lines 177-178.
2. If possible, please compare the data with the GWAS/3C data in PMID: 36070683 (also sorted alpha and beta cells) to identify any overlaps in T2D-associated loci.
3. In Figure 4c-e, are the colors showing paired samples the same samples throughout the figure? For example, are the red cells in c the same cells as in d? If so, why is there an n=4 for 4c-d but only an n=3 for 4E?
4. The authors attribute the dysfunction observed in OE-ONECUT2 islets solely to mitochondrial dysfunction; however, GCK is downregulated in this model. Please discuss this gene/protein as a potential contributor to the dysfunction observed upstream of the K-ATP channel.
5. I applaud the clear labeling for cell types used in Figure 7. However, it highlights the use of a rat cell line. Why were they used instead of the EndoC cells used in other parts of the manuscript? To be clear, I'm not asking for these experiments to be performed in another cell line; however, if there was difficulty in using the EndoC cells for this purpose, it should be reported.
6. Please discuss why it makes sense that changes in adenylyl cyclase activity would result in changes downstream of the K-ATP channel.
7. ONECUT1 and ONECUT2 have nearly identical binding sites, and ONECUT1 (also known as HNF6) has established roles in beta-cell development and function and is linked to T2D and MODY. What are the relative expression levels between ONECUT1 and ONECUT2 in beta cells? They don't seem to be able to compensate for each other. Please discuss the known roles of ONECUT1 in light of your discovery.

Minor points

1. In Supplementary Tables 3 and 7G, what does "Previously reported" refer to? For those sites that have not been reported before, does this mean your data shows a DMR, but the gene it's associated with has not been identified as a diabetes-related gene?
2. For Figure 3g-o (and maybe for other similar data), would it be possible to color-code the individual data based on the sex of the donor?
3. Line 210—States supplementary Data 3 and 6. Does this refer to Tables or Figures?
4. It's somewhat circular to argue that ONECUT2 is upregulated in both your overexpression model and in T2D beta cells, since you tested what would happen in normal islets when you upregulate ONECUT2 (Lines 380-382). Please explain why this gene is upregulated in both models within the text. Something like "In addition to ONECUT2, these genes were upregulated in both the ONECUT2 and T2D islets: . . ."
5. MODY is in parentheses but not written out to explain what it stands for (line 440).

6. Other labs have had problems using siRNA in islets without the use of a virus or a light dissociation with accutase or trypsin. Was this a new protocol worked out for islets? Was it based on any other cell type?

Reviewer #2

(Remarks to the Author)

In this study, Ofori and colleagues established the DNA methylomes and transcriptomes of purified human alpha and beta cells from several donors, with or without T2D. These results allow the systematic analysis of genetic/epigenetic changes that accompany the development of T2D. In combination with T2D risk genes identified through GWAS and subsequent functional analysis, these data will likely provide a valuable resource for understanding the etiology of T2D development. There are several main issues.

1. The number of duplicates is low, especially for the T2D cells. Granted, the materials needed are rare, and the DNA methylation process is demanding. To compensate, the authors should provide some analysis to inspect the degree of variation between individual samples (especially for the three T2D samples). Revealing such variations will allow readers to judge whether some of the data repeatability (or the lack of it) issues arise from donor variation or lab technique differences.
2. Similar to the above, the authors should utilize some of the recently published gene expression data in alpha/beta cells (by others) to investigate the repeatability of gene expression in human donor islet cells. Particularly intriguing would be exploring how repeatable the T2D-caused gene expression changes from different research groups are (e.g., comparing the authors' new findings with those reported in PMC11374460). The findings should be instructive for readers if similar studies were to be conducted to explore T2D etiology (at the molecular level).'
3. The statistical analyses need clarification. When presenting gene list overlaps, the authors need to perform hypergeometric analyses to show enrichment or anti-enrichment and p values. Similarly, the authors need to justify when different standards of cutoffs were used for gene selection: on lines 115-116, the authors set >2 mean normalized count for gene expression cutoff; on line 183, it was ≥ 20 . Similarly, FDR was set at <5% in some places but 10% in others – did the authors get different conclusions when staying with a same FDR?
4. The authors did not provide a correlation between higher or lower methylation levels in DMRs with gene down- or up-regulation in alpha or beta cells. Why is this so. This reviewer believes that establishing such a correlation is necessary because the findings will suggest an activating/repressive role of methylation in gene expression.

Minor suggestions:

1. This reviewer commends the functional studies of several candidate genes. As a resource, it would be ideal if the authors could establish the causal relations between DNA methylation and transcription in a few more genes (not only INS as presented).
2. The authors may consider a few more rescue experiments using human T2D islets.
3. The presentation needs some clarification e.g.:
 - 3.1. The first paragraph in the result section, the statement from lines 91-92 suggests that the authors obtained a total of ~300,000 cells from 24 samples. But it's actually ~300,000 cells from each of the 24 donors. In addition, the number of controls/T2D samples are missing from the text, which could have been very helpful (without the need to go to figure 1 and the sup. Tables).
 - 3.2. Lines 152-154: does the sentence mean that each of the 5 TFs bind to 415 DMRs of 415 DMRs have at least one motif of the 5 TFs?
 - 3.3. Lines 161-162: it is not clear what the sentence means.
 - 3.4. What is the implication of data presented in the paragraph from lines 283-289, when islet data were integrated with that of pure cells?
 - 3.5. If kept in the text, the authors need to give more detailed instructions on data access on the webpage described from lines 308-314.
4. Some comments on the levels of gene over-expression on data interpretation are needed (Fig. 7)
5. Some discussion on the repeatability of human islet/beta-cell related data will be helpful

Reviewer #3

(Remarks to the Author)

Ofori et al describe whole genome bisulfite sequencing data, as well as transcriptomes, for pancreatic alpha and beta cells sorted from people with and without type 2 diabetes. They used samples from 14 donors – not a huge number but certainly sufficient – and sequenced methylation at a relatively low depth of 10x.

They report thousands of differentially methylated regions that distinguish alpha from beta cells, as well as ND vs diabetic alpha and beta cells, and link some of these changes to altered gene expression (in the classic expected sense – inverse correlation between methylation and expression).

They also perform some manipulations – forced methylation of the insulin gene which causes reduced insulin expression, and silencing or overexpression of genes that shows differential expression in beta cells from T2D, in human islets and even in the GK rat model. This part (in particular the latter, involving knockdown or overexpression of genes based on the transcriptome data, and particularly the rat data) represents a large body of work with impressive results, but is only loosely connected to the overall theme of the paper on DNA methylation, which is unfortunate.

Overall, the resource generated in this study will be important for the community for many studies. I have several concerns and comments, which the authors should be able to address.

1. Quality of data. I could not find a clear statement regarding purity of the sorted cell populations. Evidence should be presented based on deconvolution of the methylomes to identify the fraction of non-beta (or non-alpha) cells, with support from transcriptome analysis. This should be detailed for each sample, with a particular attention to the possibility that small

contaminants are affecting the differences identified between ND and diabetic samples.

2. Along the same line of quality control – ND / diabetic samples should be reshuffled to ensure that that differences observed do not represent an overfit.

3. Novelty. While the dataset is expected to be useful for many in the field for multiple applications, there are no novel insights from the analysis so far, which is unfortunate. The authors should make efforts in this directions. Some loose suggestions below.

4. Figure 2h – interesting to see expression (though low level) and demethylation of SLC2A2 (Glut2), the transporter that is famously functioning as a glucose sensor in mouse beta cells but does not play a role in human beta cells. Please discuss this. It could potentially reflect a history of expression in neonate human beta cells, where its absence causes transient neonatal diabetes (Hattersley).

5. Figure 4 – forced methylation of the insulin gene in a beta cell line. This is a missed opportunity! Nobody will be surprised to hear that INS methylation reduces expression. Insights may come from comparing the effect of multiple gRNAs: which CpGs matter, and must be demethylated to allow for insulin expression? Cracking the code of which CpGs matter and how many of them matter would be an important addition to the study. At a minimum, profile methylation more broadly than with pyro sequencing, and show the methylation status of the entire locus after editing. How many CpGs were successfully targeted?

6. One important finding is that almost half of the genes that are differentially methylated in alpha vs beta cells are adjacent to T2D SNPs, while only 22% of the differentially expressed genes overlap with T2D genes. This deserves some discussion and interpretation.

7. Genetics vs environment: the authors report 18% of T2D candidate genes identified by GWAS are differentially methylated in T2D vs ND beta cells (Figure 6m). This is nice, and gives a minimum estimate for the extent of genetic effect on methylation (likely much broader due to effects on remote loci). Please draw the reverse chart: what percent of the beta or alpha cell T2D DMRs identified here are adjacent to a known GWAS T2D hit? This may give very important information about the potential impact of the environment on epigenetics in T2D, compared with the extreme hypothesis that all differential methylation in diabetes is dictated genetically. From looking at Supp Table 8 I think, but a, not sure, that there are 5000 T2D DMR in beta cells, and of these 1300 are adjacent to a locus genetically linked to T2D.

8. Figure 5b: the KEGG pathway names are meaningless (alpha cells have nothing to do with synapses or morphine addiction). I realize that this represents enrichment of islet genes that were assigned to these brain sets, but authors should find a better way to describe or at least explain the findings that will be comprehensible to most readers and will also provide some biological sense. The circadian link is potentially interesting and should be further developed – what are the genes? What is the expected phenotypic outcome?

9. Comparisons were made in the past between the epigenome, including the methylome, of alpha and beta cells. Please cite and discuss differences if any and the novelty here.

Version 1:

Reviewer comments:

Reviewer #1

(Remarks to the Author)

Major Points:

There are several issues with the website. I accessed the tool using Google Chrome, the most updated version, and encountered an error message when attempting to analyze alpha versus beta DMRs. Additionally, the color for age DMRs is not clearly defined.

It would be beneficial to align the chromosomal position numbers with the gene layout in the application for improved user comprehension.

Minor Points:

Line 540—It appears that "recourse" should be corrected to "resource."

Reviewer #2

(Remarks to the Author)

In this revised manuscript, Ofori and colleagues present an useful resource to look at the methlome/gene expression changes in alpha and beta cells from control and T2D subjects. This reviewer's main concern for the first version is the clarity of statistical analysis and the context of the presented studies against the field. In this revision, the authors have added a substantial amount of work to address these concerns. T No more main issue is left.

A few minor comments are (no need for re-review):

1) Throughout the manuscript, it appears that the authors use "expression" to refer to mRNA levels. Because sometimes the authors also examined protein levels (also considered "expression"), it might be better to replace "expression" with

“transcript level”, or define “expression” ahead of time.

2) The studies using dCas9-DNMT3a and dCas9-Tet1 for manipulating site-specific methylation are interesting and the technique is useful. The degree of forced methylation here appears lower than in some previous studies (e.g., PMC6327977, which methylates nearly all CpG dinucleotides close to the gRNA). This may merit a comment somewhere: the more physiological levels of methylation method described here, using viral vectors, might be more useful than the previous plasmid-delivery approach.

3) There might be a typo in Fig. 4b and 4h: is cg10781463 or cg10781483 the correct number?

Reviewer #3

(Remarks to the Author)

The authors have addressed my concerns and I support publication. Note that the alpha-beta-methylome website is not fully functional (e.g a comparison of alpha vs beta cells gives an error message) – please fix before publication, as I expect this resource to be popular. Congratulations!

**Point-by-point response to the comments by the Reviewers and Editors
regarding the manuscript:**

NATMETAB-RS250816494

“Cell-specific DNA methylation in human pancreatic α - and β -cells controls gene expression and impacts type 2 diabetes”

We want to thank the Editors and Reviewers for reviewing our manuscript. We are very happy with the positive response and the valuable comments that helped us improve our manuscript.

Importantly, we have strongly addressed all concerns of the reviewers, and we provide new evidence from additional experiments and statistical analyses, including:

i) New epigenetic editing experiments using 2 additional gRNAs targeting a DMR annotated to both *TH* and an *INS* enhancer, and a DMR in *GCG* (encoding glucagon) together with either CRISPR-dCAS9-DNMT3A or CRISPR-dCAS9-TET1. Following epigenetic editing, we analyzed methylation using both pyrosequencing and Illumina’s EPIC v2 array, covering larger regions. Notably, these novel editing experiments found several methylation sites that regulate expression of *INS*, *TH*, and *GCG* in β -cells.

ii) Rescue experiment, where silencing of *ONECUT2* in β -cells exposed to high glucose plus palmitate (HGP), to mimic a diabetogenic environment, rescued the HGP-induced perturbed insulin secretion.

iii) Addition of heatmaps and dendrograms showing variance between samples, hypergeometric analyses for the overlaps performed in our study, and correlations between the DMRs and DEGs.

iv) Further details for the web-tool <https://alpha-beta-methylome.serve.scilifelab.se/app/alpha-beta-methylome> in the result section and in the web-tool.

v) Overlaps between our data and 3D map data in PMID: 36070683, T2D-associated DEGs in PMID: 38049589 and PMID: 41145477, and we refer to the study by Manduchi *et al* (PMID: 40631278) and compare our results to their results and discuss similarities.

Please find our response to each comment below, where changes to the text in the manuscript are marked in red.

Response to the editors’ comments

1] In order for us to consider the revised manuscript, it will be important to maximize the resource value of your data by furthering the analysis and ensuring that the browsable database is freely accessible. All reviewers have made clear suggestions in this regard and we strongly encourage you to incorporate the suggestions from Reviewers 2 and 3.

Response: We appreciate the valuable comments by all three reviewers and the editors, which strongly helped us improve our manuscript. We thoroughly answered all the review comments, and performed all additional analyses suggested by the reviewers. Our goal was to have a freely accessible browsable database, and we apologize if there were problems with access to the browser/database during the review process. The reviewers should be able to access the browser by using the link included in the submitted manuscript. However, if the reviewers experience any limitations in access to the browser, we will investigate this further. Based on reviewer comments, we also included more detailed instructions on data

access on the webpage and in the result section of the manuscript.

2] In light of comments from all three reviewers, we encourage you to provide additional insights into the functional relevance and mechanistic importance of your findings, ideally using additional *in vitro* experimentation, or at minimum by furthering the analysis of your data as highlighted in point 1]. Addition of more *in vitro* experiments will not be essential for consideration of the revised manuscript.

Response: Based on valuable comments from the reviewers, we have performed additional *in vitro* epigenetic editing experiments and rescue experiments. Specifically, we: **i)** used Illumina's EPIC v2.0 arrays to analyze DNA methylation of additional CpG sites near the guide RNA (gRNA) target site in the *INS* promoter as well as in regions surrounding the *INS* gene, **ii)** performed epigenetic editing using one more gRNA overlapping a DMR in the *TH* locus and an *INS* enhancer, **iii)** performed epigenetic editing targeting a DMR in the promoter of *GCG*, encoding glucagon, and **iv)** performed successful rescue experiments. These new experiments and novel results are presented in detail in minor comment 1 and 2 from reviewer 2 and in comment 5 from reviewer 3.

3] As highlighted by all three reviewers, please integrate, cross-reference and analytically contextualize your findings with existing literature in the area (for example with PMID: 36070683, PMID: 38049589).

Response: Based on good suggestions, we are now comparing our data to the results presented in the two mentioned papers, as well as the preprint of the manuscript that we are co-submitting our manuscript with (see, major comment 2 from reviewer 1, comment 2 from reviewer 2, and comment 9 from reviewer 3).

4] In their confidential comments, two of the reviewers highlighted a preprint with a somewhat similar study and dataset (PMID: 40631278) and suggested that you compare your findings directly with those in the preprint, e.g. to identify differently methylated regions that were observed in both studies. This sort of independent replication would increase the confidence of the reported results of both studies and increase their combined impact on the field. We would encourage you to consider whether perhaps a co-submission of the revised manuscript would be feasible, in which case you might wish to reach out to the authors of PMID: 40631278 proactively. We have managed various co-submissions successfully in the past (see, for example, a related Editorial: <https://www.nature.com/articles/s42255-021-00496-3>). If you have any questions about the process do not hesitate to contact us.

Response: We value this good suggestion, and in our revised manuscript, we now refer to the preprint (PMID: 40631278), and we compare some of our results to their data (see page 6 and 20). Importantly, the preprint does not include a full list of the 78,389 α - versus β -cell DMRs identified, rather, they list names of genes with DMRs that are also enriched in specific biological pathways (n=706 unique genes). Additionally, the list does not contain chromosomal coordinates for the DMRs. We could therefore only compare our list of 7,975 genes with α - versus β -cell DMRs with these 706 genes. Nevertheless, and importantly, the overlap between our results and theirs is highly significant. We added this information to the revised manuscript.

Page 6: *"We then compared the list of 7,975 genes with α - versus β -cell DMRs identified in the current study with a list of genes presented in **Supplementary Table 9 and 10** of a preprint based on a similar study (accessed on 2025-10-28)³⁰. This showed that 468 out of 706 (66%) genes, including*

GCG, INS, MAFA, PDX1, in their study exhibit DMRs also in our study (hypergeometric enrichment score (HES)=2.07, p=2.63E-58)."

In the revised manuscript, we also discuss our data in relation to the data in the preprint, see:

Page 20: *"Interestingly, 75.5% of the T2D-associated β -cell DMRs identified in this study were hypomethylated, which is in line with our previous study where 77% of T2D-associated methylation sites had lower methylation in whole islets from donors with T2D¹², and a recent preprint supporting hypomethylation in β -cells from donors with T2D³⁰."*

Based on this valuable suggestion, we contacted the authors of the pre-print (PMID: 40631278). They were happy to co-submit their paper with ours, and we subsequently would like to co-submit our study with their study.

5] Please strengthen the statistical analysis, data presentation and methodological details in the manuscript, in line with comments from the reviewers.

Response: We appreciate this suggestion, and we addressed all such review comments in the revised manuscript. For example, we; **i)** added information of variance, e.g., heatmaps and dendrograms (see page 23), according to comment 1 by reviewer 2, **ii)** we performed region-based permutation and gene set enrichment hypergeometric analyses to show enrichment and p-values for overlaps, and we justified the different standards of cutoffs that were used for gene selection according to comment 3 by reviewer 2, **iii)** we had already performed permutations when generating DMRs (see page 23-24) but based on comment 2 by reviewer 3, we clarified this further in the revised manuscript, and **iv)** we added correlations between DMRs and DEGs according to comment 4 by reviewer 2 (see Page 8).

6] I would like to draw your attention to our trial of a stricter implementation of the Sex and Gender Equity in Research (SAGER) guidelines, which was recently announced (see our Editorial: <https://www.nature.com/articles/s42255-022-00581-1>). If the study is heavily skewed towards one sex, we ask that you indicate this in the abstract, and perhaps even in the title.

Response: Our data is not heavily skewed towards one sex, and we consider sex throughout our study. For example, we included comparison between men and women in our publicly available browsable database, and we analyze the sex chromosomes separately in men and women.

7] Finally, we strongly encourage you to pay special attention to our formatting guidelines, which will be required should the manuscript be accepted for publication at a later stage. As a minimum, please prepare your figures in editable formats, present data as scatter plots, state the number of samples in the figure legends, provide source data for each figure, and provide uncropped scans of all blots and gels. Additionally, please take full advantage of our Extended figures (max. of 10) before making use of Supplementary figures (unlimited). The difference between them is that Extended figures are more easily available to readers, as they appear in the PDF and the HTML versions of the paper, whereas Supplementary figures need to be downloaded separately. Detailed formatting guidelines can be found here: <https://www.nature.com/natmetab/submission-guidelines/aip-and-formatting>. Key points are also summarized in a related Editorial (<https://www.nature.com/articles/s42255-023-00841-8>).

Response: We have paid attention to the formatting guidelines, and we prepared our figures in editable formats, present data as scatter plots (except for the DMR figures and the “trace graphs” for Seahorse and PercevalHR as there are so many data points that changing these figures to scatter plots would make them impossible to interpret), state the number of samples in the figure legends, provide source data for each figure, and provide uncropped scans of all blots and gels. In the revised manuscript, we also take advantage of the option of having Extended figures instead of Supplementary figures.

Reviewers' expertise:

Reviewer #1: Islet, Omics

Reviewer #2: Islet, DNA Methylation

Reviewer #3: Islet, DNA Methylation

Response to the Reviewers' comments:

Reviewer #1 (Remarks to the Author):

This manuscript explores CpG methylation throughout the genomes of alpha and beta cells in non-diabetic and T2D or pre-T2D donors. They use their dataset to identify a few critical genes that have DMRs and DEGs in diabetes and go on to test gain- and loss-of-function models of these genes in human islets or beta-cell lines. The authors further have made a public database for their data; however, although I would have liked to explore its functionality, it is not available yet. This manuscript adds a novel perspective that enriches our understanding and complements existing research in the field. Nonetheless, there are still areas that require clarification or further development to meet the standards for publication.

Response: We are very happy that reviewer 1 is generally positive towards our manuscript. We have addressed all important issues raised and believe these modifications improved our manuscript considerably. We also apologize that the public database / resource, alpha-beta-methylome (<https://alpha-beta-methylome.serve.scilifelab.se/app/alpha-beta-methylome>), was not available when reviewer 1 reviewed our manuscript. Our goal was that it should have been available during the review process. We have checked that it is available and hope reviewer 1 can access it now.

While revising our manuscript, we realized that there are two duplicate genes in the study by Suzuki et al (<https://doi.org/10.1038/s41586-024-07019-6>), hence they identified 849, not 851, T2D candidate genes. We have revised our manuscript accordingly. Also, when we performed hypergeometric analyses (based on comment 3 by reviewer 2), slight modifications were made to figures and tables where overlap between TFs binding and DMRs are presented.

Additionally, based on a comment by the Editor and after communication with her, we co-submit our paper with, and refer to the preprint PMID: 40631278 (see **Page 6** and **20** in the revised manuscript), that contains data from a similar analysis.

Major points

1. The donor characteristics table is insufficient. De-identified data should be provided for each donor, including the duration of diabetes and treatment of those with diabetes, if available. For example, see the Human Islet Checklist in the journal Diabetes. To improve clarity, the specific assays performed for each donor should be included alongside this information. Currently, the use of aggregation makes it difficult to determine which assays

were conducted on the same donor. For instance, it would have been impossible to recognize from these tables that only two nondiabetic females had WGBS data available in both alpha and beta cells, as described in lines 177-178.

Response: We appreciate this comment. The revised **Supplementary Table 1A** contains the clinical characteristics and specific assays performed for the individual islet donors included in this study. Unfortunately, we do not have information on duration of diabetes and treatment, and this information could hence not be included. Also, for purity of individual samples, see comment 1 by reviewer 3.

2. If possible, please compare the data with the GWAS/3C data in PMID: 36070683 (also sorted alpha and beta cells) to identify any overlaps in T2D-associated loci.

Response: Based on this good comment, we compared our DMR data with the Hi-C / three-dimensional chromatin map data in PMID: 36070683. When comparing our 22,544 α - versus β -cell DMRs ($q < 0.05$) with the 89 α - versus β -cell loop interaction points that had an adjusted p-value < 0.05 (presented in Supplemental Table 5 of PMID 36070683), then 30 (~34%) of their α - versus β -cell loop interaction points overlap with an α - versus β -cell DMR found in our study. Furthermore, 15 interaction points overlap with more than one DMR. This data has been added to **Supplementary Table 3H**, and on **Page 7**:

*“Finally, we intersected our α - versus β -cell DMRs with three-dimensional chromatin map data³⁴. Here, 30 of 89 chromatin interaction points that differ between α - and β -cells overlap with an α - versus β -cell DMR, with 15 overlapping with more than one DMR (**Supplementary Table 3H**, overlap z-score=8.3, $p < 0.001$).”*

3. In Figure 4c-e, are the colors showing paired samples the same samples throughout the figure? For example, are the red cells in c the same cells as in d? If so, why is there an n=4 for 4c-d but only an n=3 for 4E?

Response: We thank reviewer 1 for this comment. The colors of the dots are now matching, so that they represent samples from the same experiments in all panels for **Figure 4**. This has been clarified in the updated legend for **Figure 4**. Of note, based on review comments, in the revised manuscript, Figure 4 includes more experiments.

Legend, Fig. 4: *“The colors of the dots in g-m and n-q represent samples from the same experiments”*

For the original Figure 4e, now **Fig. 4m**, it's only n=3 as we unfortunately did not harvest total protein during the first experiment. Similarly, there are 6 experiments for GCG pyrosequencing and expression (**Fig. 4n** and **p**), but only 5 for glucagon content (**Fig. 4q**).

4. The authors attribute the dysfunction observed in OE-ONECUT2 islets solely to mitochondrial dysfunction; however, GCK is downregulated in this model. Please discuss this gene/protein as a potential contributor to the dysfunction observed upstream of the K-ATP channel.

Response: We thank reviewer 1 for this comment and have reworded the text slightly, changing from “mitochondrial dysfunction”, to e.g. “reduced mitochondrial activity” throughout the revised manuscript (see **Page 2, 14, 15, and 18**). Additionally, we have added a brief discussion on the possible role of reduced GCK in the phenotype of *ONECUT2*-overexpressing cells.

Page 18: *“OE-ONECUT2 in human islets and INS-1 β -cells, mimicking T2D, caused reduced GSIS, OCR, ATP-production, and expression of gene-sets regulating glucose metabolism and insulin*

secretion. The latter includes reduced expression of GCK, which encodes glucokinase, the rate-limiting enzyme in glycolysis. This could be an important change, leading to reduced production of pyruvate to fuel the mitochondrial tricarboxylic acid (TCA) cycle, oxidative phosphorylation, and ATP production⁷¹.”

5. I applaud the clear labeling for cell types used in Figure 7. However, it highlights the use of a rat cell line. Why were they used instead of the EndoC cells used in other parts of the manuscript? To be clear, I’m not asking for these experiments to be performed in another cell line; however, if there was difficulty in using the EndoC cells for this purpose, it should be reported.

Response: The reviewer raises a valid question. For the epigenetic editing experiments, we used the human EndoC cells as their genome matches the human islet α - and β -cells we used in our main analysis. Hence, we could design gRNAs with exactly the same genomic sequences as for identified DMRs. We also used the EndoC β -cells for the new rescue experiments where we silenced ONECUT2 in β -cells exposed to high glucose (19 mM) plus 1 mM palmitate for 72 h, mimicking a diabetogenic environment, because while there was Silencer Select siRNA available targeting human ONECUT2, there were no Silencer Select siRNAs targeting rat *Onecut2* (silencer select are overall more reliable siRNAs with reduced off-target effects), we hence did not do the rescue experiments in INS-1 832/13 β -cells.

However, in general, for the functional siRNA and OE-experiments, many groups within our research centre, LUDC, prefer the rat INS-1 832/13 β -cell line as it is our experience that they more closely match human islets in terms of response to manipulation. Below are examples from two different labs at LUDC where human islets and INS-1 832/13 β - cells responded in a similar fashion to siRNA transfection, while EndoC cells didn’t. We believe this could be due to the fetal origin of EndoC cells and potential physiological differences between immature and mature β -cells.

We have clarified this in the updated manuscript on **Page 13:**

“To mimic the situation in pre-T2D/T2D, selected genes showing lower expression in β -cells from donors with pre-T2D/T2D, i.e. ADCY9, FBLL1, ZNF431, and ZSWIM9, were silenced using siRNA

transfection in both human islets, and rat *INS-1* 832/13 β -cells (hereafter called *INS-1* β -cells) (Fig. 7c-e). In these experiments, we chose this cell line rather than *EndoC- β H1* as, in our hands, *INS1* β -cells behaved more like primary human β -cells than the fetal human *EndoC- β H1* in gene manipulation experiments (JKO and KB, unpublished observations).”

6. Please discuss why it makes sense that changes in adenyly cyclase activity would result in changes downstream of the K-ATP channel.

Response: We have added a brief expansion on this in the revised manuscript:

Page 15: “These data support that *OE-ONECUT2 dampens mitochondrial activity*, while *Adcy9*-deficiency impairs stimulus-secretion coupling in β -cells. The latter fits well with the extensive literature on the role of cAMP in enhancing insulin granule exocytosis in β -cells⁵⁸.”

7. ONECUT1 and ONECUT2 have nearly identical binding sites, and ONECUT1 (also known as HNF6) has established roles in beta-cell development and function and is linked to T2D and MODY. What are the relative expression levels between ONECUT1 and ONECUT2 in beta cells? They don't seem to be able to compensate for each other. Please discuss the known roles of ONECUT1 in light of your discovery.

Response: We thank reviewer 1 for raising this comment. We didn't detect *ONECUT1* expression in our sorted β -cell fraction, nor was it detected in fetal or adult β -cells sorted by Blodgett *et al* (PMID: 25931473) (see **Supp. Table 5B**). Moreover, there was a 10-fold higher expression of *ONECUT2* in adult compared to fetal β -cells (see Figure 1 below, PMID: 25931473). In our single cell RNA-seq dataset of human pancreatic islets, *ONECUT1* was very lowly expressed in β -cells compared to *ONECUT2* (see Figure 2 below, doi: 10.1038/s41598-025-21595-1).

Based on the importance of ONECUT1 for T2D, we now discuss the data in these studies on **Page 18**: *“Interestingly, ONECUT2 and ONECUT1 have nearly identical DNA binding sites, but data from our study and published studies suggest different roles for these two TFs⁷⁶. Mutations and common variants in ONECUT1 contribute to neonatal diabetes and T2D, respectively, and there is evidence supporting that ONECUT1 impacts pancreas development⁷⁷⁻⁷⁹. However, ONECUT1 expression was not detected in β -cells in our study, nor in fetal or adult β -cells sorted by Blodgett et al³⁵. Furthermore, there was a 10-fold higher expression of ONECUT2 in adult compared to fetal β -cells³⁵, and ONECUT1 showed very low expression compared to ONECUT2 in β -cells in our human islet single-cell RNA-seq data⁷⁵.”*

Minor points

1. In Supplementary Tables 3 and 7G, what does “Previously reported” refer to? For those sites that have not been reported before, does this mean your data shows a DMR, but the gene it’s associated with has not been identified as a diabetes-related gene?

Response: We apologize for this unclear information. **Supplementary Tables 3G** and **7G** were adopted from a table in Suzuki et al (<https://doi.org/10.1038/s41586-024-07019-6>), and column D with the title “previously reported” states whether a SNP included in Suzuki et al had previously been reported in other GWAS studies. However, since we do not believe this information adds value to our study, and to avoid confusion, we deleted this column in the **Supplementary Tables 3G, 6G, 7G, and 8G** of the revised paper.

2. For Figure 3g-o (and maybe for other similar data), would it be possible to color-code the individual data based on the sex of the donor?

Response: Based on this good suggestion, we color-coded the individual data in **Figure 3g-o** based on sex. We did the same in **Figure 2d-i** and **n** (right part), **Figure 3d-e**, **Figure 5m**, **Figure 6o-q**, **Figure 7c**, **Extended Data Figure 11-n** (right part), **Extended Data Figure 4b-c** (right part) and **e-g**, **Extended Data Figure 5b-k**. We added the following information to the legend of each of these figures:

“Women and men are represented by filled and open circles, respectively.”

3. Line 210—States supplementary Data 3 and 6. Does this refer to Tables or Figures?

Response: We thank reviewer 1 for spotting this mistake. It refers to **Supplementary Table 3** and **6**, and we corrected this in the revised manuscript.

4. It’s somewhat circular to argue that ONECUT2 is upregulated in both your overexpression model and in T2D beta cells, since you tested what would happen in normal islets when you upregulate ONECUT2 (Lines 380-382). Please explain why this gene is upregulated in both models within the text. Something like “In addition to ONECUT2, these genes were upregulated in both the ONECUT2 and T2D islets:”

Response: We agree with the reviewer, this wording was a mistake and we have changed the sentence on **Page 15**:

“In addition to ONECUT2, also SYNPO2, CFB, VEGFB, and PER3 were DEGs in both OE-ONECUT2 islets (Supplementary Table 12A) and T2D β -cells (Supplementary Table 10A).”

5. MODY is in parentheses but not written out to explain what it stands for (line 440).

Response: Thank you for detecting this mistake, which has been fixed on **Page 17** in the revised manuscript.

6. Other labs have had problems using siRNA in islets without the use of a virus or a light dissociation with accutase or trypsin. Was this a new protocol worked out for islets? Was it based on any other cell type?

Response: We first developed and reported this protocol in a paper published in Diabetes in 2022 (PMID: 34753799), where we double-transfected islets with siRNA and this yielded higher transfection efficiency compared to a single-transfection protocol used by other labs. Our double-transfection protocol has also been replicated in other studies (PMID: 36656641 and PMID: 38086799). However, we have not tested this protocol on any other cell types. Based on this valid comment, we added some information and references in the method section.

Page 27: *“In line with previous studies^{12,27}, a second transfection was performed 24 h after the first, and all functional experiments were performed 72 h after the first transfection.”*

Reviewer #2 (Remarks to the Author):

In this study, Ofori and colleagues established the DNA methylomes and transcriptomes of purified human alpha and beta cells from several donors, with or without T2D. These results allow the systematic analysis of genetic/epigenetic changes that accompany the development of T2D. In combination with T2D risk genes identified through GWAS and subsequent functional analysis, these data will likely provide a valuable resource for understanding the etiology of T2D development.

Response: We are very pleased that reviewer 2 believes that our study provides a valuable resource for understanding the etiology of T2D development, and we appreciate the very good comments that helped us improve our manuscript.

While revising our manuscript, we realized that there are two duplicate genes in the study by Suzuki et al (<https://doi.org/10.1038/s41586-024-07019-6>), hence they identified 849, not 851, T2D candidate genes. We have revised our manuscript accordingly. Also, when we performed hypergeometric analyses (based on comment 3 by reviewer 2), slight modifications were made to figures and tables where overlap between TFs binding and DMRs are presented.

Additionally, based on a comment by the Editor and after communication with her, we co-submit our paper with, and refer to the preprint PMID: 40631278 (see **Page 6** and **20** in the revised manuscript), that contains data from a similar analysis.

There are several main issues.

1. The number of duplicates is low, especially for the T2D cells. Granted, the materials needed are rare, and the DNA methylation process is demanding. To compensate, the authors should provide some analysis to inspect the degree of variation between individual samples (especially for the three T2D samples). Revealing such variations will allow readers to judge whether some of the data repeatability (or the lack of it) issues arise from donor variation or lab technique differences.

Response: Based on this good comment, we added analyses inspecting the degree of variation between individual WGBS samples. The heatmaps in the new **Extended Data Fig. 7** show an overall good correlation between the WGBS samples, and the dendrogram in the new **Extended Data Fig. 8** shows a clear separation between the α - and β -cells. See **Page 23:**

“Extended Data Fig. 7 and 8 present heatmaps and dendrogram of the WGBS samples, respectively.”

Additionally, all statistical analyses of our WGBS and RNA-seq data in human α - and β -cells take variance into account, and all Supplementary Tables presenting DMRs and DEGs include variance.

2. Similar to the above, the authors should utilize some of the recently published gene expression data in alpha/beta cells (by others) to investigate the repeatability of gene expression in human donor islet cells. Particularly intriguing would be exploring how repeatable the T2D-caused gene expression changes from different research groups are (e.g., comparing the authors' new findings with those reported in PMC11374460). The findings should be instructive for readers if similar studies were to be conducted to explore T2D etiology (at the molecular level).'

Response: We appreciate this comment and performed the suggested analysis. We found 20 T2D-associated DEGs in β -cells overlapping with that of Walker *et al* (PMC11374460), while 3 T2D-associated DEGs in α -cells were found in both studies. We added these results to **Supplementary Table 9** and **10**. One should remember that Walker *et al* used a gentler cut-off for what they considered expressed compared to our study; they required a minimum of 10 reads in 25% of samples, while we required a minimum of 20 reads in 40% of samples. Hence, some otherwise overlapping genes might have been filtered away in our analysis. We also investigated the overlap with another very recent single cell study (<https://doi.org/10.1038/s41467-025-65060-z>). This has been added to the results of the revised manuscript. See **Page 12:**

“When integrating these data with our published RNA-seq data from whole islets of a large T2D case-control cohort²⁷, and a sorted α -cell T2D case-control cohort⁵¹, we found 11 and 3 overlapping DEGs, respectively, including GNAL (Extended Data Fig. 5a and Supplementary Table 9B-C).”

*“Moreover, 20 and 19 DEGs found in our β -cells overlap with T2D-associated β -cell DEGs found in studies by Walker *et al* and Martínez-López *et al*, respectively (Supplementary Table 10E-F)^{51,52}.”*

We also discuss these results on **Page 20:**

*“A limitation is that only a few T2D-associated α -cell DEGs could be validated in two other studies^{27,51}. However, one needs to keep in mind that Walker *et al* used a gentler cut-off for inclusion of genes compared to our study⁵¹.”*

3. The statistical analyses need clarification. When presenting gene list overlaps, the authors need to perform hypergeometric analyses to show enrichment or anti-enrichment and p values. Similarly, the authors need to justify when different standards of cutoffs were used for gene selection: on lines 115-116, the authors set >2 mean normalized count for gene expression cutoff; on line 183, it was ≥ 20 . Similarly, FDR was set at <5% in some places but 10% in others – did the authors get different conclusions when staying with a same FDR?

Response: Based on this good suggestion, we have performed both gene set enrichment hypergeometric testing and region-based permutation testing in cases where spatial association between genomic regions was determined, on overlaps included in the manuscript. For hypergeometric tests, we defined different background genes universe sets for DMR-associated and DEGs gene sets, in the way that represent complete sets of genes capable of being detected and assessed for differential methylation/expression, respectively. In the case of overlaps between genomic regions we opted for a permutation testing approach, where the query regions of interest were randomized 1000 times across the genome while preserving their size and chromosomal distribution to create an empirical null

distribution. To assess the statistical significance, we compared the observed overlaps to the overlaps of the distribution using an empirical p-value. In both tests, we also calculated enrichment scores, which represents the strength of the association. Details in the methodology can be found on Page 34-35 (too long to add here). In our previous overlap analyses with TF binding sites, some DMRs were counted more than once because they are annotated to several genes. While performing the statistical analyses on the overlaps we only counted a DMR once, resulting in some changes in number of overlaps, marked in red in the text and updated in the Supplementary Tables and Figures, compared to the original version of our manuscript. The hypergeometric enrichment scores (HES), overlap z-scores and p-values from these analyses are included on:

Page 6: *“We then compared the list of 7,975 genes with α - versus β -cell DMRs identified in the current study with a list of genes presented in **Supplementary Table 9 and 10** of a preprint based on a similar study (accessed on 2025-10-28)³⁰. This showed that 468 out of 706 (66%) genes, including GCG, INS, MAFA, PDX1, in their study exhibit DMRs also in our study (hypergeometric enrichment score (HES)=2.07, $p=2.63E-58$).”*

On Page 7:

*“Additionally, we present DMRs that overlap with sequences bound by the islet-specific TFs, FOXA2, MAFB, NKX2.2, NKX6.1, and PDX1 in **Fig. 2k**, and **Supplementary Table 3F** (enriched overlap, $p<0.001$)²⁵.”*

*“Interestingly, 381 of the 849 (45%) T2D candidate genes also have DMRs in α - versus β -cells (**Fig. 2l** and **Supplementary Table 3G**, HES=1.74, $p=2.63E-34$).”*

*“Finally, we intersected our α - versus β -cell DMRs with three-dimensional chromatin map data³⁴. Here, 30 of 89 chromatin interaction points that differ between α - and β -cells overlap with an α - versus β -cell DMR, with 15 overlapping with more than one DMR (**Supplementary Table 3H**, overlap z-score=8.3, $p<0.001$).”*

Page 8: *“Additionally, we present X chromosome DMRs that overlap with sequences bound by FOXA2, MAFB, NKX2.2, NKX6.1, and/or PDX1 in **Extended Data Fig. 4d** and **Supplementary Table 4D** (enriched overlap, $p<0.001$).”*

Page 9: *“Next, we intersected these DEGs with the 849 GWAS T2D candidate genes² and 186 (22%) genes, including ADCY5, GLIS3, GLP1R, HNF4A, and INS, were overlapping (**Fig. 3q** and **Supplementary Table 6G**, HES=1.67, $p=1.07E-31$).”*

Page 10: *“Using HOMER, we found enrichment of TF motifs, including HNF1B and OCT4, in the T2D-associated α -cell DMRs (**Fig. 5i** and **Supplementary Table 7E**), and we identified DMRs that overlap with sequences bound by the islet-specific TFs, FOXA2, MAFB, NKX2.2, NKX6.1, and PDX1 (**Fig. 5j**, and **Supplementary Table 7F**, enriched overlap, $p<0.001$). Additionally, 103 of 849 (12%) T2D candidate genes discovered by GWAS² also have T2D-associated α -cell DMRs (**Fig. 5k** and **Supplementary Table 7G**, HES=7.34, $p=1.37E-238$).”*

Page 11: *“Additionally, we present T2D-associated β -cell DMRs that overlap with sequences bound by islet-specific TFs, FOXA2, MAFB, NKX2.2, NKX6.1, and PDX1 in **Fig. 6l**, and **Supplementary Table 8F** (enriched overlap, $p<0.001$). Further, 153 of 849 (18%) T2D candidate genes identified by GWAS² also have T2D-associated β -cell DMRs, including CDKN2A, GLIS3, INS, KCNQ1, NKX6-3, and LPL (**Fig. 6m** and **Supplementary Table 8F**, HES=2.69, $p=3.13E-85$).”*

Moreover, we agree with reviewer 2 that we should better justify/explain the different cutoffs (>2 mean normalized count versus ≥ 20) in these two analyses. Our goal with the following analysis: *“Genes with <2 mean normalized counts were categorized as not expressed, and the*

remaining 24,730 and 24,726 genes in α - and β -cells, respectively, were divided into three groups of similar size, categorized into low- (0-33 quantile), medium- (33-66 quantile) and high- (66-100 quantile) expression.”, was to divide all genes into 4 groups with different expression levels, in order to better comprehend how the degree of DNA methylation in different genomic regions relates to the level of gene expression. Here, we subsequently used a less stringent cutoff for non-expressed genes, because we also wanted to have one group of genes with “low-expression”.

In the other analysis: “Next, we studied expression in α - and β -cells from 16 non-diabetic donors (**Supplementary Table 1B**). Using RNA-seq, we found 13,001 and 12,876 genes expressed in α - or β -cells, respectively (≥ 20 mean normalized counts in $\geq 80\%$ of samples, **Supplementary Table 5A-B**).”, we were more conservative, as here our focus was to include genes that could have a biological role in the cells, and we believe >2 mean normalized count may be too low.

However, it is difficult to know whether these are the best cutoffs, and based on this good comment, we now discuss this on **Page 19**:

“We used different cutoffs for unexpressed genes in our study. We first used <2 mean normalized counts to categorize genes as not expressed, and we categorized the remaining genes into low-, medium- and high- expression. In the second analysis, we used ≥ 20 mean normalized counts in $\geq 80\%$ of samples for a gene to be considered expressed in α - and β -cells. The reason for using different cutoffs was that in the second analysis we wanted to include expressed genes that could have a biological role in the cells, while in the first analysis our goal was to investigate the degree of DNA methylation in relation to four different levels of expression. However, one should keep in mind that the cutoffs are based on our “best guess”.

FDR is an estimate of the number of false positives accepted in an analysis. We generally used $FDR \leq 5\%$, thereby accepting less than 5% false positives. However, for the T2D-associated DEGs in β -cells we had fewer samples and therefore lower power and used $FDR \leq 10\%$, hence accepting 10% false positives (see **Supplementary Table 10A**). Here, 8 protein coding DEGs had $q=0.083$. Hence, we accepted 8.3% false positives, which means that $0.083 \times 8 = 0.66$ false positives were included, i.e., less than 1 of the 8 DEGs should be a false positive. We then performed functional follow up experiments mimicking the T2D-associated expression of these 8 DEGs in human islets and clonal β -cells and found that most DEGs affected insulin secretion.

4. The authors did not provide a correlation between higher or lower methylation levels in DMRs with gene down- or up-regulation in alpha or beta cells. Why is this so. This reviewer believes that establishing such a correlation is necessary because the findings will suggest an activating/repressive role of methylation in gene expression.

Response: We thank reviewer 2 for this relevant comment. Our results in **Figure 1f-g** show that the global/genome-wide degree of DNA methylation is associated with the level of gene expression in α - and β -cells, and that the effect of DNA methylation on expression differs depending on genomic location of the methylation. In promoters and 5'UTR regions, low DNA methylation was associated with higher gene expression, while in intragenic regions, e.g., exons, introns and 3'UTR, slightly lower DNA methylation was associated with lower expression (**Fig. 1f-g**). This result is in line with our previous study in whole islets (doi: 10.2337/db16-0996).

However, it is not straight forward to correlate the DMRs with gene expression. The DMRs were generated by dmrseq (see **Page 23-24**) based on the degree of methylation of several

CpG sites (≥ 3 CpGs) and the full output from dmrseq, presented in **Supplementary Table 3**, does not provide individual methylation/DMR levels for the individual samples.

Page 23: *“To identify DMRs, we used dmrseq²⁶, which uses transformed methylation proportions, and fits a linear regression model using generalized least squares with a nested autoregressive correlated error structure. The dmrseq package “weighs” samples based on coverage and methylation state,”*

Therefore, in the original version of our manuscript, we explored how many of the DEGs between α - and β -cells also have DMRs. We found that $\sim 40\%$ (3,502) of these DEGs also have DMRs (**Supplementary Table 5F**), supporting epigenetic regulation of cell-specific expression (see e.g., **Fig. 2d-i**, **Extended Data Fig. 1l-n**, and **Fig. 3g-o**).

However, based on this comment, we performed a new analysis where we first divided the DMRs into two groups, those with higher methylation in α - versus β -cells, and the opposite. We then divided them depending on their genomic location and kept all DMRs annotated to genes. We then explored how certain groups of DMRs correlated with the DEGs and added the results on **Page 8** and in **Supplementary Table 5G**.

Page 8: *“Additionally, the DMRs correlated negatively with the DEGs (**Supplementary Table 5G**).”*

Of note, the location of differential methylation is generally believed to have a large impact on the effect on gene expression. As some genes have several DMRs annotated to them, and some DMRs are annotated to several genes, and a DMR can be annotated to several genomic regions, it is not “straight forward” to interpret this result.

Minor suggestions:

1. This reviewer commends the functional studies of several candidate genes. As a resource, it would be ideal if the authors could establish the causal relations between DNA methylation and transcription in a few more genes (not only *INS* as presented).

Response: Based on this excellent comment, we performed epigenetic editing using CRISPR-dCAS9-DNMT3A or CRISPR-dCAS9-TET1 together with two more gRNAs in/near two additional genes in human EndoC- β H1 β -cells. One new gRNA (*TH*-DMR-gRNA) targets *TH*, encoding tyrosine hydroxylase (see **Fig. 4c**). There is an α - versus β -cell DMR annotated to *TH* (see **Supplementary Table 3B** and **Fig. 4c, e**), and there is a “potential” *INS* enhancer in this region (see **Fig. 4a, c**). The other new gRNA (*GCG*-DMR-gRNA) is in an α - versus β -cell DMR located in the promoter of *GCG* encoding glucagon (see **Fig. 2g** and **4d**).

Based on comment 5 by reviewer 3, in the revised manuscript, we analyzed DNA methylation using both pyrosequencing, targeting a few selected CpG sites, and the EPIC v2.0 array (Illumina), targeting more CpG sites in/near the three studied genes (*INS*, *TH*, and *GCG*).

Importantly, epigenetic editing using both *TH*-DMR-gRNA and *GCG*-DMR-gRNA altered DNA methylation and gene expression. In our new experiments, we also identified altered methylation of more CpG sites in the *INS* locus. These new results are included in **Figure 4** and on **Page 8-9:**

*“Epigenetic editing reveals a causal role for DNA methylation in regulation of **INS**, **TH** and **GCG** expression*

*The cell-specific DNA methylation differences, which we found linked to large differences in expression of key α - and β -cell genes, suggest that the methylome directly regulates expression and potentially hormone levels in individual islet cell-types (**Fig. 2d-i** and **Supplementary Table 3B**). To*

test this, in human EndoC-βH1 β-cells, we used epigenetic editing with CRISPR-dCas9-DNMT3A or CRISPR-dCas9-TET1³⁶ together with guide RNAs (gRNAs) targeting three α- versus β-cell DMRs annotated to INS, TH, and GCG (Fig. 2e and, g, and Fig. 4a-f).

CRISPR-dCas9-DNMT3A-mediated editing, with the INS-DMR-gRNA targeting the INS DMR (Fig. 2e and Fig. 4a-b) or the TH-DMR-gRNA targeting a region annotated to TH (Fig. 4c and e), resulted in increased methylation of 21 and 8 CpG sites, respectively, as analyzed by pyrosequencing and EPIC v2.0 methylation arrays (Fig. 4g-j). Notably, after editing, five CpGs quite near the INS-DMR-gRNA target location showed absolute increased methylation of 20-43%, representing 2.46-30-fold increases, and these sites may have stronger impact on expression than the other sites (Fig. 4h). The DNMT3A-mediated editing using INS-DMR-gRNA and TH-DMR-gRNA caused reduced INS and TH expression, respectively (Fig. 4k-l). The targeted region in TH is located very near a potential INS enhancer (Fig. 4a and c) but editing using the TH-DMR-gRNA did not alter INS expression (Fig. 4k). Importantly, DNMT3A-mediated editing using the INS-DMR-gRNA led to reduced insulin content in human EndoC-βH1 β-cells (Fig. 4m).

Next, we used CRISPR-dCas9-TET1 and a gRNA (GCG-DMR-gRNA) targeting a DMR located in the promoter of GCG, encoding glucagon (Fig. 2g and 4d). This led to decreased DNA methylation at two CpG sites analyzed by pyrosequencing and EPIC v2.0 (Fig. 4n-o). Additionally, the TET1-based editing using the GCG-DMR-gRNA resulted in increased GCG expression and glucagon content in the EndoC-βH1 β-cells (Fig. 4p-q).

Importantly, this is to our knowledge the first study where epigenetic editing reveals a causal role for DNA methylation on INS, TH and GCG expression in β-cells, supporting future development of epigenetic-based therapies for regulation of insulin.”

The methods for these new experiments are included on Page 24-27.

2. The authors may consider a few more rescue experiments using human T2D islets.

Response: We agree with reviewer 2 that this would have been valuable. Unfortunately, we only receive islets from approximately two donors with T2D per year (33 T2D donors since 2008), making such experiments unfeasible during the revision of this manuscript. Instead, we performed rescue experiments in human EndoC-βH1 β-cells, which we exposed to a diabetogenic environment (19 mM glucose plus 1 mM palmitate (HGP) for 72h) that reduced glucose-stimulated insulin secretion (see **Extended Data Fig. 6j**). We chose to perform this experiment in the EndoC-βH1 β-cells since there was Silencer Select siRNA available for human ONECUT2, but no available Silencer Select siRNA targeting rat *Onecut2*. We could hence not do these experiments in the rat INS1 832/13 β-cells. Interestingly, here, we found that knockdown of *ONECUT2* rescued the perturbed insulin secretion caused by the diabetogenic environment (HGP). This new data has been added to the results on Page 18:

“Finally, we showed that silencing ONECUT2 in EndoC-βH1 β-cells exposed to high glucose plus palmitate (HGP, 19 mM glucose plus 1 mM palmitate), mimicking a diabetogenic environment, could rescue the perturbed insulin secretion seen in β-cells exposed to HGP (Extended Data Fig. 6i-j).”

We also added the following information to the methods section on Page 33-34:

“Rescue experiment in human EndoC-βH1 β-cells exposed to high glucose plus palmitate

One day before treatment and transfection, 180,000 EndoC-βH1 β-cells were seeded per well in a 48-well plate containing 150 μL DMEM medium without antibiotics (penicillin/streptomycin). The following day, cells received either fresh 150 μL control media or media containing high glucose plus palmitate (HGP, 19 mM glucose + 1 mM palmitate, mimicking diabetogenic conditions) with or without Silencer® Select Pre-Designed siRNA against ONECUT2 (50 nM; s18176; ThermoFisher

Scientific) using Silencer® Select Negative Control No. 2 siRNA as negative control (50 nM; 390846; ThermoFisher Scientific). Palmitate was administered conjugated to BSA, and unconjugated BSA was used as control. A final transfection volume of 200 µL per well contained 50 nM siRNA in Opti-MEM reduced serum media and 0.5 µL Lipofectamine RNAiMAX. A second transfection was performed 24 h after the first transfection. All functional experiments were performed 72 h after the first transfection. Cells were assayed for insulin secretion after reaching ~100% confluence. Confluent plates were carefully washed twice with 1 mL prewarmed SAB, pH 7.2 (1.16 mM MgSO₄, 4.7 mM KCl, 1.2 mM KH₂ PO₄, 114 mM NaCl, 2.5 mM CaCl₂, 25.5 mM NaHCO₃, 20 mM HEPES, and 0.2% BSA), containing 1 mM glucose. Cells were preincubated in a new 0.5 mL SAB with 1 mM glucose for 2 h. The cells were stimulated in 0.25 mL SAB at 37°C for 1 h with 1 mM glucose or 20 mM glucose. Although EndoC-βH1 β-cells are normally starved overnight before insulin secretion experiments to ensure a full response to glucose, we didn't starve the cells as this would interrupt the HGP treatment. Insulin was measured by ELISA (10-1113-01, Merck), and secreted insulin was normalized to the total protein. ONECUT2 silencing was confirmed by qPCR as described above.”

3. The presentation needs some clarification e.g.:

3.1. The first paragraph in the result section, the statement from lines 91-92 suggests that the authors obtained a total of ~300,000 cells from 24 samples. But it's actually ~300,000 cells from each of the 24 donors. In addition, the number of controls/T2D samples are missing from the text, which could have been very helpful (without the need to go to figure 1 and the sup. Tables).

Response: Based on this valid comment, we have rewritten this paragraph (**Page 5**) and we also changed one sentence in the introduction (**Page 3**):

Page 3: *“To fill these knowledge gaps, we used MARIS²³ to sort larger numbers (~300,000 per donor) of α- and β-cells from human pancreatic islets of donors with or without T2D, followed by whole-genome bisulfite sequencing (WGBS), the most comprehensive method to study methylation genome-wide²⁴, and RNA-sequencing (RNA-seq).”*

Page 5: *“Using MARIS for cell-sorting of human islets, we obtained on average 322,627±73,055 α-cells and 311,550±61,250 β-cells from 24 donors with or without T2D (**Fig. 1a** and **Extended Data Fig. 1a**). Characteristics of all donors in this study (17 non-diabetic controls and 7 pre-T2D/T2D) and donors for each analysis are presented in **Supplementary Table 1**.”*

3.2. Lines 152-154: does the sentence mean that each of the 5 TFs bind to 415 DMRs of 415 DMRs have at least one motif of the 5 TFs?

Response: We appreciate this comment. The sentence ‘All five TFs bound to 415 DMRs’ means that each of the 5 TFs binds to 415 DMRs. When we revised the manuscript and performed the statistical overlap analyses suggested in comment 3 this result changed because, in our previous overlap analyses with TF binding sites, some DMRs were counted more than once because they are annotated to several genes. However, while performing the statistical analyses on the overlaps we only counted a DMR once, resulting in some changes in number of overlaps. Subsequently, we changed the text on **Page 7:**

“Moreover, 270 DMRs, annotated to e.g., CREB5, GLP1R, HDAC9, OPRD1, and SYT13, were bound by all five TFs.”

3.3. Lines 161-162: it is not clear what the sentence means.

Response: We apologize for this and have tried to clarify the meaning of the sentence, now on

Page 7:

*“Next, since genetics affects T2D, especially via islet dysfunction², and since we demonstrated interactions between genetics and epigenetics^{32,33}, we studied the overlap between genes linked to T2D-associated SNPs² and the 7,975 genes with DMRs presented in **Supplementary Table 3**. Interestingly, 381 of the 849 (45%) T2D candidate genes also have DMRs in α - versus β -cells (**Fig. 2l** and **Supplementary Table 3G**, $HES=1.74$, $p=2.63E-34$). Additionally, 25 T2D-associated SNPs are located within DMRs annotated to e.g., *INS-IGF2*, *CDKN2B*, *PCSK1*, *GLP1R*, *MEG3*, and *KCNQ1* (**Fig. 2m** and **Supplementary Table 3G**). Notably, we found that 446 of 1,289 T2D SNPs² are in CpGs (**Supplementary Table 3G**). Together, this supports that T2D-risk alleles can affect DNA methylation, and potentially alter T2D-risk, via effects on methylation³².”*

3.4. What is the implication of data presented in the paragraph from lines 283-289, when islet data were integrated with that of pure cells?

Response: We thank reviewer 2 for this comment and our data presented on line 283-289 in our original manuscript show that 11 T2D-associated DEGs found in sorted α -cells were T2D-associated DEGs in whole islets in the study by Bacos *et al* (PMCID: PMC9927941). In the revised manuscript, we also compared the T2D-associated DEGs found in sorted α -cells in the present study with results in the study by Walker *et al* (PMC11374460), and here we could validate 3 DEGs. Based on these findings and this comment, we changed the sentence in the results, and we added a sentence to the discussion. See **Page 12:**

*“When integrating these data with our published RNA-seq data from whole islets of a large T2D case-control cohort²⁷, and a sorted α -cell T2D case-control cohort⁵¹, we found 11 and 3 overlapping DEGs, respectively, including *GNAL* (**Extended Data Fig. 5a** and **Supplementary Table 9B-C**).”*

We discuss these results on **Page 20:**

*“A limitation is that only a few T2D-associated α -cell DEGs could be validated in two other studies^{27,51}. However, one needs to keep in mind that Walker *et al* used a gentler cut-off for inclusion of genes compared to our study⁵¹.”*

For the T2D-associated β -cell DEGs, the validation in other cohorts was better, and here our functional validation experiments also support the robustness of these data. See **Page 12:**

*“Notably, 43 of the DEGs found in β -cells overlap with T2D-associated DEGs found in whole islets,²⁷ including *GABRA2*, *SOCS1*, *SYT1*, *PAX5*, *CABLES1*, and *TBC1D4*; and 90% of these went in the same direction, (**Extended Data Fig. 5a-c** and **Supplementary Table 10B**). Moreover, 20 and 19 DEGs found in our β -cells overlap with T2D-associated β -cell DEGs found in studies by Walker *et al* and Martínez-López *et al*, respectively (**Supplementary Table 10E-F**)^{51,52}.”*

3.5. If kept in the text, the authors need to give more detailed instructions on data access on the webpage described from lines 308-314.

Response: We appreciate this comment and added information in the result section of the revised manuscript as well as in the web-tool. See **Page 13:**

*“We proceeded to develop a web-based tool, alpha-beta-methylome (<https://alpha-beta-methylome.serve.scilifelab.se/app/alpha-beta-methylome>), a comprehensive open resource based on the human WGBS and RNA-seq data included in this study (**Fig. 1a-b** and **Supplementary Table 1B**), for investigators to explore and visualize the impact of age, sex, and pre-T2D/T2D on autosomal DNA methylation and/or expression of any gene in α - and β -cells, as well as differences between α - and β -cells, with the option to select p-value cutoffs and DNA methylation sequencing coverage. More detailed information is available at <https://alpha-beta-methylome.serve.scilifelab.se/app/alpha-beta-methylome>.”*

Information added in the web-tool:

“About

alpha-beta-methylome is a comprehensive open resource based on the whole genome bisulfite sequencing and RNA-seq data from sorted human pancreatic islet α - and β -cells included in (link to article). alpha-beta-methylome lets the user explore cell type-, T2D-, age-, and sex-associations in DNA methylation and gene expression.

How To

Min coverage: Choose the minimum sequencing coverage per CpG site in the DNA methylation analysis. **Note!** Choosing a higher cutoff will result in higher resolution but may also reduce the number of CpG sites and/or samples included in the analysis.

Comparison: Choose a comparison of interest.

- Age in α -cells
- Age in β -cells
- α - vs β -cells
- ND vs pre-T2D/T2D in α -cells
- ND vs pre-T2D/T2D in β -cells
- sex in α -cells
- sex in β -cells

In the age comparison a linear model adjusted for sex, BMI and number of days cultured is fitted to the data from individual CpG sites and gene expression. The plots and tables display the β -coefficient (slope), standard error for each CpG site and gene together with the t-statistics, and p-value.

*For the α - vs β -cell comparison, Wilcoxon matched pairs signed rank test is performed on individual CpG sites. Only samples with a coverage \geq **Min coverage** are included in the analysis and only CpG sites with data from at least 3 sample pairs are included in the analysis. Differential expression analysis was performed using DESeq2 (Wald test). The plots and tables display the mean β -value/expression and standard error of the mean for each CpG site and gene together with the p-value.*

*In the ND vs T2D comparison a Wilcoxon Rank-Sum test is performed on individual CpG sites. Only samples with a coverage \geq **Min coverage** are included in the analysis and only CpG sites with data from at least 3 samples in each group are included in the analysis. Differential expression analysis was performed using DESeq2 (Wald test). The plots and tables display the mean β -value/expression and standard error of the mean for each CpG site and gene together with the p-value.*

*In the sex comparison a Wilcoxon Rank-Sum test is performed on individual CpG sites. Only samples with a coverage \geq **Min coverage** are included in the analysis and only CpG sites with data from at least 3 samples in each group are included in the analysis. Differential expression analysis was performed using DESeq2 (Likelihood Ratio Test). The plots and tables display the mean β -value/expression and standard error of the mean for each CpG site and gene together with the p-value.*

Gene: Enter a gene symbol.

P-value cutoff: p-value cutoff to be used to only show results with a p-value lower than:

- 1
- 0.05
- 0.01
- 0.001

Extended chr search (bp): Extend the region around the selected gene, to include CpG sites, upstream of the transcription start site and downstream of the transcription end site by:

- 0 bp
- 200 bp

- 1000 bp
- 3000 bp
- 5000 bp

For more information regarding sample characteristics and data availability see accompanying scientific article: link to article

We also describe data access on Page 36:

“Data availability

The WGBS and RNA-seq data generated from human α - and β -cells were deposited in the LUDC repository (<https://www.ludc.lu.se/resources/repository>, accession numbers: LUDC2025.08.1 for the α -cell WGBS data, LUDC2025.08.3 for the β -cell WGBS data, LUDC2025.08.4 for the α -cell RNA-seq data, and LUDC2025.08.5 for the β -cell RNA-seq data).”

4. Some comments on the levels of gene over-expression on data interpretation are needed (Fig. 7)

Response: This is a valid comment, and we have added a brief discussion of this on Page 18:

“While our data strongly implicate ONECUT2 in T2D pathogenesis, one should keep in mind that the overexpression of ONECUT2 is stronger in our in vitro model than in T2D β -cells in vivo.”

5. Some discussion on the repeatability of human islet/beta-cell related data will be helpful

Response: We appreciate this comment, and in the original version of our manuscript we discussed the repeatability of our results in comparison to some other studies. We compared, e.g., age and sex associations in the sorted β -cells identified using the web-tool alpha-beta-methylome (<https://alpha-beta-methylome.serve.scilifelab.se/app/alpha-beta-methylome>) in the present study with our previously published data showing associations between age and sex and DNA methylation in whole human pancreatic islets (see Page 19). We also discuss the T2D-associated DMRs in β -cells in the present study in comparison with T2D-associated DNA methylation in whole islets in our previous study (doi: 10.1038/s41467-023-43719-9, reference 12, Page 20). Based on this comment, we expanded this discussion slightly.

Page 19: *“This study focused on dissecting cell-specific and pre-T2D/T2D-associated methylation in α - and β -cells. However, since we previously showed that age and sex impact DNA methylation, expression and insulin secretion in human islets^{80,81}, we included analyses of age- and sex-associations on methylation and expression in our web-tool, alpha-beta-methylome. Interestingly, sex-associated methylation and expression differences previously found in islets of e.g., *BCL11A*⁸¹, were also identified in β -cells by exploring alpha-beta-methylome. *Bcl11a*-silencing reduced GSIS in β -cells, mimicking lower *BCL11A* expression and insulin secretion found in islets from males⁸¹. Additionally, age-associated β -cell methylation of e.g., *KLF14*, *FHL2*, and *ZNF518B*, was identified using our web-tool. These genes also showed age-associated methylation in islets⁸⁰.”*

Page 20: *“While this study provides novel data, some cell-specific findings could be “validated” in larger cohorts of human islets, e.g. sex- and age-associations described above^{80,81}, and T2D-associations described in the results e.g., 65 T2D-associated β -cell DMRs overlap 91 CpGs exhibiting differential islet methylation in T2D¹², supporting their robustness. Functional validations and results from pathway analyses further support validity of our data.”*

We also discuss the data in the present study in relation to a recent preprint (<https://doi.org/10.1101/2025.07.02.662814>).

Page 20: *“Interestingly, 75.5% of the T2D-associated β -cell DMRs identified in this study were hypomethylated, which is in line with our previous study where 77% of T2D-associated methylation sites had lower methylation in whole islets from donors with T2D¹², and a recent preprint supporting hypomethylation in β -cells from donors with T2D³⁰. A limitation is that only a few T2D-associated α -cell DEGs could be validated in two other studies^{27,51}. However, one needs to keep in mind that Walker et al used a gentler cut-off for inclusion of genes compared to our study⁵¹.”*

Due to lack of space/limitation in word count, we did not add more examples of repeatability to the discussion. However, in the result section, there are several examples, e.g., in relation to a study by Blodgett et al (doi.org/10.2337/db15-0039).

Page 8: *“Next, we studied expression in α - and β -cells from 16 non-diabetic donors (Supplementary Table 1B). Using RNA-seq, we found 13,001 and 12,876 genes expressed in α - or β -cells, respectively (≥ 20 mean normalized counts in $\geq 80\%$ of samples, **unique Ensembl gene ID, Supplementary Table 5A-B**). Notably, 93% and 94% of the genes found to be expressed in α - and β -cells in our data set overlap with previously published work³⁵, after reanalyzing that data using the same conditions as ours (Supplementary Table 5A-B).”*

Reviewer #3 (Remarks to the Author):

Ofori et al describe whole genome bisulfite sequencing data, as well as transcriptomes, for pancreatic alpha and beta cells sorted from people with and without type 2 diabetes. They used samples from 14 donors – not a huge number but certainly sufficient – and sequenced methylation at a relatively low depth of 10x.

They report thousands of differentially methylated regions that distinguish alpha from beta cells, as well as ND vs diabetic alpha and beta cells, and link some of these changes to altered gene expression (in the classic expected sense – inverse correlation between methylation and expression).

They also perform some manipulations – forced methylation of the insulin gene which causes reduced insulin expression, and silencing or overexpression of genes that shows differential expression in beta cells from T2D, in human islets and even in the GK rat model. This part (in particular the latter, involving knockdown or overexpression of genes based on the transcriptome data, and particularly the rat data) represents a large body of work with impressive results, but is only loosely connected to the overall theme of the paper on DNA methylation, which is unfortunate.

Overall, the resource generated in this study will be important for the community for many studies. I have several concerns and comments, which the authors should be able to address.

Response: We want to thank reviewer 3 for the constructive comments that helped us improve our manuscript and we were happy to see that reviewer 3 believes the resource generated in this study will be important for the community.

While revising our manuscript, we realized that there are two duplicate genes in the study by Suzuki et al (<https://doi.org/10.1038/s41586-024-07019-6>), hence they identified 849, not 851, T2D candidate genes. We have revised our manuscript accordingly. Also, when we performed hypergeometric analyses (based on comment 3 by reviewer 2), slight modifications were made to figures and tables where overlap between TFs binding and DMRs are presented.

Additionally, based on a comment from the Editor and after communication with her, we co-submit our paper with, and refer to the preprint PMID: 40631278 (see **Page 6** and **20** in the revised manuscript), that contains data from a similar analysis.

1. Quality of data. I could not find a clear statement regarding purity of the sorted cell populations. Evidence should be presented based on deconvolution of the methylomes to identify the fraction of non-beta (or non-alpha) cells, with support from transcriptome analysis. This should be detailed for each sample, with a particular attention to the possibility that small contaminants are affecting the differences identified between ND and diabetic samples.

Response: We thank reviewer 3 for this comment. The purity of the α - and β -cell fractions, based on flow cytometry, was included on **Page 22** in the original version of our manuscript:

Page 22: *“The sorting of α - and β -cells was performed on an Aria Fusion (BD Biosciences, San Jose, CA, USA), achieving purity of $96.5\% \pm 4.0$ for β -cells and $92.0\% \pm 7.6$ for α -cells.”*

In the revised manuscript, we also calculated α - and β -cell purity by deconvolution, using our expression and methylation data in α - and β -cells, according to the method described by Loyfer et al (<https://doi.org/10.1038/s41586-022-05580-6>). Here, GCG and INS were used as marker genes and we added this information to **Page 22** of the revised manuscript. Furthermore, since purity estimated by using the gene expression data was available for most donors, we included this data for each donor/sample in **Supplementary Table 1A**. Additionally, there was no difference in purity between donors with and without T2D (P-values between 0.9-0.21 for the different methods estimating purity, data not included in the manuscript). Overall, the different methods used to estimate purity generated similar results, supporting high purity. Furthermore, our α - versus β -cell DMR Figures of known marker genes, e.g. PDX1, further support high purity of the sorted cell fractions.

Page 22: *“Sample purity was also determined by deconvolution according Loyfer et al²⁰ using GCG and INS as marker genes for α - and β -cells, respectively, and either the gene expression (purity of α -cells = $99.3\% \pm 0.3$ and β -cells = $99.5\% \pm 0.1$) or DNA methylation (purity of α -cells = $94.2\% \pm 0.8$ and β -cells = $99.1\% \pm 0.4$) data generated by RNA-seq or WGBS (see below). Purity for individual samples was included in **Supplementary Table 1A**.”*

2. Along the same line of quality control – ND / diabetic samples should be reshuffled to ensure that that differences observed do not represent an overfit.

Response: We thank reviewer 3 for this comment. We already performed such analyses and apologize that this was not clear in the original version of our manuscript. As mentioned in the methods on **Page 23**, we used permutations in the DMR calling, and in **Supplementary Tables 7A-C** and **8A-C** we write “*permutation p-value*” as a heading in the column presenting the p-values. However, based on this comment we added the following information, see **Page 23-24**:

“The permutations involved calculating an initial test statistic from the observed data, then repeatedly shuffling the data and recalculating the statistic to build a reference distribution. The final p-value is the proportion of shuffled results that are as extreme or more extreme than the original statistic.”

3. Novelty. While the dataset is expected to be useful for many in the field for multiple applications, there are no novel insights from the analysis so far, which is unfortunate. The authors should make efforts in this directions. Some loose suggestions below.

Response: Based on this comment and comments below, we have tried to emphasize the novel insights from this study. However, since previous instructions by Nature journals were to avoid the use of “novel” or “for the first time” in the text of the manuscript, we avoided such wordings in the manuscript. Nevertheless, we believe this study includes many novel results, such as; **i**) the individual DMRs presented in **Supplementary Tables 3, 4, 7** and **8**, as well as in many of our figures, that have never been published before, **ii**) the T2D-

associated SNPs overlapping α - versus β -cell DMRs and T2D-associated DMRs is important new information, **iii**) the epigenetic editing experiments provide key information that for the first time show direct evidence that DNA methylation causes expression changes of *INS*, *GCG* and *TH* in human β -cells, **iv**) the functional follow-up experiments of eight T2D-associated DEGs in β -cells clearly show that *ONECUT2* affects insulin secretion via reduced ATP production, and **v**) the unique web-based tool that allow users to explore age-, sex-, T2D-associated and α - versus β -cell differences is DNA methylation and gene expression.

4. Figure 2h – interesting to see expression (though low level) and demethylation of *SLC2A2* (*Glut2*), the transporter that is famously functioning as a glucose sensor in mouse beta cells but does not play a role in human beta cells. Please discuss this. It could potentially reflect a history of expression in neonate human beta cells, where its absence causes transient neonatal diabetes (Hattersley).

Response: We appreciate this comment and agree with reviewer 3 that the role of *GLUT2* in human β -cells should be discussed.

Page 19: *“This study provides a valuable recourse for islet research by identifying genomic regions where DNA methylation seems to regulate cell-specific expression in human α - and β -cells. For example, we discovered a region annotated to the promoter of *SLC2A2*, encoding *GLUT2*, with hypomethylation in β -cells compared to α -cells. In line, we found *SLC2A2* expression in β - but not α -cells. *GLUT2* is an important glucose transporter in rodent islets⁸², but its importance in human islets has been questioned⁸³. Nevertheless, Hattersley’s group found *SLC2A2* mutations causing neonatal diabetes, and they proposed that *GLUT2* affects insulin secretion in humans⁸⁴. The epigenetic and expression data of *SLC2A2* in human β -cells in the present study together with data from our previous study, where we found reduced *SLC2A2* expression in islets from donors with T2D and that *SLC2A2* knock-down reduced insulin secretion in human islets²⁷, support a role for *GLUT2* in mature β -cells.”*

5. Figure 4 – forced methylation of the insulin gene in a beta cell line. This is a missed opportunity! Nobody will be surprised to hear that *INS* methylation reduces expression. Insights may come from comparing the effect of multiple gRNAs: which CpGs matter, and must be demethylated to allow for insulin expression? Cracking the code of which CpGs matter and how many of them matter would be an important addition to the study. At a minimum, profile methylation more broadly than with pyro sequencing, and show the methylation status of the entire locus after editing. How many CpGs were successfully targeted?

Response: Based on this excellent comment, we dissected the epigenetic regulation of *INS* gene expression in more detail, and in the revised manuscript, we analyzed DNA methylation using both pyrosequencing, investigating a few selected CpG sites, and with the EPIC v2.0 array (Illumina), investigating more CpG sites in/near the studied genes (see **Fig. 4a, b, g, and h**, and **Pages 9 and 22-26**).

Moreover, based on minor comment 1 by reviewer 2 and comments from the Editor, we also added epigenetic editing experiments targeting two additional genes, *TH* and *GCG*. Here, we performed epigenetic editing using CRISPR-dCAS9-DNMT3A or CRISPR-dCAS9-TET1 with two more gRNAs (*TH*-DMR-gRNA and *GCG*-DMR-gRNA) in/near *TH* or *GCG*, respectively, in human EndoC- β H1 β -cells (see **Fig. 4**, and **Pages 9 and 22-26**).

Overall, in the revised manuscript, we analyzed DNA methylation using both pyrosequencing, investigating a few selected CpG sites, and with the EPIC v2.0 array (Illumina), investigating more CpG sites in/near the three studied genes (*INS*, *TH* and *GCG*). The number of analyzed CpG sites per gRNA/locus are presented in the methods section on **Page 22-26**. Unfortunately, several pyrosequencing assays covering the genomic regions targeted by and surrounding the three investigated gRNAs failed, potentially due to CG rich

regions. Subsequently, we were unable to analyze methylation of all CpG sites in these regions using pyrosequencing.

When analyzing methylation in larger genomic regions, we identified altered DNA methylation of more CpG sites, e.g., in total, 21 sites showed altered methylation in the *INS* locus after DNMT3A-based editing with the *INS*-DMR-gRNA (see **Fig. 4a, b** and **g-h**). Among these, six CpG sites are located within the *INS* promoter (-300bp), whereas 15 CpG sites are located either in the gene body or downstream of the *INS* gene. We observed absolute methylation changes of up to 40% (30-fold) for these CpG sites. When it comes to identifying which CpGs that matter the most, those sites showing the biggest effect size after editing may be most important for regulation of gene expression. Based on the relatively large effect sizes, for *INS*, we found several sites that could have a big impact on the regulation on gene expression. For *TH* and *GCG*, the effect sizes on methylation were less dramatic. Based on the sites possible to analyze with pyrosequencing and the EPIC v2.0 array, for those two genes, we may not have analyzed the sites with the biggest effects on gene expression.

The new results are included in **Figure 4** and on **Page 8-9:**

“Epigenetic editing reveals a causal role for DNA methylation in regulation of *INS*, *TH* and *GCG* expression

*The cell-specific DNA methylation differences, which we found linked to large differences in expression of key α - and β -cell genes, suggest that the methylome directly regulates expression and potentially hormone levels in individual islet cell-types (Fig. 2d-i and Supplementary Table 3B). To test this, in human EndoC- β H1 β -cells, we used epigenetic editing with CRISPR-dCas9-DNMT3A or CRISPR-dCas9-TET1³⁶ together with guide RNAs (gRNAs) targeting three α - versus β -cell DMRs annotated to *INS*, *TH*, and *GCG* (Fig. 2e and, g, and Fig. 4a-f).*

*CRISPR-dCas9-DNMT3A-mediated editing, with the *INS*-DMR-gRNA targeting the *INS* DMR (Fig. 2e and Fig. 4a-b) or the *TH*-DMR-gRNA targeting a region annotated to *TH* (Fig. 4c and e), resulted in increased methylation of 21 and 8 CpG sites, respectively, as analyzed by pyrosequencing and EPIC v2.0 methylation arrays (Fig. 4g-j). Notably, after editing, five CpGs quite near the *INS*-DMR-gRNA target location showed absolute increased methylation of 20-43%, representing 2.46-30-fold increases, and these sites may have stronger impact on expression than the other sites (Fig 4h). The DNMT3A-mediated editing using *INS*-DMR-gRNA and *TH*-DMR-gRNA caused reduced *INS* and *TH* expression, respectively (Fig. 4k-l). The targeted region in *TH* is located very near a potential *INS* enhancer (Fig. 4a and c) but editing using the *TH*-DMR-gRNA did not alter *INS* expression (Fig. 4k). Importantly, DNMT3A-mediated editing using the *INS*-DMR-gRNA led to reduced insulin content in human EndoC- β H1 β -cells (Fig. 4m).*

*Next, we used CRISPR-dCas9-TET1 and a gRNA (*GCG*-DMR-gRNA) targeting a DMR located in the promoter of *GCG*, encoding glucagon (Fig. 2g and 4d). This led to decreased DNA methylation at two CpG sites analyzed by pyrosequencing and EPIC v2.0 (Fig. 4n-o). Additionally, the TET1-based editing using the *GCG*-DMR-gRNA resulted in increased *GCG* expression and glucagon content in the EndoC- β H1 β -cells (Fig. 4p-q).*

*Importantly, this is to our knowledge the first study where epigenetic editing reveals a causal role for DNA methylation on *INS*, *TH* and *GCG* expression in β -cells, supporting future development of epigenetic-based therapies for regulation of insulin.”*

The methods for these new experiments were added to **Page 24-27:**

“Epigenetic editing, pyrosequencing, MethylationEPIC v2.0 array analyses, real-time quantitative PCR and insulin ELISA

Three different sets of epigenetic editing experiments were done in human EndoC- β H1 β -cells, which were seeded (180,000 cells/well) in 48-well plates 48 h before transduction. Next, the cells were transduced with all-in-one lentiviral constructs³⁶ at a multiplicity of infection (MOI) of 10 in the growth medium without antibiotics (penicillin/streptomycin) for 6 h. The constructs contain inactivated Cas9 (dCas9) fused to the DNA methyltransferase DNMT3A, and either INS-DMR-gRNA, a gRNA (sequence TGAGACTATAAAGCCAGCGGGGG) targeting an INS DMR (Fig. 2e and 4a-b), or TH-DMR-gRNA, a gRNA (sequence GCAGCTGTACTGGTTCACGG) targeting TH (Fig. 4c) as well as a potential INS enhancer (Fig. 4a and c). In the third set of editing experiments, EndoC- β H1 β -cells were transduced with a construct containing GCG-DMR-gRNA, a gRNA (sequence CATGCGTGATTGAAAGTAGAGCG) targeting the DMR presented in Fig. 2g and 4d, dCas9 fused to TET1 (Ten-eleven translocation methylcytosine dioxygenase 1), an enzyme involved in demethylating DNA. The virus containing media was then removed and replaced with complete growth medium. 48 h after transduction, cells were transferred into 24-well plates, and after an additional 48 h, cells were put into selection medium, including 4 μ g/ml puromycin, for 7 days. The selection medium was renewed after 3 days. After puromycin selection, cells from the same well were collected for RNA and DNA extraction, and cells from different wells were collected for protein extraction.

DNA and RNA were then prepared from the EndoC- β H1 β -cells using the Allprep RNA/DNA Mini Kit (#80204, QIAGEN, Hilden, Germany) according to the manufacturer's instructions. The DNA and RNA concentrations were measured using a NanoDrop (ND-1000; ThermoFisher Scientific, Waltham, MA, USA).

Next, we bisulfite converted 200 ng of DNA using the EpiTect Bisulfite Kit (#59104, QIAGEN) according to the manufacturer's instructions. Using the PyroMark Assay design software 2.0 (QIAGEN), the following pyrosequencing assays were designed: INS-forward primer GATTTTTTATTTTAGGTTTTAATGGG, INS-reverse primer ACAAACCTACTTAATAACCTCTTCTAATA, INS-sequencing primer TTTATTTTAGGTTTTAATGGGT, TH-forward primer GGGGTGATTTTATTTAAGGAATTTATTTAG, TH-reverse primer CCTAACCCACCTAAACTTATCCTT, TH-sequencing ATTGTTTGTGAGGAG, GCG-forward primer GGATATGTATAAAATAGGGATGGTTATGG, GCG-reverse primer AAAAAACAACCTAAAAATCCACCTTCTA and GCG-sequencing primer AGGGATGGTTATGGG. Pyrosequencing was then run on a PyroMark Q48 autotprep instrument (QIAGEN). Of note, several pyrosequencing assays covering the genomic regions targeted by and surrounding the three investigated gRNAs failed, potentially due to CG rich regions. Subsequently, we were unable to analyze methylation of all CpG sites in these regions by using pyrosequencing.

We proceeded to use the Infinium MethylationEPIC v2.0 BeadChip array (Illumina) to study DNA methylation of CpG sites covered by EPIC v2.0 probes in the INS, TH, and GCG loci in samples from the epigenetic editing experiments. These arrays were analyzed at Eurofins Denmark. DNA was bisulfite-treated using the EZ-96 DNA Methylation kit (Zymo Research Corp.). Preprocessing and quality control of the methylation data was performed using the R Statistical Software version 4.3.3 (<https://www.R-project.org/>). Probes on the array that target single nucleotide polymorphisms, non-CpG sites, cross-reactive probes⁹⁶, Y chromosome-specific probes, probes targeting single nucleotide variants of common cancer driver mutations, and probes with a detection p-value > 0.01 in more than 10% of the samples were removed. The intensities from the remaining probes were then corrected for background and dye bias using the NOOB-function from the R-package minfi version 1.48.0⁹⁷, and intra-sample normalized with the BMIQ-function included in the R-package wateRmelon version 2.8.0⁹⁸. To minimize batch effects, all paired editing samples (control and gRNA) were placed on the same array. Paired t-tests were then used to analyze the methylation data covered by EPIC v2.0 probes 10kb upstream of the TSS, the gene body, and 10kb downstream of INS, TH, and GCG. Here,

the EPIC v2.0 array covered 56 sites for INS, 62 for TH, and 14 for GCG. M-values were used for all bioinformatic and statistical analyses, and they were then converted to beta-values (1-100% methylation) for presenting the results.

cDNA was generated with the RevertAid First Strand synthesis kit (#K1622, ThermoFisher Scientific). Gene expression was analyzed by real-time quantitative PCR (qPCR) in triplicates on a 384-well plate with TaqMan assays (ThermoFisher Scientific) for INS (HS00355773_m1) TH (Hs00165941_m1), GCG (Hs01031536_m1), and PPIA (Hs04194521_s1, endogenous control), in a QuantStudio 7 Flex Real Time PCR system (ThermoFisher Scientific) under default cycling parameters.

Total protein from each well with EndoC-βH1 β-cells was extracted by using 200 μL RIPA buffer: 0.1% SDS, 150 nM NaCl, 1% Triton X-100, 50 mM Tris-Cl, pH 8 and EDTA-free protease inhibitor (Roche, Basel, Switzerland). The protein amount/concentration was analyzed using the BCA Protein Assay kit (ThermoFisher Scientific). Insulin and glucagon levels in the cells were then analyzed by ELISA (insulin: #10-1145-01; glucagon: #10-1271-01, Mercodia, Uppsala, Sweden). BCA and ELISA plates were read on a CLARIOstar microplate reader (BMG Labtech, Ortenberg, Germany).”

6. One important finding is that almost half of the genes that are differentially methylated in alpha vs beta cells are adjacent to T2D SNPs, while only 22% of the differentially expressed genes overlap with T2D genes. This deserves some discussion and interpretation.

Response: Based on this valid comment, we added information to **Page 18** of the discussion. However, we find this quite difficult to discuss and believe future studies should investigate the reason behind this finding further.

Page 18: *“We also found that 45% and 22% of T2D-associated genes/locus identified by GWAS overlap the α- versus β-cell DMRs and DEGs, respectively. The reason for the higher overlap with genes annotated to DMRs versus DEGs could be because DNA methylation is more stable than gene expression, where expression of some genes represents more of a “snapshot” at the time samples were collected. However, this needs to be further investigated.”*

7. Genetics vs environment: the authors report 18% of T2D candidate genes identified by GWAS are differentially methylated in T2D vs ND beta cells (Figure 6m). This is nice, and gives a minimum estimate for the extent of genetic effect on methylation (likely much broader due to effects on remote loci). Please draw the reverse chart: what percent of the beta or alpha cell T2D DMRs identified here are adjacent to a known GWAS T2D hit? This may give very important information about the potential impact of the environment on epigenetics in T2D, compared with the extreme hypothesis that all differential methylation in diabetes is dictated genetically. From looking at Supp Table 8 I think, but a, not sure, that there are 5000 T2D DMR in beta cells, and of these 1300 are adjacent to a locus genetically linked to T2D.

Response: We appreciate this comment that addresses an important point. When drawing the reverse chart, 5.5% and 5.1% of the genes annotated to T2D-associated DMRs overlap known GWAS T2D genes for α- and β-cells, respectively (see figures below). Hence, ~95% of the T2D-associated DMRs do not overlap GWAS T2D genes, supporting the theory that a smaller proportion of differential methylation in T2D is genetically linked, while a larger proportion seems to be linked to environmental factors. In β-cells, there are 5106 T2D-associated DMRs, annotated to 3020 genes, and among these 153 genes were found to be linked to T2D-SNPs by Suzuki *et al.*

We added this information to the result section. However, due to limited space, we did not include the new overlap figures in the revised manuscript. However, if reviewer 3 believes it would be necessary, we will be happy to add them.

Page 10-11: “Additionally, 103 of 849 (12%) T2D candidate genes discovered by GWAS² also have T2D-associated α-cell DMRs (Fig. 5k and **Supplementary Table 7G**, $HES=7.34$, $p=1.37E-238$). This represents 5.5% of the 1,871 genes annotated to T2D-associated α-cell DMRs. The overlapping genes include *GIP*, *GLIS3*, *GRB10*, *HDAC9*, *IGF2*, *IGF2BP3*, *KCNQ1*, *KDM4B*, *MEG3*, and *TCF7L2*.”

Page 11: “Further, 153 of 849 (18%) T2D candidate genes identified by GWAS² also have T2D-associated β-cell DMRs, including *CDKN2A*, *GLIS3*, *INS*, *KCNQ1*, *NKX6-3*, and *LPL* (Fig. 6m and **Supplementary Table 8F**, $HES=2.69$, $p=3.13E-85$). This represents 5.1% of the 3,020 genes annotated to T2D-associated β-cell DMRs.”

8. Figure 5b: the KEGG pathway names are meaningless (alpha cells have nothing to do with synapses or morphine addiction). I realize that this represents enrichment of islet genes that were assigned to these brain sets, but authors should find a better way to describe or at least explain the findings that will be comprehensible to most readers and will also provide some biological sense. The circadian link is potentially interesting and should be further developed – what are the genes? What is the expected phenotypic outcome?

Response: We agree with reviewer 3 and modified **Figure 5b** to only show Glutamatergic synapse, Insulin secretion and Circadian entrainment KEGG pathways. Also, we have indicated in **Supplementary Table 7D** that Morphine addiction and Axon guidance KEGG pathways may not directly be linked to α-cell function.

The genes linked to the enriched Circadian entrainment pathway include: *ADCY8*, *ADCY9*, *ADCY10*, *GRIN2A*, *GNGT1*, *KCNJ5*, *PRKCB*, *PRKCA*, *GNG7* etc. For the expected phenotypic outcome, we can only speculate since further studies are needed to elucidate this but interestingly, cAMP signaling (*ADCY8*, *ADCY9*, *ADCY10*) is implicated in circadian entrainment/oscillation (e.g., via CREB, PKA, cAMP rhythms) (PMID: 18487196) and cAMP acts as a central intracellular amplifier of glucagon secretion in α-cells, primarily by activating PKA and Epac pathways that enhance calcium signaling and exocytosis (PMID: 30953108). However, due to limited space in the manuscript, we only made the following changes to

Page 10:

“These DMRs were annotated to 1,871 genes, including *GCG*, *INS-IGF2*, and *PDX1*, enriched in KEGG pathways, including circadian entrainment containing *ADCY8*, *ADCY9*, and *ADCY10* etc. (Fig. 5b and **Supplementary Table 7D**).”

9. Comparisons were made in the past between the epigenome, including the methylome, of alpha and beta cells. Please cite and discuss differences if any and the novelty here.

Response: We appreciate this comment, and a preprint by Manduchi et al (PMID: 40631278), was uploaded to bioRxiv in July 2025, shortly before we submitted our manuscript to Nature Metabolism. We refer to their study in our revised manuscript, and we made some comparisons between our study and theirs (see **Page 6** and **20**). Additionally, based on a comment from the Editor and after communication with her, we co-submit our paper with PMID: 40631278.

Page 6: *“We then compared the list of 7,975 genes with α - versus β -cell DMRs identified in the current study with a list of genes presented in **Supplementary Table 9 and 10** of a preprint based on a similar study (accessed on 2025-10-28)³⁰. This showed that 468 out of 706 (66%) genes, including GCG, INS, MAFA, PDX1, in their study exhibit DMRs also in our study (hypergeometric enrichment score (HES)=2.07, p=2.63E-58).”*

Page 20: *“Interestingly, 75.5% of the T2D-associated β -cell DMRs identified in this study were hypomethylated, which is in line with our previous study where 77% of T2D-associated methylation sites had lower methylation in whole islets from donors with T2D¹², and a recent preprint supporting hypomethylation in β -cells from donors with T2D³⁰. A limitation is that only a few T2D-associated α -cell DEGs could be validated in two other studies^{27,51}. However, one needs to keep in mind that Walker et al used a gentler cut-off for inclusion of genes compared to our study⁵¹.”*

Although there are some similarities between our study and theirs, for example both measured DNA methylation and gene expression genome-wide in human α - and β -cells, Manduchi *et al* focused on epigenetic adaptation of β -cells across lifetime and disease, and they found that age-related demethylation is advanced in donors with T2D, while our study focused on more detailed presentation of all DMRs in α - versus β -cells as well as T2D-associated DMRs in α - and β -cells, and we then related the identified DMRs to the transcriptome, TFs and T2D candidate genes and SNPs identified by GWAS. Their preprint did not include tables with detailed information of the α - versus β -cells DMRs or T2D-associated DMRs in α - and β -cells as can be seen in our **Supplementary Tables 3A-C, 7A-C, and 8A-C**, instead it includes lists with only gene symbols. Additionally, in contrast to the study by Manduchi *et al*, our study includes many functional experiments, such as epigenetic editing of regions in/near discovered DMRs in α - versus β -cells. Here, we demonstrate for the first time that altered DNA methylation in/near DMRs causes changes in gene expression of *INS* (encoding insulin), *GCG* (encoding glucagon) and *TH* (encoding tyrosine hydroxylase), and protein levels of insulin and glucagon in human β -cells. Furthermore, we performed deep phenotyping of eight T2D-associated DEGs identified in β -cells, also showing differential methylation of individual CpG sites. Here, we silenced or overexpressed the eight candidates in human islets and clonal β -cells and investigated islets from diabetic GK rats. Overall, these experiments show that overexpression of *ONECUT2* causes impaired glucose-stimulated insulin secretion, via reduced ATP production and mitochondrial function. Additionally, while our study describes the global methylome in human α - and β -cells, the study by Manduchi et al does not. Our study also includes an open web-tool where users can investigate the impact of T2D, age or sex on the methylome and/or expression of any gene of interest in human α - and β -cells. Together, our study and the study by Manduchi *et al* (PMID: 40631278) include both different and some complementary novel results.

**Point-by-point response to the comments by the Reviewers and Editors
regarding the manuscript:**

NATMETAB-RS250816494A

“Cell-specific DNA methylation in human pancreatic α - and β -cells controls gene expression and impacts type 2 diabetes”

Thank you for submitting your revised manuscript "Cell-specific DNA methylation in human pancreatic α - and β -cells controls gene expression and impacts type 2 diabetes" (NATMETAB-RS250816494A). It has now been seen by the original referees and their comments are below. The reviewers find that the paper has improved in revision, and therefore we'll be happy in principle to publish it in Nature Metabolism, pending minor revisions to satisfy the referees' final requests and to comply with our editorial and formatting guidelines.

-- Please address the issues regarding the website raised by Reviewers 1 and 3. A fully functioning website will be a pre-requisite to acceptance of the manuscript.

Response: We apologize for the problem which we have thoroughly addressed. Due to a difference in how text is interpreted by the operating system on the machine used for the development of the web application and the host server, an unforeseen error occurred that now has been addressed. We have also comprehensively tested the website on different computers without any problems.

Furthermore, based on a review comment, additional functionality has been added to the web application. For better user experience, we added a slider, allowing a more intuitive way of viewing the data.

-- Please incorporate the textual changes suggested by Reviewers 1 and 2

Response: We have incorporated all textual changes suggested by Reviewers 1 and 2. These changes are marked with red text in the revised manuscript.

-- Some reviewers highlighted in their confidential comments that there is some overlap but also some differences in DEGs obtained in your work vs others. Based on these comments, it will be useful to highlight the inherent heterogeneity associated with T2D and how that in itself affects reproducibility in these types of studies, despite technical/methodological soundness.

Response: Based on this good suggestion, we added the following sentence to page 20 in the discussion.

Page 20: *“Additionally, since T2D is a heterogenous and polygenic disease, diverse mechanisms and different genes contribute to the disease in different groups of people^{2,77}. This may affect the reproducibility between some studies, despite technical soundness.”*

-- When citing the paper from the Kaestner lab, please do not cite the preprint but instead use a placeholder citation which will be updated by our team: ‘Manduchi et al ‘Epigenetic adaptation of beta cells across lifespan and disease: age-related demethylation is advanced in type 2 diabetes’, Nature Metabolism (2026)’

Response: We appreciate this comment, and on page 6 and page 20 of the revised manuscript we no longer cite the preprint, and we only use a placeholder citation (reference 28) that can be updated by the Nature Metabolism team. Klaus Kaeastner has also confirmed that the tables we refer to on page 6 have not been updated. We also added some more text on page 19 in the discussion, comparing our study with the paper by Manduchi et al.

Response: We provide editable Microsoft Word format.

Reviewer #1 (Remarks to the Author):

Major Points:

There are several issues with the website. I accessed the tool using Google Chrome, the most updated version, and encountered an error message when attempting to analyze alpha versus beta DMRs. Additionally, the color for age DMRs is not clearly defined.

It would be beneficial to align the chromosomal position numbers with the gene layout in the application for improved user comprehension.

Response: We thank reviewer 1 for bringing this to our attention and we apologize for the problem. Due to a difference in how text is interpreted by the operating system on the machine used for the development of the web application and the host server, an unforeseen error occurred which has now been resolved.

Furthermore, a legend explaining the colors in the age analysis have been added to the plot.

We acknowledge that it would be beneficial to align the gene layout to the chromosomal position. However, due to the limitation of processing power in combination with the complexity of multiple transcripts being annotated to the same region, adding the gene layout to the plots will not be possible at this time.

Minor Points:

Line 540—It appears that "recourse" should be corrected to "resource."

Response: We thank reviewer 1 for spotting this mistake, which has been fixed (see page 19).

Reviewer #2 (Remarks to the Author):

In this revised manuscript, Ofori and colleagues present a useful resource to look at the methylome/gene expression changes in alpha and beta cells from control and T2D subjects. This reviewer's main concern for the first version is the clarity of statistical analysis and the context of the presented studies against the field. In this revision, the authors have added a substantial amount of work to address these concerns. No more main issue is left.

Response: We thank reviewer 2 for helping us improve our manuscript, and we are very happy to see that he/she believes we successfully responded to most comments.

A few minor comments are (no need for re-review):

1) Throughout the manuscript, it appears that the authors use "expression" to refer to mRNA levels. Because sometimes the authors also examined protein levels (also considered "expression"), it might be better to replace "expression" with "transcript level", or define "expression" ahead of time.

Response: We thank reviewer 2 for this this comment. Throughout the text, we tried to specify when we refer to mRNA expression, and we write "gene expression" and/or use italic gene symbols. Based on this valid comment, in the revised manuscript, we have added

“gene” where this was missing (see page 4, 8, 9, 12, 13, 17, 19, 21, 23, 34). We also specify when we refer to protein expression. To further clarify, on pages 23, 27 and 28 in the methods we added the following text:

Page 23: “*Gene symbols referring to human mRNA expression are capitalized and italicized, and mRNA expression is referred to as “gene expression”.*”

Page 27: “*Gene symbols referring to human mRNA expression are capitalized and italicized, whereas when referring to human protein levels, protein symbols are capitalized but not italicized.*”

Page 28: “*Gene symbols denoting rodent mRNA expression are written in italics with only the initial letter capitalized; rodent protein symbols follow the same capitalization but are not italicized.*”

2) The studies using dCas9-DNMT3a and dCas9-Tet1 for manipulating site-specific methylation are interesting and the technique is useful. The degree of forced methylation here appears lower than in some previous studies (e.g., PMC6327977, which methylates nearly all CpG dinucleotides close to the sgRNA). This may merit a comment somewhere: the more physiological levels of methylation method described here, using viral vectors, might be more useful than the previous plasmid-delivery approach.

Response: We thank reviewer 2 for this good suggestion. However, after careful reading of the study, PMC6327977, we could not find any information on where exactly their sgRNA was placed, and how far it is away from the CpG sites they display, and where exactly the displayed CpG sites are located. Considering this, we are unable to comment on whether nearly all CpG sites close to the sgRNA were methylated by epigenetic editing in their study.

Regarding our study, as presented in the results and methods sections, we used pyrosequencing to analyze DNA methylation of CpG sites covering the genomic regions targeted by and surrounding the three investigated gRNAs. However, several pyrosequencing assays covering the genomic regions targeted by and surrounding the three investigated gRNAs failed, potentially due to CG rich regions. Subsequently, we were unable to analyze methylation of all CpG sites in these regions. Hence, we cannot exclude that the sites, not possible to analyze, were affected by epigenetic editing. Additionally, both CpG sites that we analyzed and that are directly within the locus that is targeted by the *INS*-DMR-sgRNA were affected by the editing (see Fig 4g).

We also appreciate the comment regarding different delivery approaches. We have explored different ways of delivering the gRNA / CRISPR-dCas9-epigenetic enzyme plasmids and found that β -cells are very difficult to transfect with these larger constructs using classical approaches e.g. lipofectamine (unpublished data). Hence, we started using viral vectors, which work much better. We are also developing new delivery methods using nanotechnology (see e.g., doi: 10.1039/d4ra02791d and doi: 10.1039/d5nr02352a). Our conclusion from these experiments is that different cell types are differently hard to transfect with larger constructs such as the gRNA / CRISPR-dCas9-epigenetic enzyme plasmids. Hence, at this stage, we believe more work is needed, exploring the effects of different delivery systems in different cell types, and we would rather refrain from adding a comment suggesting that viral vectors are mediating a more physiological effect when actively rewriting the DNA methylation signature compared with other delivery systems.

3) There might be a type in Fig. 4b and 4h: is cg10781463 or cg10781483 the correct number?

Response: We thank the reviewer for noticing this and have changed cg10781483 to cg10781463 in Figure 4b.

Reviewer #3 (Remarks to the Author):

The authors have addressed my concerns and I support publication. Note that the apha-beta-methylome website is not fully functional (e.g a comparison of alpha vs beta cells gives an error message) – please fix before publication, as I expect this resource to be popular. Congratulations!

Response: We thank reviewer 3 for spotting the problem with the website and we apologize for the problem. Due to a difference in how text is processed on the machine used for development of the web application and the host server an error occurred that now has been resolved.